# Niche-specific dermal macrophage loss promotes skin capillary ageing

Kailin R. Mesa[1✉], Kevin A. O'Connor[1], Charles Ng[2], Steven P. Salvatore[2], Alexandra Dolynuk[1], Michelle Rivera Lomeli[1] & Dan R. Littman[1,3,4✉]

All mammalian organs depend on resident macrophage populations to coordinate repair and facilitate tissue-specific functions[1–3]. Functionally distinct macrophage populations reside in discrete tissue niches and are replenished through a combination of local proliferation and monocyte recruitment[4,5]. Declines in macrophage abundance and function have been linked to age-associated pathologies, including atherosclerosis, cancer and neurodegeneration[6–8]. However, the mechanisms that coordinate macrophage organization and replenishment within ageing tissues remain largely unclear. Here we show that capillary-associated macrophages (CAMs) are selectively lost over time, contributing to impaired vascular repair and reduced tissue perfusion in older mice. To investigate resident macrophage behaviour in vivo, we used intravital two-photon microscopy in live mice to non-invasively image the skin capillary plexus, a spatially well-defined vascular niche that undergoes rarefaction and functional decline with age. We find that CAMs are lost at a rate exceeding capillary loss, resulting in macrophage-deficient vascular niches in both mice and humans. CAM phagocytic activity was locally required to repair obstructed capillary blood flow, leaving macrophage-deficient niches selectively vulnerable under homeostatic and injury conditions. Our study demonstrates that homeostatic renewal of resident macrophages is less precisely regulated than previously suggested[9–11]. Specifically, neighbouring macrophages do not proliferate or reorganize to compensate for macrophage loss without injury or increased growth factors, such as colony-stimulating factor 1 (CSF1). These limitations in macrophage renewal may represent early and targetable contributors to tissue ageing.

Tissue homeostasis is dependent on multiple macrophage populations that reside in distinct sub-tissue compartments or niches, such as epithelia, blood vessels or nerves, and are thought to support specialized tissue functions[4,12,13]. Recent research has suggested that functional decline and rarefaction of vascular niches may contribute to various age-associated tissue pathologies (including sarcopenia, chronic wounds and Alzheimer's disease)[14–16]. It is unclear how tissue-resident macrophages resist or potentiate such niche-specific ageing processes[6,17].

## Skin capillary macrophages are lost with age

To model mammalian tissue ageing, we adapted an intravital microscopy technique to visualize skin-resident macrophage populations non-invasively in live mice throughout the lifetime of the organism[18,19] (Fig. 1a and Supplementary Videos 1 and 2). Unexpectedly, longitudinal imaging of skin macrophages (marked by $Csf1r^{eGFP}$) revealed a niche-specific decline in macrophage populations. Subdividing the skin into three anatomical layers—the epidermis, and upper (papillary) and lower (reticular) dermis—we observed that macrophages of the upper

dermis were lost with age at a greater rate than macrophages from both the epidermis and lower dermis (Fig. 1b,c). Further characterization of this upper-dermal CSF1R+ population revealed expression of additional macrophage markers[20–22], including the chemokine receptor CX₃CR1, lysozyme M (LysM) and the mannose receptor CD206 (Extended Data Fig. 1a,b,f–h). A major component of the upper dermal niche is the superficial capillary plexus, which supplies nutrient exchange for the overlying epidermis. To visualize this structure, we used third harmonic generation from our imaging to track red blood cell (RBC) flow through capillary vessels[23,24], which was comparable to conventional rhodamine dextran labelling (Extended Data Fig. 2 and Supplementary Video 3) and was not significantly altered when using a coverslip during our imaging sessions (Extended Data Fig. 3a,b). With this in vivo marker of RBC flow, we found that these macrophages were closely associated with blood capillaries of the superficial plexus, suggesting that they may provide support for this capillary niche (Extended Data Fig. 1c–e and Supplementary Video 4). To investigate whether these macrophages have a role in capillary function, we assessed whether RBC blood flow was altered in the presence of CAMs. Performing time-lapse

[1]Department of Cell Biology, New York University School of Medicine, New York, NY, USA. [2]Department of Pathology and Laboratory Medicine, Weill Cornell Medicine, New York-Presbyterian Hospital, New York, NY, USA. [3]Perlmutter Cancer Center, New York University Langone Health, New York, NY, USA. [4]Howard Hughes Medical Institute, New York, NY, USA. ✉e-mail: kai.mesa@med.nyu.edu; dan.littman@med.nyu.edu

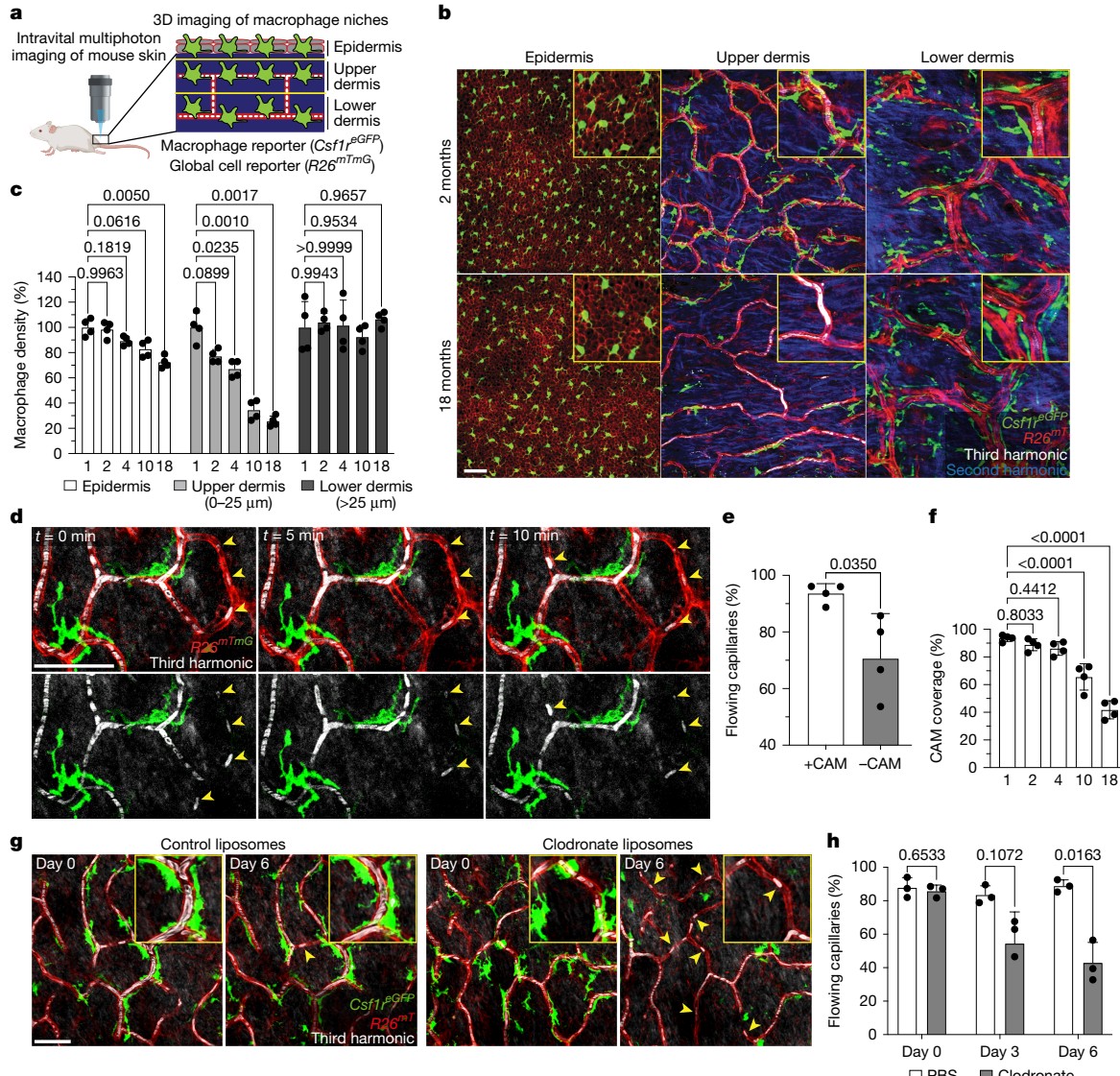

**Fig. 1 | Niche-specific macrophage loss with age correlates with impaired skin capillary blood flow. a**, Intravital imaging schematic of resident macrophage populations in mouse skin using the macrophage reporter *Csf1r^eGFP* in combination with the universal cell membrane reporter *R26^mTmG*. The diagram was created using BioRender. **b**, Representative optical sections showing distinct epidermal and dermal macrophage populations in young (aged 2 months) and old (aged 18 months) mice. **c**, Quantification of the macrophage density in distinct skin niches across ages (1, 2, 4, 10 and 18 months). *n* = 4 mice per age; two 500 μm² regions per mouse. Statistical analysis was performed using two-way repeated-measures analysis of variance (RM-ANOVA) with Tukey's test. Data are mean ± s.d. **d**, Skin resident macrophage labelling using *Cx3cr1^creERT2 R26^mTmG* mice was performed after a single high-dose intraperitoneal injection of tamoxifen (2 mg) in 1-month-old mice. Single optical sections at successive timepoints 5 min apart showing RBC flow (white) in capillaries (red) with or without nearby CAMs (green). The yellow arrowheads

indicate obstructed RBC capillary flow. **e**, Quantification of capillaries with blood flow as measured by stalled RBCs (described in Extended Data Fig. 2). *n* = 226 (CAM⁺) and *n* = 27 (CAM⁻) capillary segments; *n* = 4 mice. Capillary blood flow (CAM⁺ versus CAM⁻) was compared using paired Student's *t*-tests. Data are mean ± s.d. **f**, The percentage of capillary segments with at least one associated macrophage across age groups. *n* = 4 mice per group; two 500 μm² regions per mouse. Statistical analysis was performed using one-way ANOVA with Tukey's test. Data are mean ± s.d. **g**, Representative images showing macrophage depletion after intradermal clodronate-liposome injections every 3 days. Repeated imaging visualized macrophages (*Csf1r^eGFP*), capillaries (*R26^mTmG*) and RBC flow (third harmonic). **h**, The percentage of capillaries with blood flow after macrophage depletion. *n* = 194 (clodronate) and *n* = 199 (PBS) capillary segments; *n* = 3 mice per group. Statistical analysis was performed using two-way RM-ANOVA with Tukey's test. Data are mean ± s.d. Scale bars, 50 μm.

recordings of fluorescently labelled CAMs in 2-month-old mice with a Cre-dependent dual reporter system (*Cx3cr1^creERT2 R26^mTmG*), we found that capillaries lacking an associated macrophage had a higher rate of obstructed RBC flow (Fig. 1d,e and Supplementary Video 5). We next performed longitudinal analysis across multiple ages (1–18 months) to assess CAM coverage and capillary function during physiological ageing. We found across all ages tested a significant loss in RBC blood flow in capillaries lacking associated macrophages (Extended Data Fig. 3c). Furthermore, we found that the fraction of capillaries with

an associated macrophage significantly decreased with age (Fig. 1f). This decrease in CAMs and coverage outpaced the loss of capillaries (Extended Data Fig. 3e,f), which was previously shown to be an early hallmark of mouse and human ageing in multiple tissues, including the central nervous system, lung, kidney and skin[14,15,25–31]. Thus, to assess whether this phenomenon also occurs in humans, we examined skin samples of both young (<40 years old) and older (>75 years old) patients. Consistent with our observations in mice, human capillary-associated macrophages also displayed a decline with age. Moreover, CAM decline

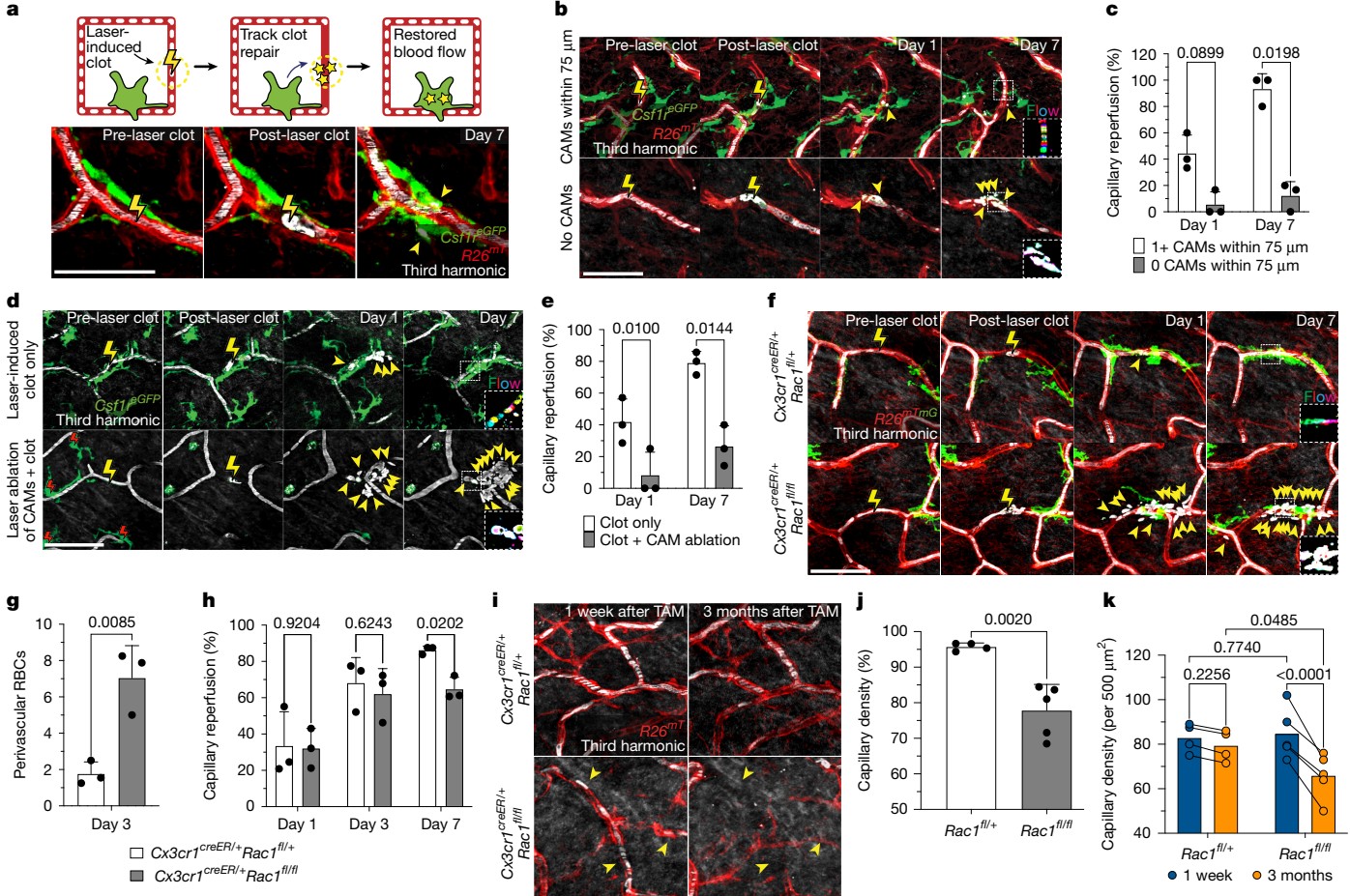

**Fig. 2 | CAMs are required to clear vascular damage and preserve skin capillaries during ageing. a**, Schematic of laser-induced capillary clotting (top). Bottom, time-lapse imaging of *Csf1r^eGFP^R26^mTmG^* mice after clot formation (yellow lightning bolt, 940 nm, 1 s). **b**, Laser-induced clot formation in 10-month-old mice. The yellow arrowheads indicate extraluminal vascular debris. The dotted white box highlights third harmonic optical *z*-series, pseudocoloured to visualize RBC movement in recovering capillaries. **c**, Clot recovery in 10-month-old mice. Quantification of reperfusion at days 1 and 7 after clotting in regions with CAMs (<75 μm) versus without CAMs (>75 μm) is shown. *n* = 16 clots per group; 3 mice. Statistical analysis was performed using two-way RM-ANOVA with Fisher's test. Data are mean ± s.d. **d**, Imaging of clot repair after simultaneous laser ablation of CAMs (red lightning bolt) and capillary clotting in *Csf1r^eGFP^* mice (940 nm, 1 s). **e**, Quantification of reperfusion in CAM-ablated versus control regions at days 1 and 7. *n* = 19 (CAM ablated) and

16 (control) clots; 3 mice total. Statistical analysis was performed using two-way RM-ANOVA with Fisher's test. Data are mean ± s.d. **f**, Repeated imaging of capillary clot recovery in *Cx3cr1^creER^Rac1^fl/fl^* and *Cx3cr1^creER^Rac1^fl/+^* mice. CAMs (green), capillaries (red) and RBCs (white) are visualized. **g**, Quantification of perivascular RBC debris at day 3. *n* = 52 (*Rac1^fl/+^*) and 48 (*Rac1^fl/fl^*) clots; 3 mice per group. Statistical analysis was performed using unpaired *t*-tests. Data are mean ± s.d. **h**, Capillary reperfusion at days 1, 3 and 7. *n* = 67 (*Rac1^fl/+^*) and 82 (*Rac1^fl/fl^*) clots; 3 mice per group. Statistical analysis was performed using two-way RM-ANOVA with Tukey's test. **i**, Sequential revisits over 3 months show pruning of individual capillaries (yellow arrowheads). **j,k**, Quantification of homeostatic capillary loss 3 months after tamoxifen administration. *n* = 4 (*Rac1^fl/+^*) and *n* = 5 (*Rac1^fl/fl^*) mice; two 500 μm² regions per mouse. Statistical analysis was performed using unpaired *t*-tests (**j**) and two-way RM-ANOVA with Fisher's test (**k**). Data are mean ± s.d. Scale bars, 50 μm.

also outpaced capillary loss with age, suggesting a similar loss in macrophage coverage of the capillary niche (Extended Data Fig. 3g–j). These observations therefore suggest that local macrophage loss in both mice and human may contribute to impaired capillary function with age. To assess any functional role of CAMs in maintaining capillary blood flow, we performed chemical and genetic ablation of CAMs through clodronate liposomes and *Cx3cr1^DTR^* depletion, respectively, and observed both acute CAM loss as well as a loss in capillary flow (Fig. 1g,h and Extended Data Fig. 4). Collectively, this highlights an evolutionarily conserved loss in skin-capillary-associated macrophages with age, which correlates with impaired homeostatic capillary perfusion.

## CAMs required for capillary repair and preservation

Given the link between CAMs and capillary blood flow, we next assessed the long-term fate of vessels that lack an associated macrophage.

To this end, we performed a 6-month time-course analysis in *Cx3cr1^GFP^R26^mTmG^* mice from 1 to 7 months of age (Extended Data Fig. 5a) and found that capillaries fated for pruning in the 6 month time course had decreased CAM coverage compared with capillaries that were maintained (Extended Data Fig. 5b,c). This finding suggests that CAMs might be locally required to maintain proper capillary function and preservation with age. Thus, to assess the cellular mechanism(s) by which CAMs support capillary function, we used a laser-induced blood-clotting model to precisely target and stop blood flow in individual capillary segments (Fig. 2a, Extended Data Fig. 6 and Supplementary Video 6). To assess macrophage involvement, we tracked the daily displacement of surrounding CAMs to laser-induced clots. These data showed that CAM recruitment to sites of capillary damage as well as RBC engulfment are locally restricted to within approximately 80 μm and largely occur within the first 2 days after injury (Extended Data Fig. 5d–f). This is consistent with previous work describing macrophage cloaking as an

acute behavioural response to laser-induced tissue damage[32,33]. Multiple signalling pathways are involved in sensing tissue damage[32,34,35], including the chemokine receptor CX3CR1, which is homeostatically expressed by CAMs (Extended Data Fig. 1a–f). We therefore examined the role of CX3CR1 signalling in recruiting nearby CAMs to capillary damage. We performed laser-induced clotting and found significant impairment in CAM recruitment in *Cx3cr1*[GFP/GFP] mice compared with in the *Cx3cr1*[GFP/+] control mice (Extended Data Fig. 5g,h). We also found that CX3CR1 deficiency led to a significant delay and overall reduction in capillary reperfusion by day 7 after clot induction (Extended Data Fig. 5i). We next assessed capillary RBC flow during physiological ageing and found a significant loss in flowing capillaries in *Cx3cr1*[GFP/GFP] mice compared with in the *Cx3cr1*[GFP/+] controls by 6 months of age (Extended Data Fig. 5j). Together, our results support a model in which local CAM recruitment, in part through CX3CR1 signalling, is critical for both capillary repair and the long-term preservation of the vascular network during physiological ageing.

To determine whether diminished capillary function with age is directly related to macrophage loss, we took advantage of the inherent variability in CAM density in older mice. Specifically, we performed laser-induced clotting of aged skin capillaries with or without local CAMs (within 75 μm from the clot) (Fig. 2b). Notably, capillaries that retained local CAMs in aged mice were significantly better at re-establishing blood flow compared with capillaries without local CAMs in the same mice (Fig. 2c). This result suggests that local macrophage loss may drive age-associated capillary dysfunction. To directly test the role of macrophages in capillary repair, we performed laser-induced ablation of local CAMs immediately before capillary clot induction. Indeed, in regions where CAMs were ablated, repair was significantly impaired, and the blood flow was not properly re-established (Fig. 2d,e). We targeted other perivascular cells with the same laser-ablation conditions and found no impairment to capillary repair, nor did we detect neutrophil swarming as has been described for larger areas of laser-induced damage[32,33,36–39] (Extended Data Figs. 7 and 8a–c).

From our serial imaging of capillary repair, we found nearby CAMs often surrounded by and containing RBC debris. To mechanistically understand whether CAM uptake and clearance of this vascular debris is functionally important for capillary repair, we acutely impaired CAM phagocytosis through inducible Cre-dependent knockout of *Rac1*, a critical component of the phagocytic machinery[40], 1 week before laser-induced clot formation. Although there was no significant change in CAM density (Extended Data Fig. 8d,e), there was significant impairment in the clearance of RBC debris and capillary reperfusion in *Cx3cr1*[creER]*Rac1*[fl/fl] mice compared with in the *Cx3cr1*[creER]*Rac1*[fl/+] littermate controls (Fig. 2f–h), suggesting that RAC1-dependent phagocytic clearance is critical for proper capillary repair and tissue reperfusion.

To test the long-term vascular effects of RAC1-deficiency in CAMs, we assessed the capillary density 3 months after tamoxifen administration in *Cx3cr1*[creER]*Rac1*[fl/fl] mice. We found an acceleration in the rate of capillary pruning after 3 months of physiological ageing in the skin of *Cx3cr1*[creER]*Rac1*[fl/fl] mice compared with in the *Cx3cr1*[creER]*Rac1*[fl/+] littermate controls (Fig. 2i–k). It is therefore likely that the loss of RAC1-dependent behaviours, such as phagocytosis of vascular debris, directly contributes to impaired recovery and preservation of the skin microvascular network.

Together, our results support a model in which local CAM recruitment and phagocytic clearance of vascular debris is critical to maintain capillary function. Thus, as CAM density declines with age, so does capillary perfusion of the tissue.

## Niche-specific dermal macrophage renewal

Maintenance of tissue-resident macrophage populations in the skin is thought to be mediated through a combination of local proliferation and systemic replacement by bone marrow (BM)-derived blood monocytes[13,41,42]. To understand how CAMs are replenished under physiological conditions, we generated BM chimeras with *Csf1r*[eGFP]*CAG*[dsRed] BM transferred into lethally irradiated *Csf1r*[eGFP] mice. Importantly, the hind paws of these mice were lead-shielded to prevent loss of resident macrophage populations from our imaging area (Fig. 3a). Tracking macrophage populations from all anatomical layers of the skin for 10 weeks after transplantation showed that, while most lower dermal macrophages are replaced by monocytes, <5% of upper dermal macrophages were BM derived (Fig. 3b,c). Despite limited monocyte renewal, the few capillaries with BM-derived (GFP[+]dsRed[+]) CAMs showed no obvious differences in RBC blood flow compared with capillaries with host-derived (GFP[+]dsRed[−]) CAMs (Extended Data Fig. 3d), suggesting that CAM ontogeny may not have a major role in homeostatic capillary function.

Given that nearly all CAMs remained host derived, we next performed single-macrophage lineage tracing to monitor the local proliferation of dermal macrophages. Specifically, we induced sparse Cre recombination in 1-month-old *Cx3cr1*[GFP]*R26*[mTmG] mice to label and track individual macrophages from both the upper and lower dermis over weekly revisits (Fig. 3d,e). From these serial revisits, we observed marked positional stability of macrophages on the vascular network as well as local cell division (Fig. 3f). Analysis of the monthly rates of macrophage loss and division revealed that lower-dermal macrophages have a significantly lower division rate as well as a significantly higher loss rate compared with to CAMs (upper-dermal macrophages) (Fig. 3g).

Focusing on CAM turnover, we also found a significant skew toward cell loss rather than proliferative balance during the 4-month time course (Fig. 3h). Consistent with these data, there was a significant decline in the fraction of fate-mapped CAMs over the 20-week time course (Fig. 3i). These results demonstrate that CX3CR1[+] dermal macrophage self-renewal strategies are niche-specific, where blood monocyte recruitment is used in the lower dermis and local cell division is used in the upper dermis. Importantly, we find that CAMs from the upper dermis are insufficiently replenished by local proliferation, which contributes to their progressive decline in this tissue niche (Fig. 3j).

## CAM loss is insufficient to promote renewal

The ability of resident macrophages to locally self-renew has largely been studied through methods of near-total macrophage depletion that have limited niche specificity and often generate tissue-wide inflammation[9–11]. To directly interrogate the steps of macrophage self-renewal over time, we tracked both the replacement of individual macrophages after loss and the redistribution of sister macrophages after division. First, to track local macrophage replacement after CAM loss, we performed laser-induced ablation of all CAMs within a defined 500 μm² region. Serial revisits after ablation revealed limited repopulation from adjacent capillary regions that retained intact CAM populations at 2- and 8-weeks after ablation (Fig. 4a,d) as well as a reduction in local capillary blood flow by 8-weeks after ablation (Extended Data Fig. 9a). We found a similar lack in repopulation after partial CAM depletion in *Cx3cr1*[DTR] mice following administration of low-dose diphtheria toxin (Fig. 4b,d). To avoid any non-physiological effects from these cell depletion models, we developed a dual-fluorescent macrophage reporter mouse, *Cx3cr1*[creERT2]*Rosa26*[dsRed]*Csf1r*[eGFP], allowing for Cre-dependent recombination to differentially label a small fraction of macrophages and track their homeostatic replacement. Examination on serial weekly revisits indicated that, as we observed in depletion models, most capillary niches did not recruit a new macrophage for at least 2 weeks after CAM loss (Fig. 4c,d). By contrast, when we used large laser-induced damage (500 μm² region) in either the upper dermis or overlaying epidermis (Fig. 4e and Extended Data Fig. 9b), we found CAMs were readily replenished (Fig. 4e–g) in a CCR2-dependent manner, suggesting at least partial monocyte repopulation of CAMs after tissue damage.

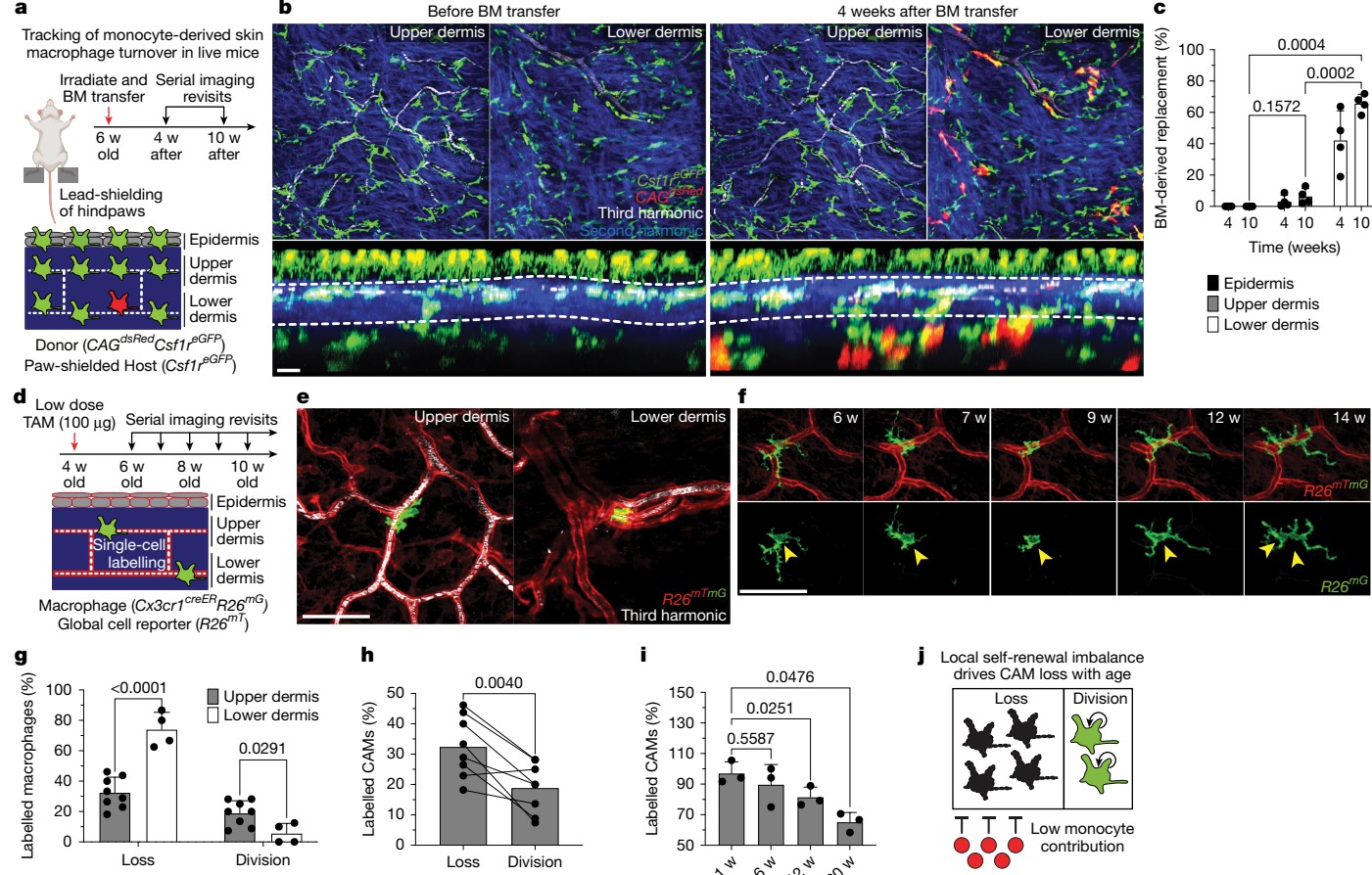

**Fig. 3 | Dermal macrophages use niche-specific self-renewal strategies leading to selective CAM loss with age. a**, Schematic of serial intravital imaging in BM chimeras: *Csf1r*^*eGFP*^*CAG*^*dsRed*^ BM was transplanted into lethally irradiated *Csf1r*^*eGFP*^ mice; the hind paws were lead-shielded to preserve resident macrophages in the imaging area. The diagram was created using BioRender. **b**, Representative optical sections of dermal macrophage before and 4 weeks (4 w) after BM transfer. **c**, Quantification of macrophage repopulation at 4 and 10 weeks after BM transplantation. *n* = 4 mice; four 500 μm² regions per skin compartment per mouse. The percentage of BM-derived GFP⁺dsRed⁺ macrophages was compared across skin compartments at 10 weeks using two-way RM-ANOVA with Tukey's test. Data are mean ± s.d. **d**, Schematic of long-term in vivo macrophage lineage tracing. **e**, Representative upper- and lower-dermis images from *Cx3cr1*^*creERT2*^*R26*^*mTmG*^ mice after intraperitoneal injection of low-dose tamoxifen (50 μg), showing single labelled macrophages

followed weekly over 20 weeks. **f**, Upper dermal CAM local proliferation during homeostasis. Serial revisit images of individual traced macrophages. **g**, Quantification of monthly division and loss rates in the upper (*n* = 262 cells, 8 mice) versus lower (*n* = 42 cells, 4 mice) dermis. Statistical analysis was performed using two-way RM-ANOVA with Fisher's test. Data are mean ± s.d. **h**, Monthly CAM-specific turnover rates (*n* = 262 CAMs, 8 mice) were compared using paired Student's *t*-tests. Data are mean with paired lines. **i**, Lineage-tracing analysis of labelled CAMs over 20 weeks. *n* = 59 CAMs; 3 mice. Statistical analysis was performed using one-way RM-ANOVA with Tukey's test. Data are mean ± s.d. **j**, Model: CAMs in the upper dermis rely on local proliferation for maintenance, in contrast to lower-dermal macrophages, which are replenished by monocytes. With age, CAM division becomes insufficient, leading to progressive CAM loss. Scale bars, 50 μm.

We also found that local CAM proliferation was significantly increased 1 week after large laser-induced damage in either the upper dermis or epidermis (Fig. 4h,i and Extended Data Fig. 9b–d). Taken together, these results suggest that CAM loss alone is not a sufficient trigger to promote macrophage replenishment and requires additional signals elicited by tissue damage.

Second, to precisely track whether proliferating CAMs readily redistribute across the capillary network after cell division, we used mice with a dual-fluorescent nuclear reporter, *Cx3cr1*^*creERT2*^*R26*^*nTnG*^, and administered a low or high dose of tamoxifen to label either a sparse subset or all CAMs, respectively (Extended Data Fig. 9e). Compared with the average distance between all nearest neighbouring macrophages, sister CAMs remained significantly closer (<15 μm) to each other for at least 2 weeks after division (Extended Data Fig. 9f,g). These findings further support the notion that CAM division and loss are not spatiotemporally coupled, which we predict would progressively lead to disorganized patterning and the accumulation of both empty and

crowded capillary regions. We tested this prediction by looking at the distribution of neighbouring CAMs in both young (2-month-old) and old (10-month-old) mice. In young mice, the majority of macrophages was within 50 μm of each other. By contrast, old mice had a biphasic distribution of macrophage patterning, with most CAMs either within 25 μm or further than 75 μm apart (Extended Data Fig. 9h). These results highlight two distinct cellular features that contribute to reduced CAM coverage with age: (1) insufficient macrophage repopulation after CAM loss; and (2) insufficient redistribution of these cells along the capillary niche, which may promote progressive erosion of the vascular network (Extended Data Fig. 9i).

## CAM renewal boosts aged capillary repair

While homeostatic CAM proliferation was insufficient to maintain a stable population density, we examined whether extrinsic cues such as large tissue damage could increase CAM density long term to

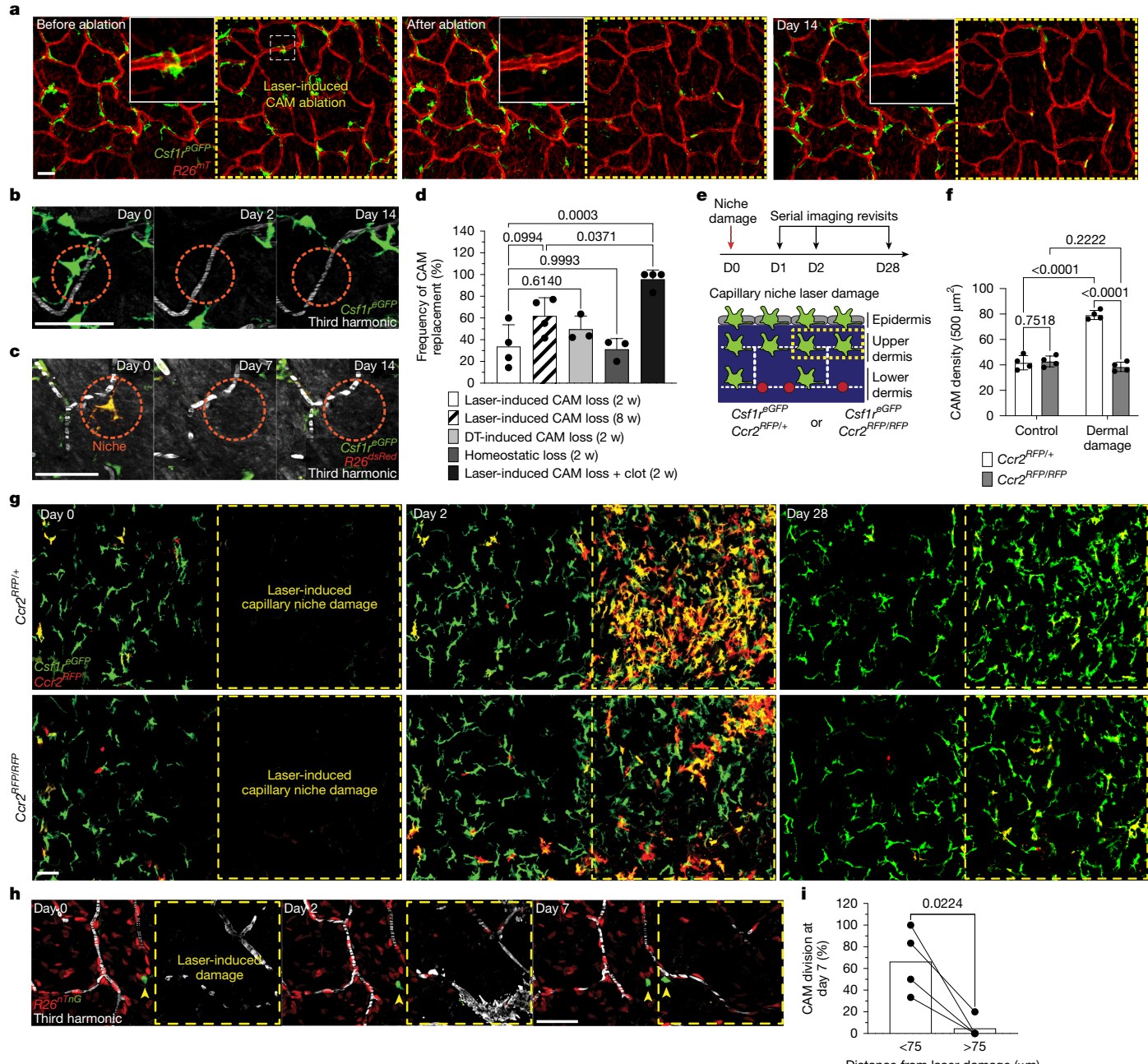

**Fig. 4 | Macrophage loss without local tissue damage is not sufficient to promote CAM renewal. a**, Representative time-lapse imaging of CAM replacement after targeted laser ablation (yellow asterisk) in $Csf1r^{eGFP}R26^{mTmG}$ mice. Ablation was performed within a 500 µm² region (yellow dashed square). **b**, CAM replacement after systemic depletion through intraperitoneal injection of diphtheria toxin (DT) (25 ng per g body weight) in $Cx3cr1^{DTR}$ mice. **c**, Serial imaging of single lineage-traced macrophages in $Cx3cr1^{creERT2}R26^{dsRed}Csf1r^{eGFP}$ mice under homeostasis. Mice received a single low-dose tamoxifen intraperitoneal injection (50 µg), and were imaged weekly for 2 weeks. **d**, Quantification of CAM niche replenishment under different conditions: 2-week laser-induced loss ($n = 44$ CAMs, 4 mice), 8-week laser-induced loss ($n = 261$ CAMs, 4 mice), DT-induced loss ($n = 53$ CAMs, 3 mice), homeostatic loss ($n = 29$ CAMs, 3 mice) or laser-induced CAM loss with capillary clot ($n = 25$ CAMs, 4 mice). Data were analysed using one-way ANOVA with Tukey's test. Data are mean ± s.d. **e**, Schematic of CAM replacement after laser-induced CAM loss with capillary damage in $Csf1r^{eGFP}Ccr2^{RFP/+}$ and $Ccr2^{RFP/RFP}$ mice. **f**, Quantification of CAM replenishment ($Csf1r^{eGFP}$-positive cells) after niche damage at 2 weeks in $Ccr2^{RFP/+}$ versus $Ccr2^{RFP/RFP}$ mice. $n = 4$ mice per group; two 500 µm² regions per mouse. Statistical analysis was performed using two-way RM-ANOVA with Fisher's test. Data are mean ± s.d. **g**, Representative imaging of CAM replacement after regional capillary niche damage. **h**, Lineage-tracing analysis of single CAMs in $Cx3cr1^{creERT2}R26^{nTnG}$ mice after capillary injury. **i**, Quantification of CAM proliferation based on proximity to capillary damage. $n = 25$ CAMs in damaged regions; 4 mice. Statistical analysis was performed using paired Student's $t$-tests at day 7 (D7). Data are mean with paired lines. Scale bars, 50 µm.

improve capillary function in older mice. To this end, we found that large laser-induced epidermal damage (500 µm² region) resulted in a lasting increase in CAMs below the damaged regions compared with in the neighbouring control regions (Extended Data Fig. 9j–l). Furthermore, we found a significant improvement in capillary repair

in these same regions after laser-induced clotting (Extended Data Fig. 9m).

Our results suggest that CAMs in old mice can be stably expanded after environmental changes, such as tissue damage. We therefore next assessed whether directly increasing CAM density, without local

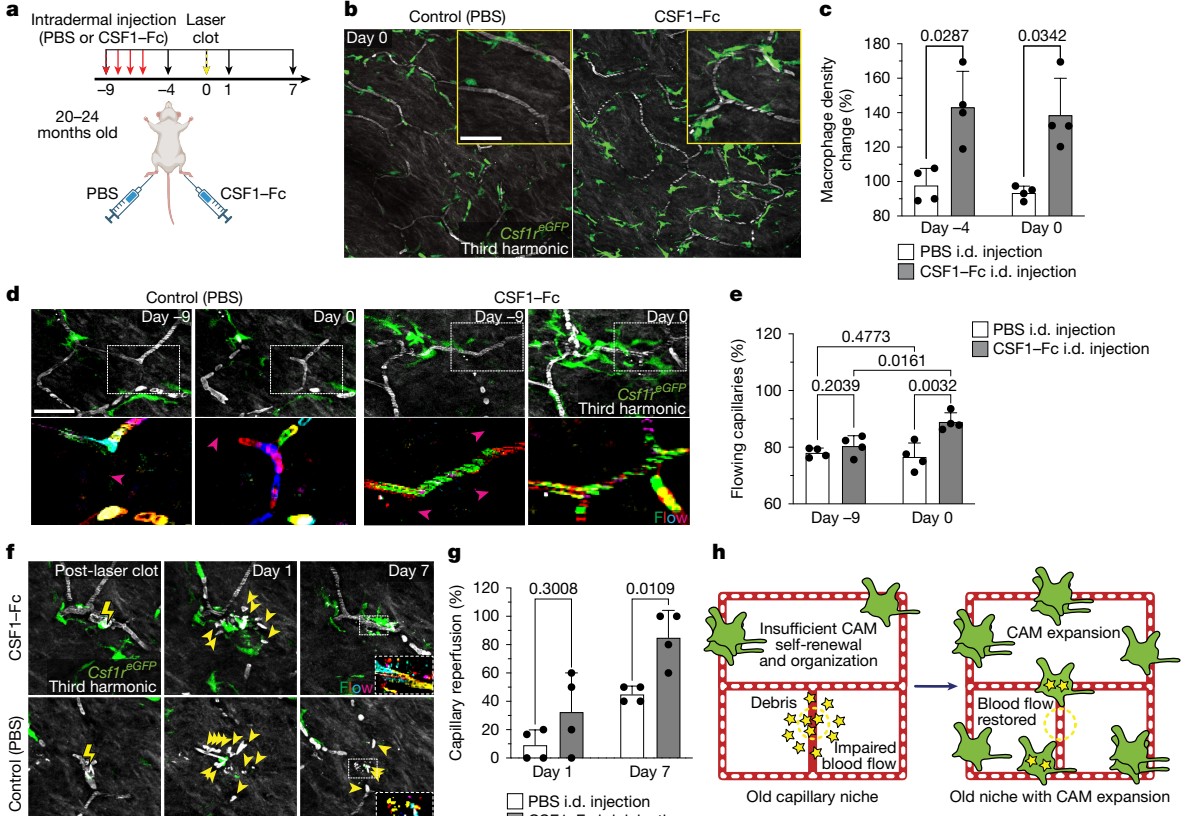

**Fig. 5 | Local CAM replenishment in old mice is sufficient to rejuvenate capillary repair and tissue reperfusion. a**, Schematic of CSF1-induced rejuvenation in the aged skin capillary niche. The diagram was created using BioRender. **b**, Representative images of CAM density in *Csf1r^eGFP* mice, 9 days after daily intradermal (i.d.) injections (4 days) of CSF1–Fc (porcine CSF1 fused to IgG1a Fc) or PBS in contralateral hind paws of 20–24-month-old mice. **c**, Quantification of CAM density change after CSF1–Fc or PBS treatment. *n* = 4 mice; two 500 μm² regions per condition per mouse; the percentage change relative to the density at day −9 was compared between day −4 and day 0 using two-way RM-ANOVA with Fisher's test. Data are mean ± s.d. **d**, Representative images of capillary blood flow (red dashed outlines) in CSF1–Fc-treated or PBS-treated regions. The magenta arrowheads indicate obstructed RBC flow. The dotted white box shows third harmonic optical *z*-series pseudocoloured to

visualize RBC movement along recovering vessels. **e**, Quantification of capillary flow after treatment. *n* = 214 segments per treatment; *n* = 4 mice. Statistical analysis was performed using two-way RM-ANOVA with Fisher's test. Data are mean ± s.d. **f**, Sequential imaging of damaged capillaries after laser-induced clotting (yellow lightning bolt) in *Csf1r^eGFP* mice. The yellow arrowheads mark extraluminal vascular debris. **g**, Quantification of reperfusion at days 1 and 7 after clotting in CSF1–Fc-treated versus PBS-treated mice. *n* = 18 (CSF1–Fc) and *n* = 20 (PBS) clots; *n* = 4 mice. Statistical analysis was performed using two-way RM-ANOVA with Fisher's test. Data are mean ± s.d. **h**, Model of resident macrophage ageing in the capillary niche. Age-associated CAM decline impairs vascular repair, which can be reversed by local macrophage expansion. Scale bars, 50 μm.

damage, would also be sufficient to improve future capillary repair and reperfusion. To this end, we used a fusion protein of the canonical macrophage growth factor CSF1 with the Fc region of porcine IgG (CSF1–Fc), as it has been reported to robustly increase macrophage density in multiple tissues, including skin[43–45]. We performed daily intradermal injections of either CSF1–Fc or PBS in the left or right hind paws, respectively, of the same mice (Fig. 5a). There was a significant increase in CAMs in the CSF1-treated paws compared with the contralateral PBS controls, which showed no significant change from before treatment (Fig. 5b,c).

To assess whether CSF1 treatment modulated local CAM survival and proliferation or simply recruited new BM-derived macrophages from blood monocytes, we performed CSF1 treatment in chimeric BM mice in which we could track the relative expansion of local and recruited populations (Extended Data Fig. 10a,b). Consistent with our previous experiments, we found that PBS-injected paws showed no significant increase in host- or BM-derived macrophages in our imaging areas (Extended Data Fig. 10c,d). Importantly we found that CSF1-induced CAM expansion did not alter the ratio of resident (GFP⁺) and recruited (GFP⁺dsRed⁺) CAMs (Extended Data Fig. 10e), as a relative increase in GFP⁺dsRed⁺ CAMs over GFP⁺ CAMs would have suggested BM-derived

monocyte recruitment. Thus, this strongly suggests that CSF1 drives local proliferation of the existing CAM population.

Notably, we found that CSF1 treatment was sufficient to improve homeostatic capillary blood flow in old mice in comparison to the PBS-treated control mice, which had significantly more obstructed capillary segments (Fig. 5d,e). Using the same aged mice, we next tested whether this increase in CAM density would be sufficient to improve capillary repair rates. After laser-induced clotting, there was a significant improvement in capillary repair and reperfusion in CSF1-treated mice compared with in the PBS-treated control mice (Fig. 5f,g), demonstrating that restoring dermal macrophage density in old mice can improve age-associated vascular dysfunction.

## Discussion

Macrophage renewal has largely been studied in non-physiological settings, such as through in vitro cell culture or severe depletion models that often are accompanied by acute inflammation[9,10,46–48]. Our work clearly demonstrates that the homeostatic renewal strategies of resident macrophages are niche specific and not as finely tuned as has been previously suggested. Specifically, we found that macrophages of the

upper dermis do not proliferate or redistribute sufficiently to maintain an optimal coverage across the skin capillary network unless they receive additional cues from acute tissue damage or increased growth factor abundance (Fig. 5h). Notably, we confirmed previous findings that show the epidermal Langerhans cell density also decreases with age, which has been associated with impaired epidermal function[49,50]. This raises the possibility that age-associated loss in macrophage density is a more general phenomenon in populations that rely on local self-renewal.

In addition to self-renewal, we also found that CAM recruitment to repair tissue damage was spatially restricted. To our knowledge, the long-term size and stability of resident macrophage territories or niches in vivo has not been reported. Moreover, our work provides strong evidence that injury-induced dermal macrophage recruitment is restricted to approximately 80 μm, as has been demonstrated in other tissues[32,33]. In young mice, CAM density is high enough to provide substantial niche/territory overlap between neighbours. However, with declining CAM density with age, we show that a significant fraction of the skin capillary network is no longer within a CAM's territory range and is susceptible to vascular damage. We find no evidence for neutrophil swarming after capillary injury in the skin, as has been described in other tissues with similar levels of acute tissue damage[32,51]. This highlights a potential limitation to our study and may suggest that a tissue's relative sensitivity to immune cell swarming may exist on a spectrum. It will therefore be important to understand whether other barrier or mucosal tissues, which are regularly exposed to environmental insults, also require higher levels of tissue damage to trigger neutrophil swarming.

We also found that dermal macrophage self-renewal and vascular support could be acutely enhanced in aged mice through local CSF1 therapeutic treatment. Multiple previous studies have demonstrated that fibroblast populations represent a major functional source of CSF1 in the skin[52,53] and are progressively lost with age[54,55]. It will therefore be important for future studies to directly assess the interplay between age-associated fibroblast and macrophage loss across different tissue microenvironments. Lastly, it will be important to understand how these properties of resident macrophages are influenced by other aspects of regional heterogeneity, such as innate immune imprinting[13], to shape local immune responses in tissues.

Collectively, this work demonstrates that loss in CAMs: (1) begins within the first few months of life; (2) is progressive throughout life; and (3) is functionally detrimental to vascular function and preservation, which has been shown to be a primary driver of age-associated tissue impairments[15,56]. Furthermore, this work provides a platform to investigate age-associated deviations in tissue homeostasis at the single-cell level in a living mammal.

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

## Methods

### Mice

Mice were bred and maintained in the Alexandria Center for the Life Sciences animal facility of the New York University School of Medicine under specific-pathogen-free conditions. Albino B6 (*B6(Cg)-Tyr^c-2J/J*, Jax, 000058), *Csf1r^eGFP* (*B6.Cg-Tg(Csf1r-eGFP)1Hume/J*, Jax, 018549), *Ccr2^RFP* (*B6.129(Cg)-Ccr2tm2.1Ifc/J*, Jax, 017586), *R26^mTmG* (*B6.129(Cg)-Gt(ROSA)26Sortm4(ACTB-tdTomato,-eGFP)Luo/J*, Jax, 007676), *R26^nTnG* (*B6N.129S6-Gt(ROSA)26Sortm1(CAG-tdTomato*,-eGFP*)Ees/J, Jax*, 023537), *LysM^cre* (*B6.129P2-Lyz2tm1(cre)Ifo/J*, Jax, 004781), *Rac1^fl/fl* (*Rac1tm1Djk/J*, Jax, 005550) and *CAG^dsRed* (*B6.Cg-Tg(CAG-DsRed*MST)1Nagy/J*, Jax, 006051) mice were purchased from Jackson Laboratories. *R26^dsRed* mice were described previously[57] and were obtained from the laboratory of G. Fishell. *Cx3cr1^creER*, *Cx3cr1^GFP* and *Cx3cr1^DTR* mice were generated in our laboratory and have been described previously[58–60]. All experimental mice for this study were albino (homozygous for *Tyr^c-2J*) as is required for intravital imaging in the skin. Mice from experimental and control groups were randomly selected from either sex for live imaging experiments. Data collection and analysis were not performed blind to the conditions of the experiments, unless otherwise stated. Cre induction for the lineage tracing or total CAM labelling experiments was induced with a single intraperitoneal injection of tamoxifen (Sigma-Aldrich, T5648) (100 µg or 4 mg in corn oil, respectively) in 1-month-old mice. *Rac1^fl/fl* recombination was induced with two intraperitoneal injections of tamoxifen (2 mg in corn oil) 48 h apart in 1-month-old mice. All imaging and experimental manipulations were performed on non-hairy mouse plantar (hind paw) skin. Preparation of skin for intravital imaging was performed as described below. In brief, mice were anaesthetized with intraperitoneal injection of ketamine–xylazine (15 mg ml⁻¹ and 1 mg ml⁻¹, respectively in PBS). After imaging, the mice were returned to their housing facility. For subsequent revisits, the same mice were processed again with injectable anaesthesia. The plantar epidermal regions were briefly cleaned with PBS pH 7.2, mounted onto a custom-made stage and a glass coverslip was placed directly against the skin. Anaesthesia was maintained throughout the course of the experiment with vaporized isoflurane delivered by a nose cone. Mice from the experimental and control groups were randomly selected for live imaging experiments. All lineage-tracing and ablation experiments were repeated in at least three different mice. All animal procedures were performed in accordance with protocols approved by the Institutional Animal Care and Usage Committee of New York University School of Medicine.

### Intravital microscopy and laser ablation

Image stacks were acquired using the Olympus multiphoton FVM-PE-RS system equipped with both InSight X3 and Mai Tai Deepsee (Spectra-Physics) tunable Ti:Sapphire lasers, using Fluoview software. For collection of serial optical sections, a laser beam (860 nm for Hoechst 33342; 940 nm for GFP, tdTomato, dsRed, RFP, rhodamine, second harmonic generation; 1,200 nm for Alexa Fluor 647; and 1,300 nm for third harmonic generation) was focused through a water-immersion lens (NA, 1.05; Olympus) and scanned with a field of view of 0.5 mm² at 600 Hz. *z* stacks were acquired in 1–2-µm steps for a ~50–100 µm range, covering the epidermis and dermis. For all animal imaging, 1 mm × 2 mm imaging fields (regions of interest) were acquired with only the second harmonic signal (collagen) as a reference guide to the same anatomical position (1 mm proximal of the most proximal walking pad on the mouse paw plantar skin). The capillary blood flow was visualized in some experiments through intravenous injection with 18 mg per kg of dextran-rhodamine 70 kDa (Sigma-Aldrich, R9379). Cell tracking analysis was performed by revisiting the same area of the dermis in separate imaging experiments through using inherent landmarks of the skin to navigate back to the original region, including the distinct organization of the superficial vasculature networks. Cells that were unambiguously separated (by at least 250 µm) from another were sampled to ensure the identity of individual lineages. For time-lapse recordings, serial optical sections were obtained between 5–10-min intervals, depending on the experimental setup. Laser-induced cell ablation, capillary clot or tissue damage was carried out with the same optics as used for acquisition. An 940 nm laser beam was used to scan the target area (1–500 µm²) and ablation was achieved using 50–70% laser power for around 1 s. The ablation parameters were adjusted according to the depth of the target (10–50 µm). Mice from experimental and control groups were randomly selected for live imaging experiments. All lineage-tracing and ablation experiments were repeated in at least three different mice.

### In situ staining of neutrophils for intravital microscopy

Anaesthetized mice were given fluorescently labelled antibodies through intravenous retroorbital injection immediately before imaging. Neutrophils were identified with 4 µg of anti-Gr1-AF647 (BioLegend, RB6-8C5, 108418) antibodies.

### Drug treatments

To induce macrophage depletion, mice received intradermal injections of either clodronate-liposomes or PBS-liposomes (stock concentration, 5 mg ml⁻¹; Liposoma, CP-005-005) (5 µl per paw) every 3 days. Depending on the experimental details, *Cx3cr1^DTR* mice received either intraperitoneal injection of diphtheria toxin (Sigma-Aldrich; D0564) every other day or a single low dose at 25 ng per g body weight in PBS. To induce macrophage expansion, mice received daily intradermal injections of CSF1-FC (Bio-Rad, PPP031) or PBS in contralateral hind paws (5 µl per paw) for 4 days.

### Generation of BM chimeric reconstituted mice

BM mononuclear cells were isolated from *Csf1r^GFP CAG^dsRed* mice by flushing the long bones. RBCs were lysed with ACK lysing buffer and the remaining cells were resuspended in PBS for retroorbital injection. $4 \times 10^6$ cells were then injected intravenously into 6–8-week-old *Csf1r^GFP* mice that were irradiated 4 h before reconstitution using 1,000 rads per mouse (500 rads twice, at an interval of 2 h, at X-RAD 320 X-Ray Irradiator). During irradiation, the hind paws of recipient mice were lead-shielded to prevent any irradiation-induced loss of resident macrophage populations from our imaging area. At 1 and 10 months after irradiation, peripheral blood samples were collected from the submandibular (facial) vein in tubes containing EDTA (BD Biosciences, Dipotassium EDTA Microtainer, 365972). RBCs were lysed with ACK lysing buffer and the remaining cells were analysed using flow cytometry on the LSR II system with FACSDiva and FlowJo v.10.10.1 software (BD Biosciences) to check for reconstitution (Extended Data Fig. 10b).

### Skin whole-mount staining

Whole skin was collected from the hind paw and fixed in 4% paraformaldehyde in PBS overnight at 4 °C, washed in PBS, permeabilized and blocked for 1 h (2% Triton X-100, 5% normal donkey serum and 1% BSA in PBS). For CD206 staining, blocked tissue was incubated in Alexa Fluor 647 rat anti-CD206 (1:500, BioLegend, C068C2) overnight at 4 °C, washed in PBS with 2% Triton X-100, washed with PBS and then mounted onto a slide with ProLong Gold antifade mounting medium (Invitrogen) with a #1.5 coverslip. Nuclear counterstaining was achieved by performing a single intravenous injection of Hoechst 33342 (15 mg per kg) 30 min before mouse euthanasia. Whole-mount skin samples were imaged with the same imaging conditions and setup that was used for intravital microscopy.

### Human skin samples

Written informed consent was obtained for post-mortem examination from next of kin for all patients. Clinical information and laboratory data were obtained from the electronic medical record. Sex and

gender information was not used. The patients in the young group were below 40 years of age (19–37 years old). The patients in the older group were aged above 75 years (79–97 years old). Patients with skin or vascular pathologies were excluded. Skin samples were obtained from the anterolateral chest and fixed in 10% formalin for at least 24 h before processing. The slides were stained with haematoxylin and eosin, CD68 (514H12) and ERG (EPR3864). Macrophages and capillaries were identified using a combination of morphology, CD68 and ERG staining. The researcher was blinded to the age group of the samples, and counting was performed on at least eight high-power fields (×40) within 100 μm of the epidermis.

### Image analysis

Raw image stacks were imported into Fiji (NIH) or Imaris (Bitplane/ Perkin Elmer) for further analysis. Provided images and Supplementary Videos are typically presented as a maximal projection of 4–8 μm optical sections. For visualizing individual labelled cells expressing the dsRed or tdTomato Cre reporters, the brightness and contrast were adjusted accordingly for the green (GFP) and red (dsRed/ tdTomato) channels and composite serial image sequences were assembled as previously described. Images were obtained as large, tiled image stacks at roughly the same positions and then manually aligned over the experimental time course in Imaris (Bitplane/Perkin Elmer) by using data from all channels. Random regions of interest were selected for image analysis. Quantification of CAM coverage and capillary blood flow was performed blinded to minimize researcher bias.

### Statistical analysis

Data are expressed as mean ± s.d. or mean with paired lines (for paired Student's $t$-tests). Student's $t$-tests were used to analyse datasets with two groups. One-way or two-way ANOVA with either Tukey's or Fisher's post hoc multiple-comparison test was used to analyse datasets with three or more groups. $P < 0.05$ was considered to be significant. Statistical calculations were performed using Prism (GraphPad).

### Reporting summary

Further information on research design is available in the Nature Portfolio Reporting Summary linked to this article.

## Data availability

All data supporting the findings of this study are available from the corresponding authors on reasonable request. Source data are provided with this paper.

## Code availability

No custom code was used in this study.

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

**Acknowledgements** We thank the members of the Littman laboratory for discussions and reading the manuscript; and M. Cammer from the NYU Microscopy Core for discussions, training and technical support. The Microscopy Core is partially supported by NYU Cancer Center Support Grant NIH/NCI P30CA016087 at the Laura and Isaac Perlmutter Cancer Center, S10 RR023704-01A1 and NIH S10 ODO019974-01A1. This work was supported by a Jane Coffin Childs Fund fellowship (K.R.M.), Kirschstein-NRSA training grant T32AR64184 (K.R.M.), Kirschstein-NRSA individual postdoctoral fellowship F32AG071336 (K.R.M.), a Charles H. Revson Senior Fellowship in Biomedical Science (K.R.M.), the Helen and Martin Kimmel Center for Biology and Medicine (D.R.L.), NIH grant R01AI158687 (D.R.L.) and the Howard Hughes Medical Institute (D.R.L.).

**Author contributions** K.R.M. and D.R.L. designed the study and analysed the data. K.R.M., K.A.O. and A.D. performed mouse experiments. K.R.M. performed intravital multiphoton imaging. K.R.M and M.R.L. performed image analysis and quantification. C.N. and S.P.S. provided human biological samples and related quantitative analysis. K.R.M. and D.R.L. wrote the manuscript, with input from the other authors. D.R.L. supervised the research and provided funding.

**Competing interests** D.R.L. is a cofounder of Vedanta Biosciences and ImmunAI; is on the advisory boards of IMIDomics, Sonoma Biotherapeutics, NILO Therapeutics and Evommune; and is on the board of directors of Pfizer. The other authors declare no competing interests.

**Additional information**
**Correspondence and requests for materials** should be addressed to Kailin R. Mesa or Dan R. Littman.

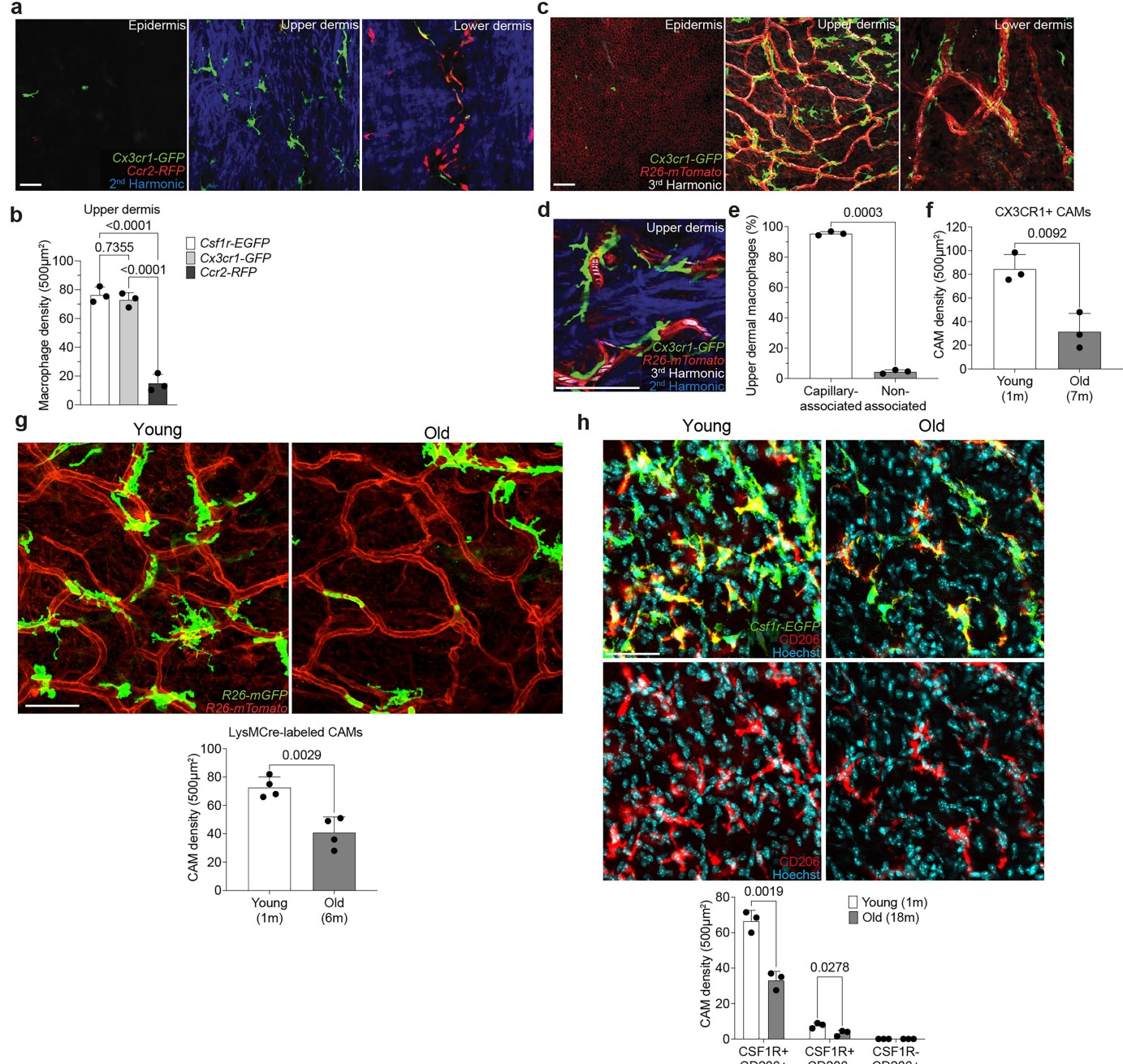

**Extended Data Fig. 1 | Upper dermal macrophages are capillary-associated and decline in density with age. a**, Representative images of cell expression of *Ccr2-RFP* and *Cx3cr1-GFP* in the epidermis, upper dermis, and lower dermis of 1 month old mice. **b**, Quantifications of cells expressing *Csf1r-EGFP*, *Cx3cr1-GFP*, *or Ccr2-RFP*, in the upper dermis (n = 3 mice in each group; two 500 μm² regions per mouse; Cell number (*Csf1r-EGFP* vs *Cx3cr1-GFP*; *Csf1r-EGFP* vs *Ccr2-RFP*) was compared by one-way ANOVA and Tukey multiple comparison tests; mean ± SD). **c**, Visualization of CX3CR1-expressing cells in all skin layers: epidermis, upper dermis, and lower dermis (*Cx3cr1-GFP;R26-mTmG*) was performed during homeostatic conditions. **d**, Representative image of labelled upper dermal macrophages in contact with capillary superficial plexus (red) in 1 month old mice. **e**, Quantifications reveal most upper dermal macrophages are capillary-associated macrophages (CAMs) (n = 3 mice in total; three 500 μm² regions per mouse; Upper dermal macrophage capillary association was compared by

paired Student's t test; mean ± SD). **f**, Quantifications of cells expressing *Cx3cr1-GFP* in the upper dermis (n = 3 mice in each group; two 500 μm² regions per mouse; CAM density (1 vs 7-month-old) was compared by unpaired Student's t test; mean ± SD). **g**, Representative optical sections of CAMs in young (1-month-old) and old (6-month-old) *LysM-Cre;R26-mTmG* mice. Quantifications of membrane-GFP+ cells in the upper dermis (n = 4 mice in each group; two 500 μm² regions per mouse; CAM density (1 vs 6-month-old) was compared by unpaired Student's t test; mean ± SD). **h**, Representative upper dermal optical sections from whole mount skin samples of young (1-month-old) and old (18-month-old) *Csf1r-EGFP* mice stained with AF647 anti-mouse CD206 (clone C068C2) antibody. Quantifications for CD206 and CSF1R co-expression in the upper dermis (n = 3 mice in each group; two 500 μm² regions per mouse; CAM density (1 vs 18-month-old) was compared by two-way RM ANOVA and Tukey multiple comparison tests; mean ± SD). Scale bar, 50 μm.

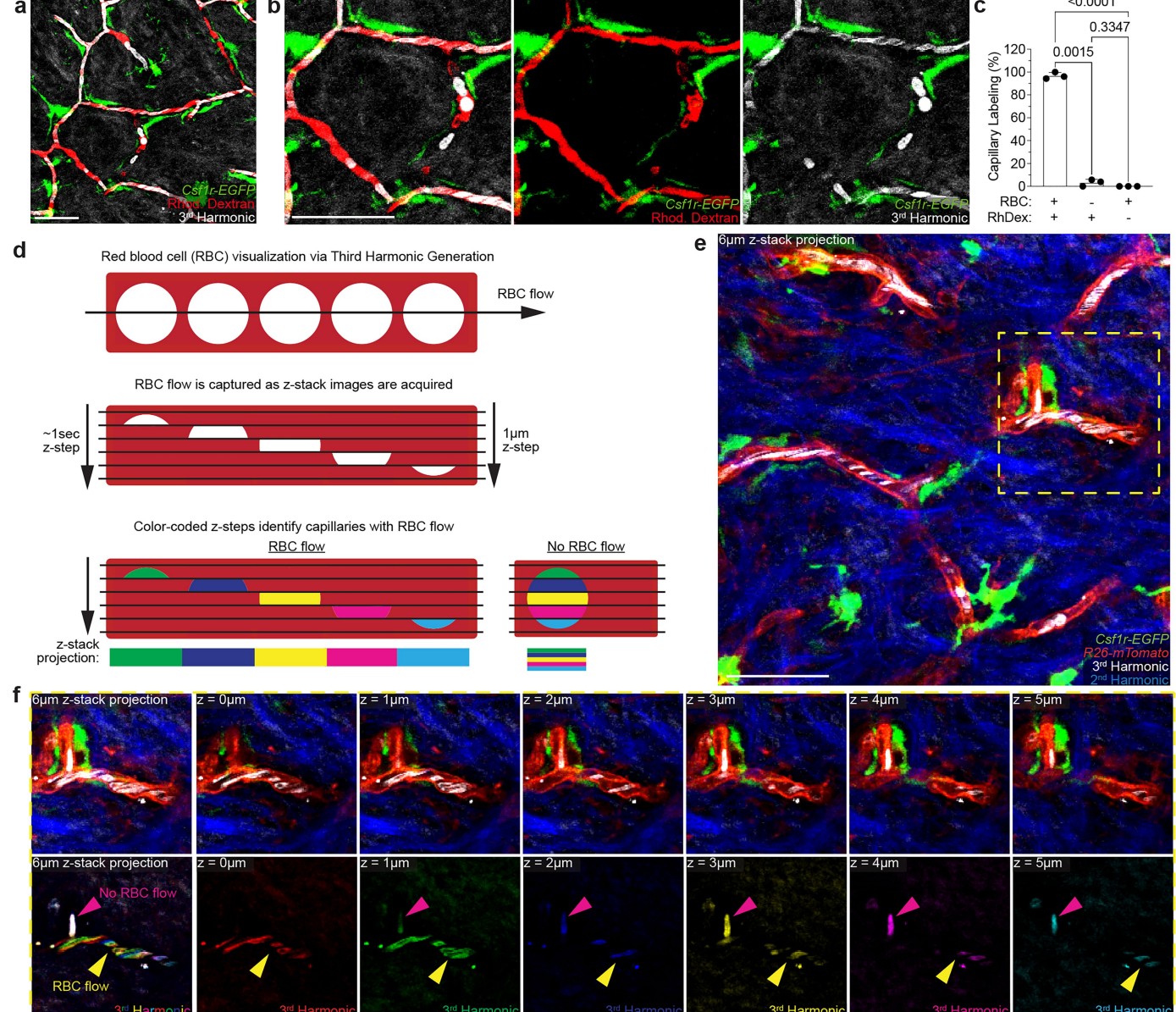

**Extended Data Fig. 2 | Label-free in vivo visualization of capillary blood flow via third harmonic effect generated by red blood cells. a-b,** Simultaneous visualization of blood flow through the superficial capillary plexus via third harmonic generation (white) from red blood cells (RBC) and intravenous rhodamine dextran (RhDex) (red). **c,** Quantifications of RBC and RhDex labelling of the upper dermal superficial capillary plexus (n = 3 mice in total; four 500 μm² regions per mouse; Capillary labelling was compared by one-way RM ANOVA and Tukey multiple comparison tests; mean ± SD). **d,** Scheme of red blood cell (RBC) flow through a segment of the superficial skin capillary network.

During three-dimensional image acquisition, flowing red blood cells are captured at different x,y positions for each z-section along the capillary segment. Pseudo colouring each z-step through a capillary segment distinguishes flowing RBCs as a multicolour patchwork or rainbow-effect and leaves obstructed/non-flowing RBCs as white (full overlap of all colours). **e,** Representative image of RBC visualization in the upper dermis in *Csf1r-EGFP; R26-mTmG* mice. **f,** Representative optical z-sections through upper dermal capillaries. Third harmonic signal is pseudo coloured differently for each z-step to visualize RBC movement along the capillary segments during image acquisition. Scale 50 μm.

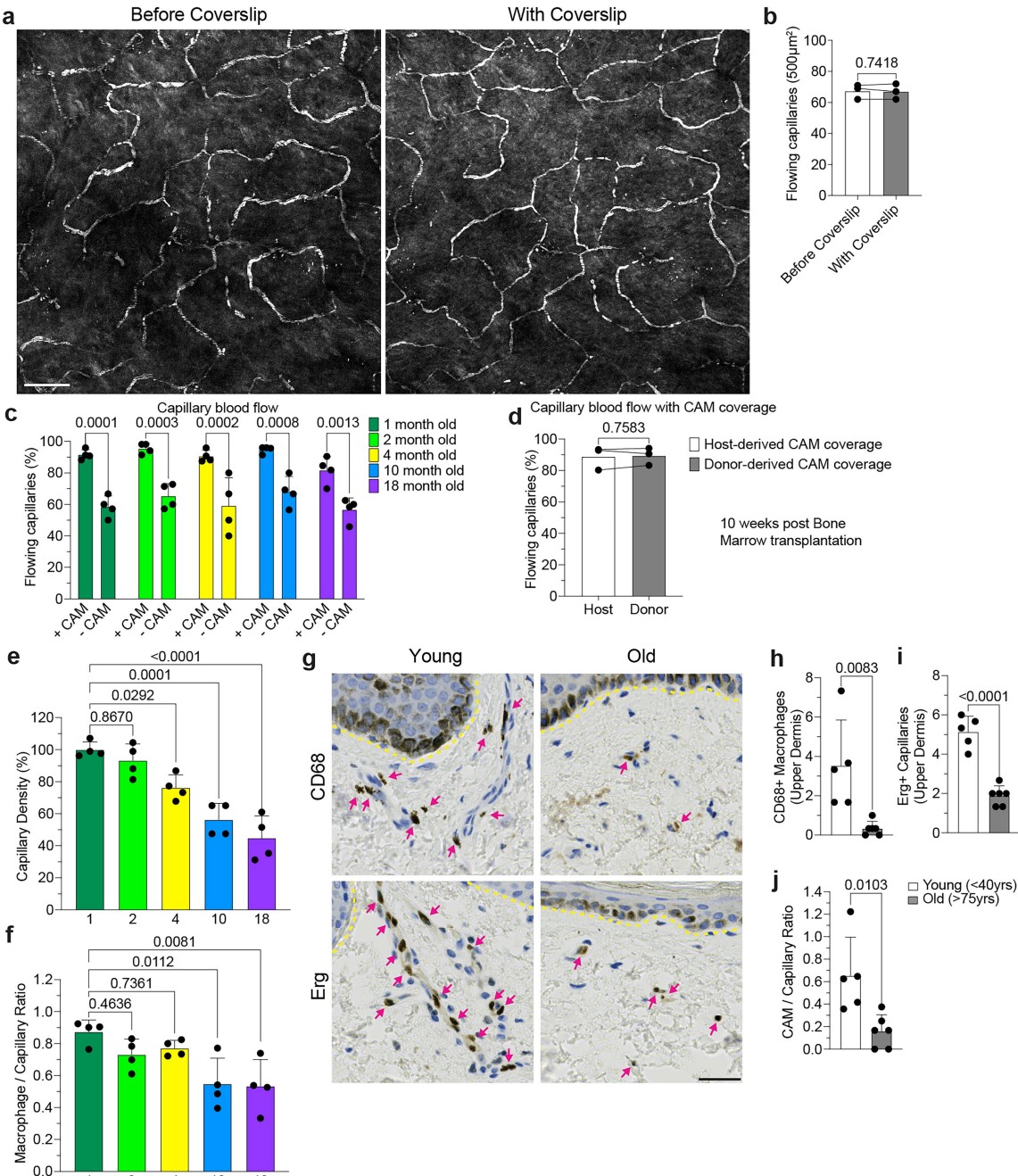

**Extended Data Fig. 3 | Capillary flow and associated macrophage loss in mice and humans. a**, Representative images of capillary blood flow in WT mice before and after placement of a coverslip on top of hind paw skin of 6 month old mice. **b**, Quantification of capillary blood flow (n = 3 mice in total; two 500 μm² regions per mouse; capillary blood flow (Before Coverslip vs With Coverslip) was compared in the same imaging areas by paired Student's t test; mean with paired lines). **c**, Quantification of capillaries with blood flow as measured by stalled RBCs as described in Extended Data Fig. 2 in 1, 2-, 4-, 10-, and 18-month-old mice (n = 878 CAM+ capillary segments, n = 215 CAM- capillary segments; n = 4 mice in each age group; capillary blood flow (CAM+ vs CAM-) was compared by two-way RM ANOVA and Tukey multiple comparison tests; mean ± SD). **d**, Quantification of capillaries with blood flow in in bone marrow chimeras with *Csf1r-GFP*;*CAG-dsRed* bone marrow transferred into lethally irradiated *Csf1r-GFP* mice (n = 207 host-derived CAM+ capillary segments, n = 79 donor-derived CAM+ capillary segments; n = 3 mice in total; CAM+ capillary blood

flow (Host-derived vs Donor-derived) was compared by paired Student's t test; mean with paired lines). **e,f** Quantification of capillary niche age-associated changes, (e) Capillary density and (f) CAM / Capillary segment ratio (n = 4 mice in each age group; two 500 μm² regions per mouse; comparison across age groups was by one-way ANOVA and Tukey multiple comparison tests; mean ± SD). **g**, Representative immunohistochemistry of CD68+ macrophage and Erg+ capillary density (magenta arrows) in the upper dermis (within 100 μm from epidermal boundary) of both young (< 40 y) and old (> 75 y) human samples. Epidermal boundary marked by yellow dashed line. **h-j**, Quantification of capillary niche age-associated changes: (h) CD68+ macrophage density, (i) Erg+ capillary density, (j) Upper dermal macrophage / Capillary endothelium ratio (n = 5 patient samples in young group; n = 6 patient samples in old group; three imaging regions per sample; Comparison by unpaired Student's t test; mean ± SD). Scale bar, 50 μm.

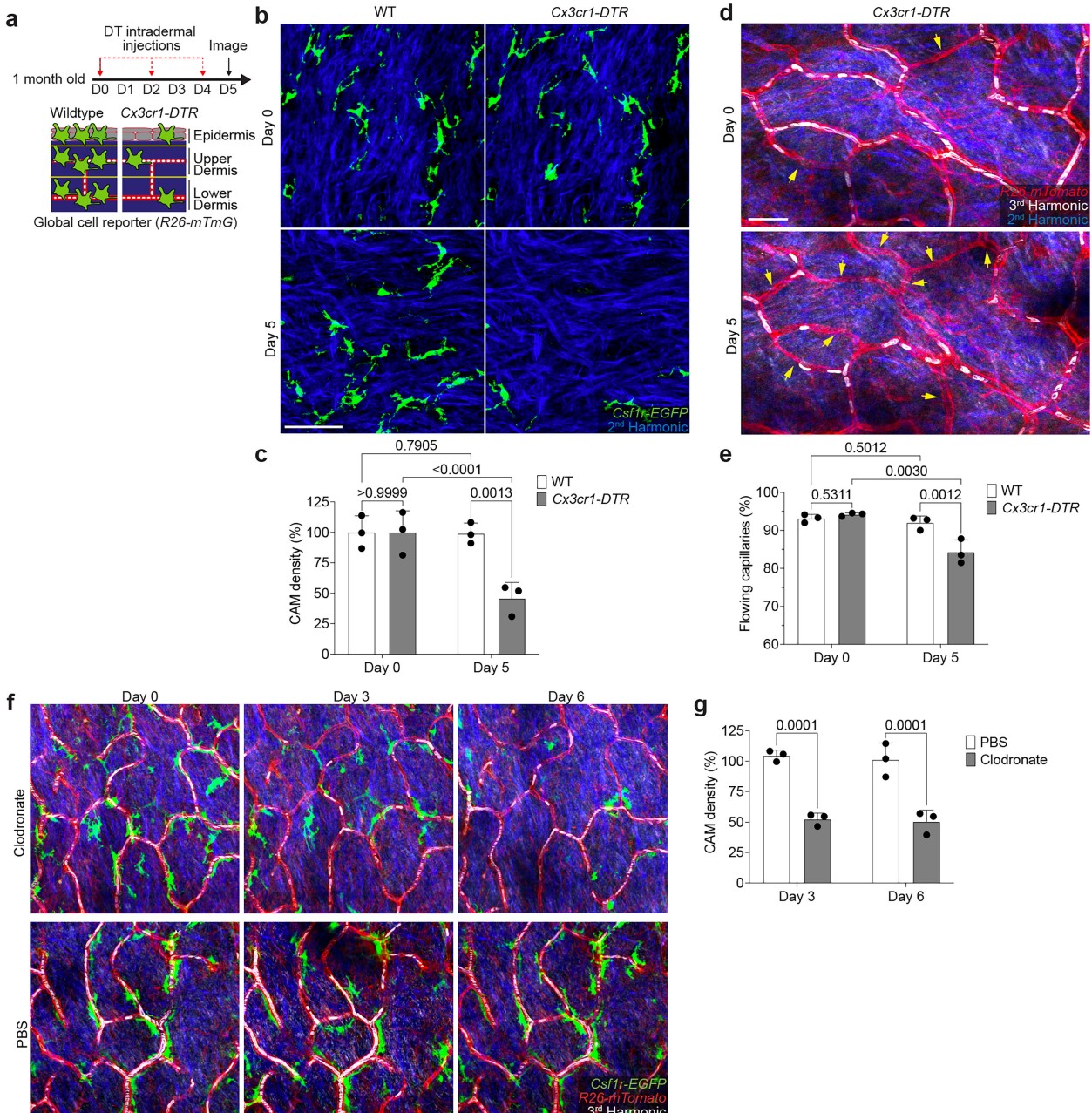

**Extended Data Fig. 4 | Impaired skin capillary blood flow following acute macrophage depletion. a**, Scheme of capillary blood flow tracking following intraperitoneal injection every other day with diphtheria toxin (DT) (25 ng/g body weight in PBS) in both *Cx3cr1-DTR; R26-mTmG* and WT control (*R26-mTmG*) mice. **b**, Representative revisits of the same upper dermal capillary niches to visualize CAMs (*Csf1r-EGFP*) and dermal collagen (Second Harmonic). **c**, Quantifications reveal a significant reduction in CAM density following DT-treatment (n = 3 mice in each group; two 500 μm² regions per mouse; CAM density (WT vs *Cx3cr1-DTR*) was compared for Day 0 and Day 5 time points by two-way RM ANOVA and Fisher multiple comparison tests; mean ± SD). **d**, Representative revisits of the same capillary network to visualize capillaries (*R26-mTmG*) and RBC flow (Third Harmonic). **e**, Quantifications reveal a significant reduction in the percentage of capillaries with blood flow following DT-induced cell depletion (n = 3 mice in each group; two 500 μm² regions per mouse; obstructed capillary flow (WT vs *Cx3cr1-DTR*) was compared for Day 0 and Day 5 time points by two-way RM ANOVA and Fisher multiple comparison tests; mean ± SD). **f**, Representative images demonstrate macrophage depletion following intradermal injections of clodronate-liposomes every 3 days. Repeated intravital imaging of the vascular niche was performed to visualize macrophages (*Csf1r-GFP*), capillaries (*R26-mTmG*) and RBC flow (Third Harmonic). **g**, Quantification of CAM density following macrophage depletion (n = 3 mice in each group; two 500 μm² regions per mouse; CAM density (clodronate vs PBS) was compared by two-way RM ANOVA and Fisher multiple comparison tests; mean ± SD). Scale bar, 50 μm.

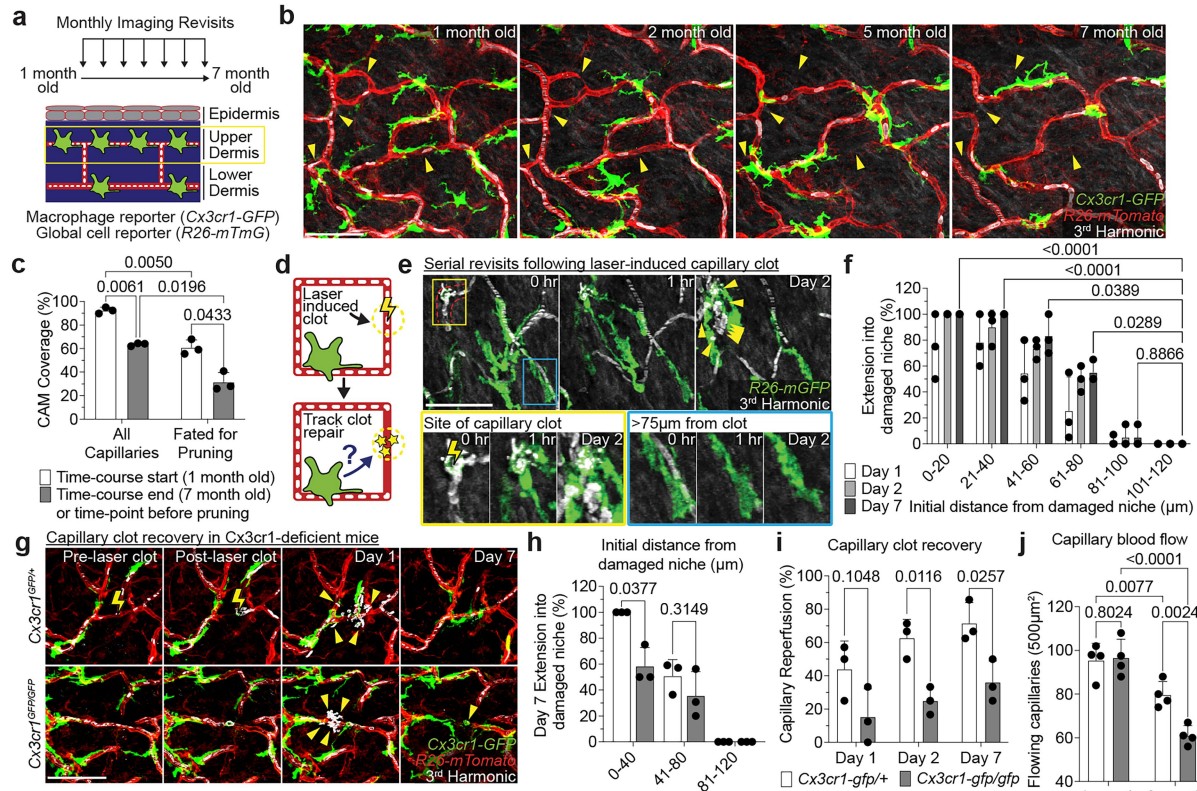

**Extended Data Fig. 5 | Local recruitment of CAMs is required to restore capillary blood flow. a**, Scheme of long-term serial imaging of capillary niche in *Cx3cr1-GFP;R26-mTmG* mice. **b**, Sequential revisits reveal progressive pruning of capillary niche over a 6-month period. Yellow arrowheads indicate capillaries that will undergo pruning. **c**, Quantification of capillary-associated macrophage coverage at 1 and 7 months of age (n = 3 mice; three 500 μm² regions per mouse; Frequency of macrophage association with capillary segments was compared between capillaries fated for pruning and all capillaries at 1- and 7-month-old time-points by two-way RM ANOVA and Fisher multiple comparison tests; mean ± SD). **d**, Scheme of laser-induced capillary clot experiment. **e**, Sequential revisits of damaged capillary segment (red dashed lines) after laser-induced clot formation in *Cx3cr1-CreERT2;R26-mTmG* mice. Yellow lightning bolt indicates site of laser-induced capillary clot. Yellow arrowheads indicate extra-luminal vascular debris. **f**, Quantification of capillary-associated macrophage extension toward damaged niche at Day 1, 2 and 7 after laser-induced clotting (n = 50 CAMs, in 3 mice; CAM extension toward capillary damage was compared by two-way RM ANOVA and Tukey multiple comparison tests; mean ± SD). **g**, Sequential revisits of damaged capillary niche after laser-induced clot formation (yellow lightning bolt) in *Cx3cr1^{gfp/+}* and *Cx3cr1^{gfp/gfp}* mice. CAMs (green), capillaries (red), RBC (white). Yellow arrowheads indicate extra-luminal vascular debris. **h**, Quantification of capillary-associated macrophage extension toward damaged niche at Day 7 after laser-induced clotting (n = 99 CAMs in *Cx3cr1^{gfp/+}* group; n = 65 CAMs in *Cx3cr1^{gfp/gfp}* group; n = 3 mice in each group; CAM extension (*Cx3cr1^{gfp/+}* vs *Cx3cr1^{gfp/gfp}*) was compared by two-way RM ANOVA and Tukey multiple comparison tests; mean ± SD). **i**, Quantification of capillary reperfusion at Day 1, 2 and 7 after laser-induced clotting (n = 20 clots in *Cx3cr1^{gfp/+}* group; n = 21 clots in *Cx3cr1^{gfp/gfp}* group; n = 3 mice in each group; capillary reperfusion (*Cx3cr1^{gfp/+}* vs *Cx3cr1^{gfp/gfp}*) was compared by two-way RM ANOVA and Tukey multiple comparison tests; mean ± SD). **j**, Quantification of capillary blood flow (n = 4 mice in each group; two 500 μm² regions per mouse; capillary blood flow (*Cx3cr1^{gfp/+}* vs *Cx3cr1^{gfp/gfp}*) was compared at 1 and 6 months of age by two-way ANOVA and Fisher multiple comparison tests; mean ± SD). Scale bar, 50 μm.

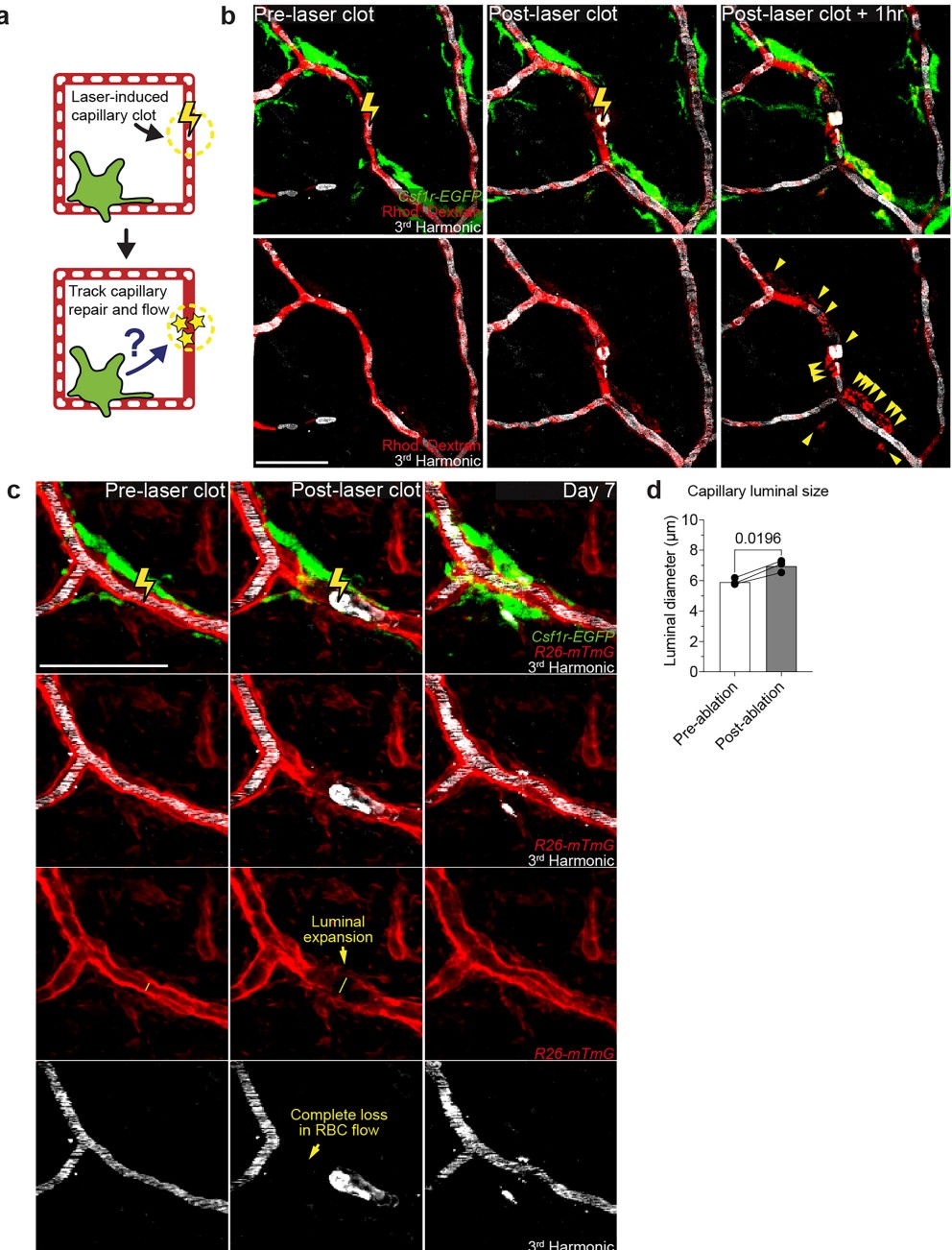

**Extended Data Fig. 6 | Laser-induced model of acute capillary clot formation and repair. a**, Scheme of laser-induced capillary clot experiment. **b**, Sequential revisits of damaged capillary niche after laser-induced clot formation in *Csf1r-EGFP* mice. Simultaneous visualization of blood flow before and after clot formation via third harmonic generation (white) from red blood cells (RBC) and intravenous rhodamine dextran (RhDex) (red). Yellow lightning bolt indicates site of laser-induced capillary clot. Yellow arrowheads indicate extra-luminal vascular debris. **c**, Sequential revisits of damaged capillary niche after laser-induced clot formation in *Csf1r-EGFP;R26-mTmG* mice. Capillary clot formation (yellow lightning bolt) was performed at 940 nm for 1 s in 4-month-old mice. **d**, Quantification of capillary lumen diameter before and immediately after laser-induced clotting (n = 24 capillary clots; 3 mice in total; capillary luminal diameter (pre-ablation vs post-ablation) was compared by paired Student's t test; mean with paired lines). Scale bar, 50 μm.

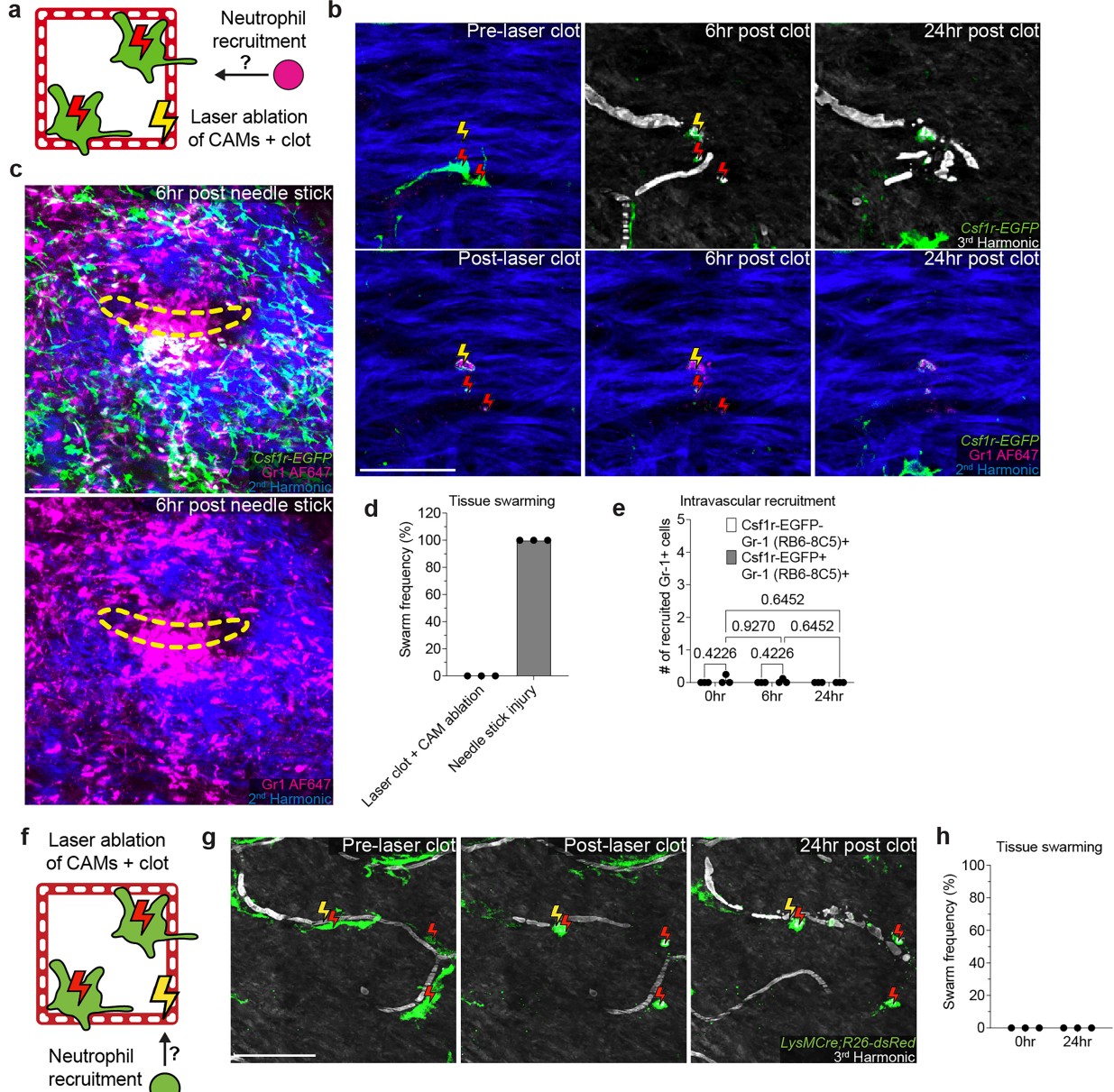

**Extended Data Fig. 7 | Laser-induced clots and CAM ablations do not recruit neutrophil swarms. a**, Scheme of tracking neutrophil recruitment and swarming after laser-induced capillary clot experiment. **b-c**, (b) Sequential revisits of damaged capillary niche after laser-induced CAM ablation and clot formation or (c) large non-sterile tissue damage (28-gauge needle stick) in *Csf1r-EGFP* mice with intravenous neutrophil antibody labelling (Gr-1, clone RB6-8C5). Macrophage laser ablation (red lightning bolt) and capillary clot formation (yellow lightning bolt) were both performed at 940 nm for 1 s. Needle injury (yellow dashed line) was performed through epidermis and dermis. **d**, Quantification of neutrophil swarming at 6 h tissue damage (n = 16 capillary clots in CAM ablated regions, n = 9 needle injuries; 3 mice in each group; mean ± SD). **e**, Quantification of intravascular recruitment of neutrophil and monocyte populations at 0, 6, and 24 h post laser-induced capillary clotting in CAM ablated regions (n = 16 capillary clots in CAM ablated regions, 3 mice in total; Gr-1+ cell recruitment was compared by two-way RM ANOVA and Tukey multiple comparison tests; mean ± SD). **f**, Scheme of tracking neutrophil recruitment and swarming after laser-induced capillary clot experiment. **g**, Sequential revisits of damaged capillary niche after laser-induced CAM ablation and clot formation in *LysMCre;R26-dsRed* mice. Macrophage laser ablation (red lightning bolt) and capillary clot formation (yellow lightning bolt) were both performed at 940 nm for 1 s. **h**, Quantification of neutrophil swarming at 6 h tissue damage (n = 32 capillary clots in CAM ablated regions; 3 mice in total; mean ± SD). Scale bar, 50 μm.

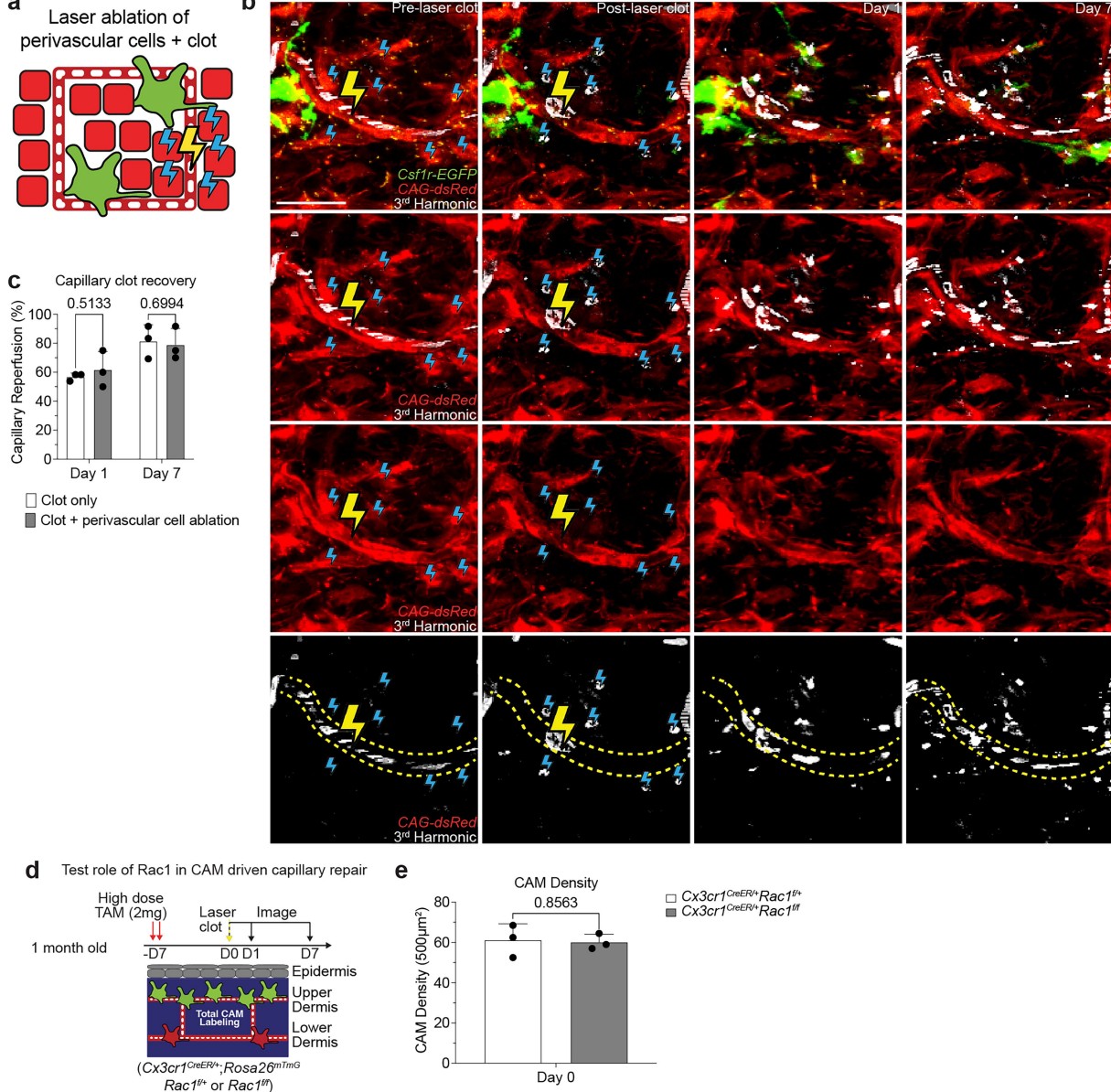

**a** Laser ablation of perivascular cells + clot

**b** Pre-laser clot | Post-laser clot | Day 1 | Day 7

*Csf1r-EGFP* *CAG-dsRed* 3rd Harmonic

*CAG-dsRed* 3rd Harmonic

*CAG-dsRed* 3rd Harmonic

*CAG-dsRed* 3rd Harmonic

**c** Capillary clot recovery

0.5133    0.6994

Capillary Reperfusion (%)

Day 1    Day 7

☐ Clot only
▨ Clot + perivascular cell ablation

**d** Test role of Rac1 in CAM driven capillary repair

High dose TAM (2mg)    Laser clot    Image
1 month old    -D7    D0 D1    D7

Epidermis
Upper Dermis
Total CAM Labeling
Lower Dermis

(*Cx3cr1^{CreER/+}*;*Rosa26^{mTmG}* *Rac1^{f/+}* or *Rac1^{f/f}*)

**e** CAM Density

☐ *Cx3cr1^{CreER/+}Rac1^{f/+}*
▨ *Cx3cr1^{CreER/+}Rac1^{f/f}*

0.8563

CAM Density (500µm²)

Day 0

**Extended Data Fig. 8 | Capillary clot repair is unimpaired folllowing ablation of non-macrophage perivascular cells and CAM density remains stable in phagocytosis-deficient mice. a**, Scheme of laser-induced capillary clot experiment. **b**, Sequential revisits of damaged capillary niche after laser-induced perivascular dermal cell ablation and clot formation in *Csf1r-EGFP*;*CAG-dsRed* mice. Perivascular dermal cell laser ablation (blue lightning bolt) and capillary clot formation (yellow lightning bolt) were both performed at 940 nm for 1 s. **c**, Quantification of capillary reperfusion at Day 1 and 7 after laser-induced clotting and perivascular cell ablation (n = 34 capillary clots in CAM ablated regions, n = 37 capillary clots in control regions; 3 mice in total; capillary reperfusion (perivascular dermal cell ablation vs control) was compared by two-way RM ANOVA and Fisher multiple comparison tests; mean ± SD). **d**, Scheme of laser-induced capillary clot in *Cx3cr1^{CreER}*;*Rac1^{fl/fl}* and *Cx3cr1^{CreER}*; *Rac1^{fl/+}* mice. **e**, Quantification of CAM density at Day 7 after laser-induced clotting (n = 3 mice in each group; two 500 µm² regions per mouse; CAM density (*Rac1^{fl/+}* vs *Rac1^{fl/fl}*) was compared at day 0 (same day of clot induction) by unpaired Student's t test; mean ± SD). Scale bar, 50 µm.

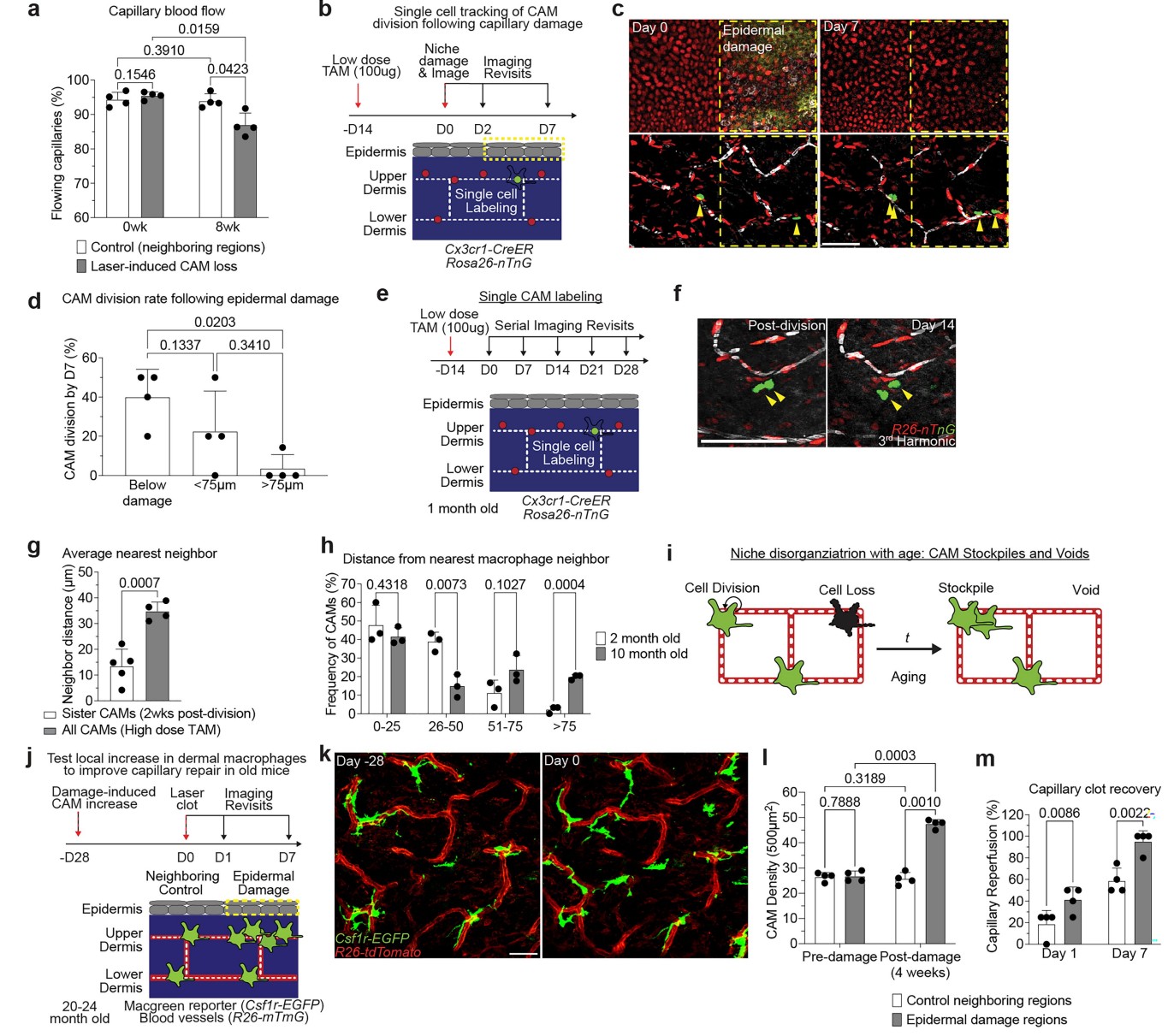

**Extended Data Fig. 9 | CAM loss and disorganization is partially restored by local epidermal damage to improve capillary repair in old mice.**
**a**, Quantification of capillary blood flow following laser-induced CAM loss (n = 293 capillary segments in CAM ablated regions (grey bar graph), n = 250 capillary segments in control neighbouring regions (white bar graph); n = 4 mice in total; capillary blood flow was compared by two-way RM ANOVA and Fisher multiple comparison tests; mean ± SD). **b**, Scheme of tracking of CAM proliferation following laser-induced damage to nearby capillary or epidermal niches. **c**, Representative revisits of single macrophage lineage tracing in *Cx3cr1-CreERT2*; *R26-nTnG* mice following epidermal damage. **d**, Quantification of CAM proliferation based on proximity to epidermal damage (n = 25 CAMs tracked in capillary damage regions, n = 56 CAMs tracked in epidermal damage regions; 4 mice in each group; CAM proliferation (based on damage proximity) was compared at day 7 by one-way RM ANOVA and Tukey multiple comparison tests; mean ± SD). **e**, Scheme of long-term tracking of CAM migration following cell division. **f**, Representative revisits of single macrophage lineage tracing in *Cx3cr1-CreERT2*; *R26-nTnG* mice. Weekly revisits were performed during homeostatic conditions for 5 weeks following a single low-dose intraperitoneal injection of tamoxifen (50 μg). **g**, Quantification of neighbouring CAM distance of recently divided sister CAMs was compared to total CAM neighbouring distance from *Cx3cr1-CreERT2*; *R26-nTnG* mice given a single high-dose intraperitoneal injection of tamoxifen (4 mg) (n = 26 sister CAM pairs, from

5 mice in low-dose group; n = 188 CAMs, from 4 mice in high-dose group; Average nearest neighbour distance (2 vs 10 month old) was compared at 0–25, 26–50, 51–75, and >75 μm intervals by unpaired Student's t test mean ± SD). **h**, Quantification of distance between nearest CAM neighbours in 2- and 10-month-old *Csf1r-EGFP* mice (n = 90 CAMs in 2-month-old mice, n = 101 CAMs in 10-month-old mice; 3 mice in each age group; Frequency of CAM distribution (2 vs 10 month old) was compared at 0–25, 26–50, 51–75, and >75 μm intervals by two-way RM ANOVA and Tukey multiple comparison tests; mean ± SD). **i**, Working model of macrophage renewal and organization in the skin ageing capillary niche. **j**, Scheme of local damage-induced expansion of CAMs in the aged capillary niche. **k**, Representative images of CAM density in *Csf1r-EGFP; R26-mTmG* mice 28 days following overlaying epidermal laser damage. **l**, Quantification of CAM density following overlaying epidermal damage (n = 4 mice in total; two 500 μm² regions for both epidermal damage and control conditions per mouse; CAM density was compared by two-way RM ANOVA and Fisher multiple comparison tests; mean ± SD). **m**, Quantification of capillary reperfusion at day 1 and 7 after laser-induced clotting (n = 17 capillary clots in epidermal damage regions, n = 17 capillary clots in neighbouring control regions; 4 mice in total; capillary reperfusion (with epidermal damage vs control) was compared at day 1 and day 7 by two-way RM ANOVA and Fisher multiple comparison tests; mean ± SD). following laser-induced damage to overlaying epidermal niche. Scale bar, 50 μm.

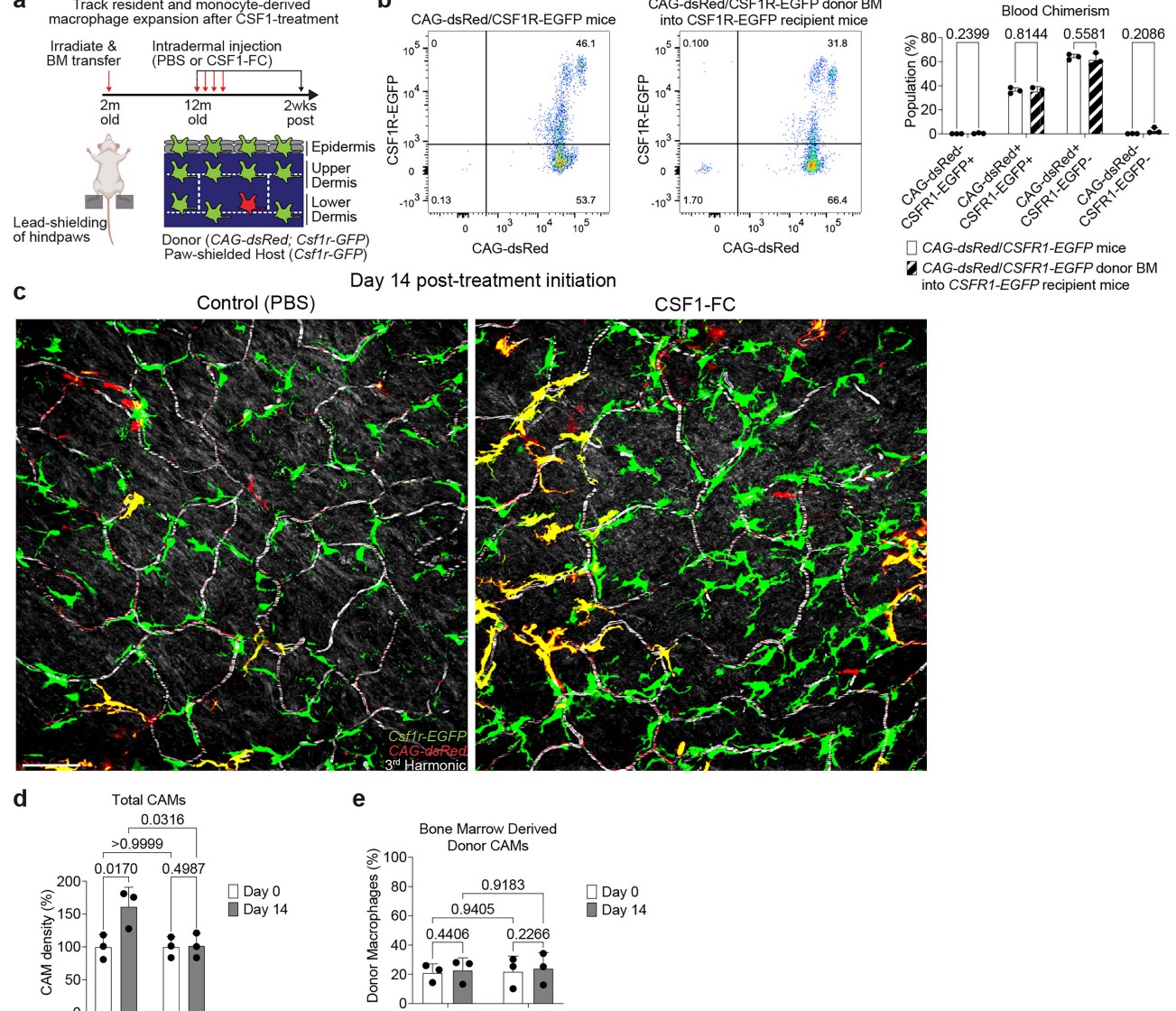

**Extended Data Fig. 10 | Intradermal CSF1 treatment locally expands CAM populations in old mice. a**, Scheme of serial imaging of skin resident macrophage repopulation in bone marrow chimeras with *Csf1r-GFP;CAG-dsRed* bone marrow transferred into lethally irradiated *Csf1r-GFP* mice following CSF1-treatment. Note hind paws of these mice were lead-shielded to prevent irradiation-induced loss of resident macrophage populations from our imaging area. Created in BioRender. Mesa, K. (2025) https://BioRender.com/3y6g97c. **b**, Quantification of blood chimerism 10 months after *Csf1r-GFP;CAG-dsRed* bone marrow transfer into lethally irradiated *Csf1r-GFP* mice (n = 3 mice in each group; Percentage of cells from blood (*Csf1r-GFP;CAG-dsRed* mice vs bone marrow transferred mice) was compared for dsRed-/GFP + , dsRed + /GFP + , dsRed + /GFP-, dsRed-/GFP- populations by two-way RM ANOVA and Tukey multiple comparison tests; mean ± SD). **c**, Representative images of CAM

density 14 days following daily intradermal injections (4 days) of CSF1-Fc (porcine CSF1 and IgG1a Fc region fusion protein) and PBS in contralateral hind paws of 12-month-old mice. **d**, Quantification of capillary-associated macrophage density following CSF1-Fc or PBS treatment (n = 3 mice in total; two 500 μm² regions of each treatment condition per mouse; Percentage of CAM density change (Day 0 vs 14) was compared for CSF1-Fc and PBS regions by two-way RM ANOVA and Fisher multiple comparison tests; mean ± SD). **e**, Quantification of donor bone marrow-derived (GFP + /dsRed + ) CAM density following CSF1-Fc or PBS treatment (n = 3 mice in total; two 500 μm² regions of each treatment condition per mouse; Percentage of donor bone marrow-derived CAM density change (Day 0 vs 14) was compared for CSF1-Fc and PBS regions by two-way RM ANOVA and Fisher multiple comparison tests; mean ± SD). Scale bar, 50 μm.

Dan R Littman

# Reporting Summary

## Statistics

For all statistical analyses, confirm that the following items are present in the figure legend, table legend, main text, or Methods section.

| n/a | Confirmed | |
|---|---|---|
| ☐ | ☒ | The exact sample size (*n*) for each experimental group/condition, given as a discrete number and unit of measurement |
| ☐ | ☒ | A statement on whether measurements were taken from distinct samples or whether the same sample was measured repeatedly |
| ☐ | ☒ | The statistical test(s) used AND whether they are one- or two-sided <br> *Only common tests should be described solely by name; describe more complex techniques in the Methods section.* |
| ☐ | ☒ | A description of all covariates tested |
| ☐ | ☒ | A description of any assumptions or corrections, such as tests of normality and adjustment for multiple comparisons |
| ☐ | ☒ | A full description of the statistical parameters including central tendency (e.g. means) or other basic estimates (e.g. regression coefficient) AND variation (e.g. standard deviation) or associated estimates of uncertainty (e.g. confidence intervals) |
| ☐ | ☒ | For null hypothesis testing, the test statistic (e.g. *F*, *t*, *r*) with confidence intervals, effect sizes, degrees of freedom and *P* value noted <br> *Give P values as exact values whenever suitable.* |
| ☒ | ☐ | For Bayesian analysis, information on the choice of priors and Markov chain Monte Carlo settings |
| ☒ | ☐ | For hierarchical and complex designs, identification of the appropriate level for tests and full reporting of outcomes |
| ☒ | ☐ | Estimates of effect sizes (e.g. Cohen's *d*, Pearson's *r*), indicating how they were calculated |

*Our web collection on statistics for biologists contains articles on many of the points above.*

## Software and code

Policy information about availability of computer code

| Data collection | Fluoview FV32S-SW (Olympus v2.3.l.163), Aperio ImageScope (Leica, v12.4.6), FACSDiva (BD Biosciences) |
|---|---|
| Data analysis | Fiji (NIH v2.14.0), Prism (GraphPad, v10.0.2), FlowJo (BD Biosciences, 10.10.1) |

For manuscripts utilizing custom algorithms or software that are central to the research but not yet described in published literature, software must be made available to editors and reviewers. We strongly encourage code deposition in a community repository (e.g. GitHub). See the Nature Portfolio guidelines for submitting code & software for further information.

## Data

Policy information about availability of data

All manuscripts must include a data availability statement. This statement should provide the following information, where applicable:
- Accession codes, unique identifiers, or web links for publicly available datasets
- A description of any restrictions on data availability
- For clinical datasets or third party data, please ensure that the statement adheres to our policy

Source data will be provided with this paper. All other data supporting the findings of this study are available from the corresponding authors on reasonable request.

## Research involving human participants, their data, or biological material

Policy information about studies with human participants or human data. See also policy information about sex, gender (identity/presentation), and sexual orientation and race, ethnicity and racism.

| | |
|---|---|
| Reporting on sex and gender | Clinical information and laboratory data were obtained from the electronic medical record. Sex and gender information was not used. |
| Reporting on race, ethnicity, or other socially relevant groupings | N/A |
| Population characteristics | Patients in young group were below 40 years of age (19 - 37 years old). Patients in old group were above 40 years of age (79 - 97 years old). Patients with skin or vascular pathologies were excluded. |
| Recruitment | Written informed consent was obtained for postmortem examination from next of kin for all patients. |
| Ethics oversight | Study protocol was approved by Weill Cornell Department of Pathology |

Note that full information on the approval of the study protocol must also be provided in the manuscript.

# Field-specific reporting

Please select the one below that is the best fit for your research. If you are not sure, read the appropriate sections before making your selection.

☒ Life sciences    ☐ Behavioural & social sciences    ☐ Ecological, evolutionary & environmental sciences

For a reference copy of the document with all sections, see nature.com/documents/nr-reporting-summary-flat.pdf

# Life sciences study design

All studies must disclose on these points even when the disclosure is negative.

| | |
|---|---|
| Sample size | Three or more mice per group were used in each experiment. The precise number of animals used are given in the figure legend. The sample size was determined from our previous experience and from what is accepted in the field. |
| Data exclusions | No samples were excluded from the analysis. |
| Replication | Experiments were replicated at least three times unless otherwise specified. All attempts were successful. |
| Randomization | Allocation into sample groups was random. In addition, experimental and control mice were used from the same litter whenever possible. Both males and females were used. |
| Blinding | For human skin samples, the age group of samples were blinded to the researcher. For mouse imaging experiments, random ROI were selected for image analysis. Quantification of CAM coverage and capillary blood flow was performed blinded to minimize researcher bias. |

# Reporting for specific materials, systems and methods

We require information from authors about some types of materials, experimental systems and methods used in many studies. Here, indicate whether each material, system or method listed is relevant to your study. If you are not sure if a list item applies to your research, read the appropriate section before selecting a response.

### Materials & experimental systems

| n/a | Involved in the study |
|---|---|
| ☐ | ☒ Antibodies |
| ☒ | ☐ Eukaryotic cell lines |
| ☒ | ☐ Palaeontology and archaeology |
| ☐ | ☒ Animals and other organisms |
| ☒ | ☐ Clinical data |
| ☒ | ☐ Dual use research of concern |
| ☒ | ☐ Plants |

### Methods

| n/a | Involved in the study |
|---|---|
| ☒ | ☐ ChIP-seq |
| ☐ | ☒ Flow cytometry |
| ☒ | ☐ MRI-based neuroimaging |

# Antibodies

| | |
|---|---|
| Antibodies used | CD68 HRP (Clone 514H12, Leica PA0273), ERG HRP (Clone EPR3864, Abcam ab92513), CD206 AF647 (Clone C068C2, BioLegend 141712), Gr1 AF647 (Clone RB6-8C5, BioLegend 108418). |
| Validation | Antibodies were used according to recommendations of the manufacturer. |

# Animals and other research organisms

Policy information about studies involving animals; ARRIVE guidelines recommended for reporting animal research, and Sex and Gender in Research

| | |
|---|---|
| Laboratory animals | Mice were bred and maintained in the Alexandria Center for the Life Sciences animal facility of the New York University School of Medicine, in specific pathogen-free conditions. Albino B6 (B6(Cg)-Tyrc-2J/J, Jax 000058), Csf1rEGFP (B6.Cg-Tg(Csf1r-EGFP)1Hume/J, Jax 018549), Ccr2RFP (B6.129(Cg)-Ccr2tm2.1Ifc/J, Jax 017586), R26mTmG (B6.129(Cg)-Gt(ROSA)26Sortm4(ACTB-tdTomato,-EGFP)Luo/J, Jax 007676), R26nTnG (B6N.129S6-Gt(ROSA)26Sortm1(CAG-tdTomato*,-EGFP*)Ees/J, Jax 023537), LysMCre (B6.129P2-Lyz2tm1(cre)Ifo/J, Jax 004781), Rac1f/f (Rac1tm1Djk/J, Jax 005550), and CAG-dsRed (B6.Cg-Tg(CAG-DsRed*MST)1Nagy/J, JAX 006051) mice were purchased from Jackson Laboratories. R26dsRed mice were described previously (Luche et al., 2007) and were obtained from the laboratory of Dr. Gordon Fishell. Cx3cr1CreER, Cx3cr1GFP, and Cx3cr1DTR were generated in our laboratory and have been described (Parkhurst, et al., 2013; Jung, et al., 2000; Diehl, et al., 2013). All experimental mice for this study were albino (homozygous for Tyrc-2J) as is required for intravital imaging in the skin. Mice 1-24 months old were used for experiments. |
| Wild animals | This study did not involve wild animals. |
| Reporting on sex | Mice from experimental and control groups were randomly selected from either sex for live imaging experiments. |
| Field-collected samples | This study did not involve samples collected from the field. |
| Ethics oversight | All animal procedures were performed in accordance with protocols approved by the Institutional Animal Care and Usage Committee of New York University School of Medicine. |

Note that full information on the approval of the study protocol must also be provided in the manuscript.

# Flow Cytometry

## Plots

Confirm that:

☒ The axis labels state the marker and fluorochrome used (e.g. CD4-FITC).

☒ The axis scales are clearly visible. Include numbers along axes only for bottom left plot of group (a 'group' is an analysis of identical markers).

☒ All plots are contour plots with outliers or pseudocolor plots.

☒ A numerical value for number of cells or percentage (with statistics) is provided.

## Methodology

| | |
|---|---|
| Sample preparation | Peripheral blood samples were collected from the submandibular (facial) vein in tubes containing EDTA (BD Biosciences; Dipotassium EDTA Microtainer; Cat# 365972). Red blood cells were lysed with ACK lysing buffer and the remaining cells were subjected flow cytometric analysis. |
| Instrument | LSR II Flow Cytometer (BD Biosciences) |
| Software | Samples were ran on the LSR III with FACSDiva and FlowJo 10.10.1 software (BD Biosciences) |
| Cell population abundance | Cell population abundance was displayed as percentage of total cells from blood sample. |
| Gating strategy | Gates were defined from blood samples of WT and single fluorescent reporter (CAG-dsRed and CSF1R-EGFP) mouse strains. |

☒ Tick this box to confirm that a figure exemplifying the gating strategy is provided in the Supplementary Information.

