## [Peer Review File · Nature]

Niche-specific dermal macrophage loss promotes skin capillary aging

Corresponding Author: Dr Kailin Mesa

Version 0:

Reviewer comments:

Referee #1

(Remarks to the Author)

In this study titled, "Niche-specific macrophage loss promotes skin capillary aging," the authors recognize that the effects of aging on tissue resident macrophage function and the cellular mechanisms behind such effects are poorly understood. As such, Mesa et al. seek to further elucidate the organization, behavior, and function of tissue resident macrophages within the skin capillary plexus niche (capillary-associated macrophages, or CAMs, hereafter) within an aging context.

Using intravital two-photon microscopy to track CAMs in vivo within the skin capillary nexus, the researchers report a close functional relationship between CAMs and blood capillaries, initially finding that capillaries that lacked associated macrophages displayed obstructed blood flow and that CAM populations appear to diminish at a faster rate than capillaries themselves. Furthermore, using laser-induced blood clotting models, the researchers find that CAMs support capillaries via phagocytosis of red blood cell debris at capillary repair sites. The authors also report that this support may be limited with age as they find that CAMs fail to sustain or replenish nearby macrophage populations, resulting in a cumulative loss of macrophage populations with age. Finally, the authors report that sufficient capillary repair in aged mice can be achieved with either extrinsic cues such as broad tissue damage or addition of growth factors such as CSF1, both of which resulted in greater capillary repair and tissue reperfusion.

While this manuscript is of potential interest, there are major concerns regarding the conclusions that are derived from their experimental data. In particular, it is unclear a) if the discontinuous 3rd harmonics flow in fact represents clots b) what the cell type that mediate clearance is, and c) mechanisms involved in CAM-mediated clot clearance is not explored in-depth. Authors might consider the below comments to strengthen their manuscript.

Concerns and Limitations

Major:

1. The association between CAMs and blood vasculature, their subsequent interactions, and the potential regulation of vascular physiology, have been previously documented (PMID: 34489419, 23451163, 36737792), suggesting that the concept of this study is not completely novel. The finding that CAMs help sustain the blood flow in capillary is of potential interest, but there are questions regarding the validity of this concept as described below.
2. Authors to identify obstructed capillary flow by utilizing 3rd harmonics. However, due to the discontinuous nature of the 3rd harmonics signal even without any manipulation, it is difficult to ascertain that discontinuous signal in fact represents disrupted capillary flow or clotting. It is rather difficult to accept that natural clots occur so frequently, as displayed in Figure 1d-f. In general, the difference in 3rd harmonics signals between capillaries with or without clots seem very subtle. Authors are encouraged to validate the existence of clots by other objective and quantifiable measures, both in experimental and WT controls.
3. It is not clearly defined which cell type(s) mediate the clearance of clots. Neither CX3CR1 nor CSF1R are exclusive macrophage markers, given their frequent expressions in various cell types, including monocytes and dendritic cells (PMID: 32582197, 29503738, 16034075), emphasizing the importance for the authors to incorporate additional markers in combination for distinguishing these subpopulations. In Figure 1b, understanding of the distribution of capillary-associated macrophages could be significantly improved via an orthogonal view of z-stack images in the epidermis, upper dermis, and

lower dermis. The use of other well-defined macrophage markers (e.g. CD64) is strongly encouraged. It also appears that DT-induced or homeostatic loss of CAMs does not necessarily lead to clots as shown in Figure 3e and 3f.

4. The work is rather descriptive in nature without much insight into molecular mechanisms.

a. Although authors have shown conditional ablation of Rac leads to impaired clearance of laser-induced clot, this not shown for homeostatic and aged conditions.

b. How do the CAMs sense clots? Are there vasculature-derived signals that promote CAM-mediated phagocytosis?

c. Do the vasculatures provide CSF1 for CAMs to survive perivascularly? If so, what effect would disrupting the CSF1-CSF1R axis have on clotting?

Other comments:

5. Human data shown in Extended Data Figure 3 is difficult to assess given the small field of view that has been displayed. Also clarify patient demographics and their underlying conditions as these may affect the numbers of macrophages.

6. Measurement of blood flow velocity and capillary diameter should be carried out both at baseline and after laser ablations to provide essential quantitative data (PMID: 34896022)

Referee #2

(Remarks to the Author)

Mesa et al. show that macrophages are essential in maintaining the health and function of the skin and how their decline with age can contribute to the disorganization of the skin capillary niche. The authors utilize elegant mouse models and imaging technology to characterize macrophage renewal (replenished through local proliferation or recruitment of monocytes) within aging tissues and their response to tissue injury. The study focuses on capillary-associated macrophages (CAMs) and their selective loss over time.

This work is exciting and important for the field, showing that skin macrophages are required to maintain blood flow since loss of CAMs negatively impacts vascular repair and tissue perfusion in aging mice. The concept of a declining macrophage population over time has already been established in different tissues; however, the mechanisms are still not understood. This study shows, for the first time, spatiotemporally resolved macrophage proliferative/repopulation behavior in vivo, which raises additional exciting questions about the macrophage niche and how it can be used to improve tissue function, especially upon aging.

Some experiments and controls should be performed to strengthen the study and the main message of the manuscript. It will be essential to prove that it is not the aging process and the accompanying loss of tissue integrity that is leading to the loss of macrophages and, subsequently to capillary dysfunction. Otherwise, the authors should consider rephrasing their title and the text within the manuscript.

1. The authors should perform a thorough analysis of the extracellular matrix (collagens, laminins etc), especially if they would like to keep the claim of the title since aging of the niche, i.e. the obstruction of the macrophage niche by ECM, may be the leading cause for lack of Csf1 availability and, therefore, lack of macrophage proliferation/recruitment/identity. See also PMID: 37676943 where it has been shown that Kupffer cells can lose their identity if they cannot inhabit their typical tissue niche. The same could happen upon aging and the aging of capillaries may not have anything to do with macrophage loss.

2. Figure 1b: also young animals have few capillaries without macrophages in the upper dermis. Is the blood flow obstructed as well in these regions? If not, the pure lack of macrophages nearby capillaries may not be the sole reason for impaired flow.

3. Figure 1e: What age is shown here? It would be interesting to see if this effect is seen throughout all ages or whether the lack of macrophages in combination with tissue aging is causing vessel obstruction.

4. Extravasation of leukocytes into many tissues is mainly happening at basement membranes of postcapillary venules, if the vasculature is intact. Could the authors try to visualize whether the lack/presence of infiltrating monocytes to replenish the empty niche (e.g. in their laser ablation model) is driven by the presence/absence of laminins?

5. Although two depletion models have been used, both have certain caveats, such as sudden simultaneous and numerous cell death and systemic inflammation. A local depletion, maybe via painting of an a-Csf1r antibody, may provide final proof that macrophage loss is causing capillary dysfunction. At least depletion of another cell type in the dermis (e.g., using Pdgfra-DTR or similar) should be used as a control to ensure that the response to cell death per se is not leading to the observed phenotype.

6. Data on macrophage depletion efficiency in the DT model is missing. Please also show macrophage depletion IF pictures for day 13 after clodronate. Is the increasing capacity for flow correlating with the increasing numbers of macrophages? If that is the case, it would be interesting to address the ontogeny of macrophages supporting blood flow and, in case they are fetal-derived, if the macrophages replacing the depleted population can take over this function.

7. The behavior of macrophages after a laser injury/clot formation in the skin resembles the behavior of cloaking macrophages described by Uderhardt et al (PMID: 30955887). Could the authors check whether it is not rather the cloaking behavior that is inhibited by Rac1 deficiency and not phagocytosis?

8. Why is the readout for the clodronate and DT treatment different? Is there a difference between % of flowing capillaries and obstructed capillary flow? If so, then both measurements should be shown for both depletion models.

9. Age-related anatomical and functional changes of the vasculature have already been described in humans. Please refer to the proper original references (see also overview article PMID 25917013 on this topic).

(Remarks to the Author)

In this submission, Mesa et al. propose that an attrition of tissue resident, perivascular macrophages occurs with age and that this loss of 'protective' macrophages results in accumulation of clots in the capillary bed of skin, with associated decline in local tissue integrity. This concept fits with the now increasingly widely accepted view that such resident macrophages, mainly derived from fetal sources rather than blood monocytes, play a key role in tissue homeostasis rather than having a primary focus on anti-pathogen defense.

The thesis proposed by the authors is an interesting one and some of the data are consistent with the main conclusions. But unfortunately, it appears that there are some major issues with the methods employed that confound interpretation of the data, mostly relating to a failure to consider or test for neutrophil infiltration and swarming in the imaged tissues. They also fail to cite some highly relevant published work showing a role of tissue resident macrophages (TRM) in maintaining tissue integrity and engaging in many of the extension and debris clearance behaviors characterized in this report. Because much of the prior work on neutrophils and RTM related to these concerns comes from my laboratory, I am identifying myself - Ronald N. Germain - as the reviewer.

Substantial clarification, and, likely, new experiments are needed to resolve the experimental concerns, and significant recrafting of the text is required to deal properly with these prior findings.

1. Except for limited low plex imaging data on human skin, all the results derive from mouse experiments employing 2-photon imaging as a main technique. The key finding emerging from these microscopy experiments is clotting in microvessels, based on imaging of RBC flow. The paper provides no explanation for why such clots or flow compromise would occur in the absence of peri-vascular TRM without extrinsic perturbation; they only suggest that once formed, such clots would not be cleared efficiently when these myeloid cells are absent. This lack of an explanation for clot formation exposes a key problem with the study as performed. In the Methods, they write "Albino B6 (B6(Cg)-Tyrc-2J/J, Jax 000058), Csf1rEGFP (B6.Cg-Tg(Csf1r-EGFP)1Hume/J, Jax 018549), Ccr2RFP (B6.129(Cg)-Ccr2tm2.1lfc/J, Jax 017586), R26mTmG (B6.129(Cg)-Gt(ROSA)26Sortm4(ACTB-tdTomato,-EGFP)Luo/J, Jax 007676), R26nTnG (B6N.129S6-Gt(ROSA)26Sortm1(CAG-tdTomato*,-EGFP*)Ees/J, Jax 023537), LysMCre (B6.129P2-Lyz2tm1(cre)lfo/J, Jax 004781), Rac1ff (Rac1tm1Djk/J, Jax 005550) mice were purchased from Jackson Laboratories." The way this is phrased, only B6 mice lacking reporters are albino. There is no indication that the several reporter and DTR mice were bred onto a homozygous albino background. It is well known in the field that 2P imaging cannot be performed on non-albino black mice because of heat damage to melanocytes. This causes intense neutrophilic infiltration of the imaging site. Such neutrophil infiltration results in swarming as described in Lammermann et al. (Nature 2013 Jun 20;498(7454):371-5), who also show (see Fig. 3 in that Nature paper) that such swarms disrupt the local collagen matrix in the skin. Thus, if the experiments of the authors involve animals that are not albino, the imaging can induce neutrophil swarming and subsequent tissue damage that can compress microvessels, leading the clots they see. Uderhardt et al. (Cell 2019 Apr 18;177(3):541-555) showed that RTM play a key role in preventing such neutrophil swarming and local tissue disruption and do so in a spatially constrained, RTM density-dependent manner. Hence, where macrophages are limiting, these effects of laser induced tissue damage and neutrophil swarming will lead to precisely the results reported here, but they will not be the results of 'physiologic' aging events, but rather, the consequence of experimental damage that the authors fail to consider.

Further, it is widely accepted in the field of 2P imaging that all surgical preparations need to be tested using neutrophil reporter mice such as LysM-GFP animals to ensure that the preparation doesn't routinely lead to such infiltration. Either histologic examination of the tissue imaged by 2P after the latter is completed can be used in each experiment to be sure this infiltration doesn't occur intermittently even with a validated surgical method or a blood tracer can be added to the experiments to look for an absence of myeloid cell rolling in the microvasculature. Without evidence that such neutrophil response tracking has been done here and that only albino mice were used for all imaging experiments, it is simply impossible to credit the claims made, as a large part of the biology is potentially occurring "in the black".

2. Even if proper breeding and use of albino mice is the case, many experiments in this paper have a similar major flaw. Laser damage has been shown in multiple papers to lead to rapid neutrophil infiltration and swarming (beside Lammermann and Uderhardt, see J Invest Dermatol. 2011 Oct;131(10):2058-68 and several other Weninger papers). Thus, even in albino mice, when laser ablation is used to remove the perivascular RTM, these experiments will cause intense neutrophil responses that will be unchecked due to the loss of the macrophages. The same problem arises when using DT in DTR mice and clodronate liposomes – both of these treatments are known to promote acute inflammation in response to the induced cell death. This is especially an issue when liposomes are directly injected into the skin site where imaging is then performed, as the injection itself will cause such a neutrophilic response. RTM can modulate this effect, giving a difference between the injected control mice and those receiving the liposomes. In short, the data are all consistent with the report of Uderhardt on the role of RTM in limiting tissue damage from neutrophils invasion and swarming, but this is never assessed or discussed in this paper. Further, the main thesis of RTM protection of vessels against clotting damage is not connected in the submission with the notion of local tissue damage and neutrophil responses, although these might occur at sites of clots and thus, one could imagine an amplification of local damage in regions devoid of RTM due to the addition of neutrophil activity on top of the blood flow limitation.

3. The methods say that the observers are not blinded. This is a potential problem, as there can be a great deal of observer bias in the selection of the specific region(s) to image in animals or in the human skin samples. Two different levels of

concern exist here. First, with the 2P experiments, the choice of the imaging region itself can bias the results, and at the very least, the microscopist should not know if the animals are deficient in RTM or their age when choosing regions to be assessed. Given that in most cases, regions with and without RTM will be obvious in the channel with the RTM fluorescent reporter, the imaging fields (ROIs) should be selected with this channel off and the identity of the animals masked so that truly random fields are collected and assessed. For the human skin data, the high power fields to be imaged should be selected with blinding to the age of the samples and preferably using random selection of the ROI.

4. While it is quite possible that the results about aging and RTM disappearance are correct, the methods used to argue that older hosts have fewer perivascular RTM can be subject to artifacts. For the mice, the authors rely on expression and detection of fluorescent reporters to assess the presence and number of RTM. But if the expression of the relevant marker gene is simply lower on average in the RTM of aged mice and the laser power settings used for the imaging do not compensate for this (which would cause more intense damage and hence, is not a good solution to the problem), then one can miss RTM present in older animals because a greater fraction of them may be simply too dim to be counted. Better would be to do immunohistochemical analysis of the tissue using antibody staining for multiple markers, with the view that not all of them would suffer from such potential age related decrease and in any case, one can increase the laser power or gain during static imaging of fixed tissue without introducing inflammatory artifacts.

5. The CSF-1 studies in Fig 4 are quite intriguing. However, there should have been a third group with no treatment at all, because injection itself will lead to neutrophil responses that promote secondary monocyte infiltration and macrophage differentiation (See Lammermann et al. and Uderhardt et al.). If an increase in monocytes/macrophages improves capillary status in older mice as claimed, then even the PBS treated animals should be 'better' than the untreated mice of the same age due to this increase in tissue macrophages following repeated inoculation. If this is not the case, then the authors need to explain why CSF-1 works but such 'physiologic' recruitment at damage sites does not. If the CSF-1 works by enhancing survival of existing RTM and recruited macrophages are unable to functionally compensate for the 'missing' RTM, that is a very important result, but more work is needed to examine this issue. In any case, the enhancement of vessel integrity by increasing the coverage of RTM, if that is what the CSF-1 is doing, is simply another example of the density-dependent RTM protective effect shown in Uderhardt.

6. As to the scholarship in the paper, as noted above, many of the salient observations made in his paper reprise work of Uderhardt showing how RTM prevent tissue damage (especially the data and conclusions associated with Fig. 2). Indeed the images provided by the authors do not show very local activity of macrophages, but exactly the extension of long processes from neighboring RTM to the site of damage/occlusion as reported in Uderhardt. Not only were these prior observations made using a laser injury model, but in two other situations that included dystrophic mice and the diaphragm in mice deprived of RTM (albeit using DT with some of the same caveats I raise here). This prior work even used perfusion-fixed animals to show that 'cloaking' of damage sites by RTM occurs in the steady state. Uderhardt et al. also explicitly show the capture of cell debris, akin to clot clearance, and this activity was analyzed in greater detail in *Science* (2020 Jan 17;367(6475):301-305). Thus, the new information here is that RTM protection extends to capillaries by more or less the same mechanisms previously reported. IF the authors can show that the effect is unrelated to protection against neutrophil-induced damage, this would be novel, but otherwise it would conceptually be a reprise of the Uderhardt study.

7. In Fig. 1d, there are many clots in even 2 month old mice. Why is this seen? When imaging skin, or any tissue, pressure from the objective can be an issue and in this case could lead to collapse of microvessels and stasis of RBC, as well as cell damage that promotes neutrophil swarming. The author state that they use a coverslip for their imaging and this can often lead to compression damage, both acutely at the global flow level as well as through local necrosis that causes neutrophil recruitment and swarming. This may be exaggerated in aged mice, because the issue is less compliant due to loss of elastin, making the vessels more susceptible to such artifactual obstruction / inflammation responses.

Referee #4

(Remarks to the Author)

In this manuscript, Mesa et al. report interesting novel observations on the dynamics of capillary-associated macrophages (CAMs) in the skin capillary plexus and their impact on capillary functions for repair damage and during aging. The authors utilize powerful intravital two-photon microscopy combined with fluorescent reporter mice for longitudinal cell tracing to show that CAMs are lost with age greater than capillary reduction itself. They go on to show that CAM-less capillaries, either in aging or CAM-ablated cytotoxically or by laser, have reduced blood flow. They demonstrate that CAM loss is largely due to limited local CAM division and replenishment of lost CAMs, but find acute tissue injury such as blood clots can promote CAM proliferation and monocyte recruitment such that even aging capillaries can be rescued. Mechanistically, they identify that CSF1, a secreted growth factor in acute tissue injury, can rescue CAM replenishment in aging mice, which appears to rejuvenate capillary repair capacity.

Despite the complexity of the experiments, the authors manage to present the high quality and comprehensive data in a clear and appealing way to corroborate several main claims of how the CAMs regulate capillary aging. It was a pleasure to read! Overall, this manuscript offers a powerful discovery that CAMs are lost with aging and that CAMs are required for damage repair, maintain blood flow, and prevent such capillary aging-associated effects. With the CSF1 CAM recruitment experiments the authors also provide a potential therapeutic angle for improving blood flow in aged capillaries. In the end,

there are a few select major concerns and minor suggestions that when addressed and implemented would help strengthening the claims further and shore up the findings.

Major concerns:

1. The authors claim that CAM loss promotes capillary aging because of slow blood flow within 14 days after CAM ablation and the association to fewer observed CAMs in aging. However, on its own the link seems a bit weak. It would strengthen the claims if the authors in addition to reduced blood flow after CAM ablation could establish an ongoing loss of capillaries and continued lack of CAM recovery in longer-term follow-up observations, akin to premature aging. For example, what happens for CAM ablation areas (Fig3d) after a couple months or more? Would it mimic the quite strong reductions of capillary density in aging after 10 and 18 months (Ext Data 3a)? Is there still no CAM replacement after a couple months of CAM ablations?
2. Fig. 1e shows that obstructed capillary flow rate is higher in CAM-less capillaries in 1-month old mice. What is the obstructed capillary flow rate in both CAM-present and CAM-less capillaries in aged mice? It would fit and strengthen the claims that CAMs presence and their repair capacity influence the blood flow in aging, if there is still relative obstructed capillary flow in CAM-less capillaries in aged mice.
3. As Rac1 is an important component for phagocytosis it is unsurprising, yet interesting to see, that Rac1 deletion leads to reduced blood clot clearance and reduced blood flow. It is not clear however if Rac1 deletion will influence blood flow in aged CAM-present capillaries to a level of CAM-less counterparts. If it does it would support phagocytosis as a main mechanism. If not, it would open interesting possibilities into CAMs secreting factors or exerting cell-cell contact effects that could affect the blood flow. It would be powerful to figure out how CAMs influence capillary aging.
4. The presence of high numbers of macrophages in the epidermis seemed surprising. It would make sense to make sure that the Csf1r reporter does not label other cells, such as Langerhans cells in the epidermis or other dendritic cells in the dermis.

Minor concerns:

1. It is unclear whether CCR2-dependent replenishment (Ext data Fig. 8c) or CAM proliferation dependent division (Ext data Fig. 10) is the main source of replenishment. Does CSF1 promote CAM division or monocyte recruitment in aging mice?
2. The authors showed most macrophages in upper dermis are CX3CR1+ in 1-month old mice (Ext data Fig 1a and 1f), but Fig 1 only showed Csf1r+ macrophages during extended period up to aging. Is CX3CR1 the same percentage?
3. Ext data Fig 1a and Ext data Fig 1f show CX3CR1+ both in upper dermis and lower dermis, but the numbers are very different. There are more CX3CR1+ cells in Ext data Fig 1f. Do CX3CR1+ cells in the lower dermis have the same division rate? If they have the same division rate, CX3CR1 decreases in the upper dermis could be related to macrophage migration.
4. How does the macrophage population renewal relate to blood circulation? In Fig 1b, the size of blood vessels is different between upper dermis and lower dermis.
5. In Ext data Fig 1a the figure legend describes "images of cells expressing Ccr2-RFP, Csf1r-EGFP or Cx3cr1-GFP", but there is no Csf1r-EGFP.

Version 2:

Reviewer comments:

Referee #1

(Remarks to the Author)

In this revised manuscript entitled "Niche-specific macrophage loss promotes skin capillary aging", the authors report that capillary-associated macrophages (CAMs) in the upper dermis of the skin play an essential role in supporting capillary function. CAMs facilitated homeostatic capillary blood flow (measured by 3rd harmonic generation of RBCs in blood vessels) and maintained capillary density, and loss of CAMs and reduction of their capillary coverage with age resulted in impaired blood flow and pruning of CAM-unassociated capillaries. Mechanistically, authors show CX3CR1 to be essential for sensing laser-induced clotting and reperfusion. RAC1 was required for laser-induced clot removal and reperfusion. CAMs were maintained more by self-renewal than replacement by monocytes, but CAM self-renewal under paced its loss, ultimately resulting in progressive reduction with age. Finally, authors showed that local CAM replenishment by administering CSF1-FC was sufficient to improve capillary blood flow and capillary repair and reperfusion following laser-induced clot formation in aged mice.

The rather heavy focus on macrophages, most of which are incremental in nature, distract from the novel aspect of this work, which is the phenomenon that CAMs contribute to homeostatic blood flow of capillaries, the mechanisms of which are still insufficiently addressed.

Major comments

1. While the authors additionally showed that flowing RBCs within vessels can be further identified by labeling z-stacks in

different colors (revised ED Figure 2d, f), this method has not been applied to other relevant Figures and ED Figures. In line with this, it is somewhat still unclear whether the reduced capillary blood flow and/or obstruction of capillaries in physiological condition is indeed due to the clot formation. Authors should consider applying the z-stack method and quantification to some representative data throughout the manuscript to reliably demonstrate reduced capillary blood flow and physiological clot formation. Additionally, please provide quantifications without both RBC (3rd HG) and RhDex labeling to represent capillary occlusion and clot formation in revised ED Figure 2.

2. Authors have made effort to address mechanisms that underly CAM-mediated regulation of blood flow by showing requirement of CX3CR1 and RAC-1 using Cx3cr1-gfp/gfp mice and Cx3cr1-crexRac1f/f mice, respectively. Authors also show that impaired blood flow and clot removal in aged mice can be improved by injecting CSF1-FC protein. All of these are interesting, but each are incomplete, expected, or lack connection.

- CAM-mediated restoration of clots require CX3CR1, but the observation is rather incomplete without showing the source of the ligand, whether it is activated by clots. Is it possible that clots activate CX3CL1 in capillaries?

- As commented in the last review, impaired phagocytosis in the absence of RAC-1 is expected. Presumably, the functional alteration of CAMs in the absence of RAC-1 is not limited to phagocytosis.

- Rescuing blood flow in aged mice with CSF1-FC protein has translational impact. However, information on which cells provide CSF1 to macrophages is lacking. Do cellular sources of CSF1 decrease with age, or does the expression of CSF1 in tissue decrease with age? Would deletion of CSF1 from the main source lead to impaired blood flow?

Others:

1. Figure 2g Capillary clot recovery following CAM ablation: Did authors ablate CAM in this figure rather simply induced clot formation to see the role of CX3CR1 signaling? Please clarify the title of the figure.

2. Authors show that donor-derived macrophages repopulate in the lower dermis, while CAMs in the upper dermis remain of host origin (Figure 4b). It is also shown that the origin of CAMs does not contribute to blood flow restoration in Extended Data Figure 3d. How can this statement be made when donor- and host-derived macrophages are distributed in different layers? Are clots effectively induced in the bigger vasculature in the lower dermis?

3. In IHCs shown in Extended Data Figure 3g, the spatial association between upper dermal capillaries and CAMs is unclear with the single-color IHC.

4. Patient information is still missing in the revised manuscript. Additionally, the number of human samples analyzed (5 younger group, 6 older group, 11 total) is different from that described in the manuscript (10 written consents were obtained).

Referee #2

(Remarks to the Author)

The authors have successfully addressed the major points raised in the initial review, and their new data provide valuable insights, unveiling interesting findings. However, the integration of these findings suggests a shift in the study's core message, which should now be reflected in the title and abstract and main text. Specifically, the revised data indicate that the loss of macrophages impacts capillary function independently of aging, with aging primarily serving as a confounding factor leading to macrophage depletion. Please see point 1 below for further clarification. Otherwise, the data is convincing and only minor points remain (see below).

1. The current title and abstract emphasize terms such as "niche," "contribute," and "aging," suggesting that niche aging is the central theme of the study. However, the revised dataset indicates that the interaction between macrophages and vasculature, rather than aging itself, drives the observed capillary dysfunction. Aging primarily acts as a factor leading to macrophage depletion rather than being an intrinsic driver of the observed effects. In the authors' rebuttal, they state: "To our surprise we found that regions of aged skin that retained CAMs also retained their ability to repair from laser-induced capillary damage, while regions in the same mice that lacked associated macrophages showed significant impairment to capillary reperfusion (revised Figure 3b)." This finding is, in fact, not surprising if aging serves mainly as a cause for macrophage loss rather than an active driver of capillary dysfunction. Additionally, other experiments in the revision point toward a "model that local macrophage loss, rather than just biological age contributes to capillary function." Since the authors themselves draw this conclusion with the new datasets, maybe then the word "niche" should disappear from the manuscript in general, and the macrophage loss (due to age, or physical/chemical ablation) leading to tissue dysfunction should be highlighted instead.

2. The manuscript predominantly uses an unpaired Student's t-test for statistical comparisons. However, this test is not appropriate for all datasets, especially when data are not normally distributed and/or more than two conditions are being compared. For example, in Figure 4C, an ANOVA test would be more appropriate than a t-test. I recommend carefully reviewing all datasets and applying the appropriate statistical tests throughout the manuscript to ensure robust and accurate conclusions.

3. For clarity and ease of review, I suggest that all changes in the revised manuscript be highlighted. This will facilitate a more efficient evaluation of the updated text and ensure that all key revisions are easily identifiable.

Referee #3

(Remarks to the Author)

The authors have added a substantial amount of new data as well as text revisions in response to the original set of reviews.

These additions and clarifications address many of the major concerns raised in response to the original submission, often by several of the referees and the major conclusions of the authors now appear more firmly supported. I leave it to the other reviewers to determine if the new data are sufficient to convince them that the issues of myeloid cell identity, effects of macrophage aging vs. changes in the surrounding tissue niche, the role of phagocytosis and clearance of debris presumably controlled by Rac, and so on are now adequately resolved. I concentrate here on the points raised just in my review:

1. In response to my point about use of albino mice for 2P imaging of animals on the black (B6) background, they write

“We appreciate the referee’s concern, as we are also aware from our previous studies (PMID: 24097351, 25849774, 26110716, 27229141, 30269903) of the tissue damaging effects of performing multiphoton imaging on skin. Therefore, we would like to clarify that we did indeed cross all of the mice used in the study to the homozygous Albino B6 (B6(Cg)-Tyr^{c-2J/J}, Jax 000058) background.”

While I am surprised that this critical point was omitted in the original Methods, this assurance is an important one in evaluating the data and I accept that they have not inadvertently caused severe local inflammation during their 2P imaging sessions due to melanocyte damage.

2. I also asked that they check for neutrophil infiltration during their imaging sessions apart from the possible effect of melanin-related heat damage. They responded

“We appreciate this concern from the referee as well and would like to clarify that there are no “surgical preparations” in our studies. Unlike the model used in Uderhardt et al. (PMID: 30955887), we do not perform any surgery on our mice. Rather, we directly image the uninjured plantar skin of anesthetized mice as previous described in Rompolas, et al., (PMID: 27229141), Mesa, et al., (PMID: 30269903), Cockburn et al. (PMID: 36357619). Regarding the referee’s comment, “surgical preparation need to be tested using neutrophil reporter mice such as LysM-GFP animals to ensure that the preparation doesn’t routinely lead to such infiltration” we have utilized LysM-Cre mice, which label neutrophils, monocytes, and macrophages (PMID: 32038641, 10621974, 32747818, 18261937, 24857755), with two different fluorescent reporters, Rosa26-mTmG mice and LysM-Cre;Rosa26-dsRed, for several multiday imaging experiments and see no evidence of LysM-expressing cell recruitment or swarming for any of our imaging preparations (revised ED Figure 1 and ED Figure 6f-h).”

I need to state that nearly all these experiments look at 24hrs after the perturbation; this is a very late time point as neutrophil responses to tissue damage occur with 1-2 hours and are usually supplanted by monocyte recruitment by the 24 hr time point, so I do not find most of these data highly relevant to my point. Further, while they are correct that they did not do invasive surgery, they use a cover slip preparation and simple pressure on tissues can cause death of a few cells, sufficient to lead to neutrophil infiltration. Indeed, when Jackson Egen was in my group developing a cover-slipped liver imaging prep, he noticed neutrophils migrating to selected location in the liver parenchyma. He used propidium iodide *in vivo* to look for dead cells and showed that these neutrophils were migrating towards a limited number of individual dead hepatocytes. This was caused by the pressure of the coverslip and relieving this pressure prevented such neutrophil responses. These data also speak to the issue of whether a 10um lesion is “too small” to evoke neutrophil responses. This is incorrect, Uderhardt did experiments with lasers tuned to kill just a single cell and saw neutrophil swarming in the absence of tissue resident macrophages – not all his lesions were the 30u microlesions mentioned by the authors. He also observed neutrophil accumulations around single dead diaphragm muscle fibers in macrophage depleted hosts. Further, as I just stated Egen saw neutrophil responses to single dead hepatocytes and Robey studying toxoplasma saw swarming in response to death of single macrophages (PMID: 18718768). Thus, this issue can only be resolved by the type of careful reporter-based imaging I suggested.

3. They go on to address this issue in detail, stating

“1. No recruitment of Gr-1+ (Ly-6G/Ly-6C) neutrophils or monocytes to laser-induced clots with macrophage ablation: To assess any neutrophil or monocyte recruitment during laser ablations, we performed simultaneous laser-induced clots and CAM ablation experiments as in revised Figure 3, but with low-dose Gr-1-AF647 antibody I.V. labeling (Clone RB6-8C5) in Csf1r-EGFP mice, which labels both neutrophils and monocytes (AF647+/EGFP+) (PMID: 17438263). We also performed intradermal 28g insulin needle stick injuries in the plantar skin as a positive control for neutrophil swarming. Please see revised Extended Data Figure 6a-e that clearly shows that laser-induced clots and laser induced CAM ablations do not recruit swarms of Gr-1+ cells at 6hr or 24hr post laser damage.”

While 6 hrs is probably early enough to see swarms if they existed, I do not understand the hesitancy to do imaging at earlier times and using the LysMGFP animals rather than low amounts of anti-GR1 ab as a label.

2. No recruitment of LysM+ neutrophils or monocytes to laser-induced clots with macrophage ablation: We utilized the LysM-Cre mouse, which targets neutrophils, monocytes, and macrophages (PMID: 32038641, 10621974, 32747818, 18261937, 24857755) crossed with the Rosa26-dsRed Cre-induced reporter, and performed simultaneous laser-induced clots and CAM ablation experiments as in revised Figure 3. Please see Extended Data Figure 6f-h that clearly shows that laser-induced clots and laser-induced CAM ablations do not result in recruitment of swarms of LysM-expressing cells at 24hr post laser damage.”

Again, 24 hrs is not the right time point for this experiment, as the initial neutrophil response will have been replaced by a monocytic response by then, as observed in Uderhardt and also by Kubes (PMID: 25800956).

3. "No recruitment of Csf1r-EGFP+ monocytes or neutrophils to laser-induced clots and/or macrophage depletion: In Lammermann et al. and Uderhardt et al., it is clearly demonstrated that macro-lesions and macrophage-depleted micro-lesions generate tissue damage that will recruit swarms of both neutrophils and monocytes, the latter accumulating for at least one day after laser damage. To assess any monocyte or neutrophil recruitment during laser ablations and/or macrophage depletion, we used Csf1r-EGFP mice, which effectively labels both monocytes and neutrophils (express Csf1r mRNA but not protein) in vivo (PMID: 17438263, 32747818). Using this monocyte/neutrophil readout (Csf1r-EGFP), we have extensively looked at multiple timepoints (including 1 and 24 hours after laser damage) in our experiments and find no evidence for monocyte or neutrophil swarming after laser-induced clotting (revised Figure 3g and 6f; revised ED Figure 5), laser-induced macrophage ablation (revised Figure 4a), simultaneous laser-induced clotting and laser-induced macrophage ablation (revised Figure 3e), or macrophage depletion via clodronate (revised Figure 1g; revised ED Figure 4f) or diphtheria toxin (revised ED Figure 4b) treatment."

These are the most relevant data and I must admit that the evidence presented using this model is very clear about a lack of myeloid cell (neutrophil or monocyte) accumulation early or late after any of the clotting or depletion manipulations used by the authors. I cannot explain why they find that laser ablation of macrophages fails to incite a response, given all the data I note above about such responses to individual cell death in various tissues as studied by others, but accept that they have addressed the issue carefully and do not find such responses in their model.

4. Laser clot and macrophage ablation size is significantly smaller than micro- or macro-lesions described in Uderhardt et al.: In the studies by Lammermann et al. and Uderhardt et al., the authors generate significantly larger sites of laser-induced damage ranging from ~30 microns (microlesion) to ~100 microns (macrolesions) in diameter. Both our laser-induced clotting and macrophage ablation models generate a damage site <10 microns in diameter ...".

I have addressed this issue above – even single cell ablation or death causes neutrophil recruitment in multiple models including those without overt large scale tissue inflammation such as the liver cover slip situation or laser-induced death of a single cell.

4. The methods say that the observers are not blinded. This is a potential problem, as there can be a great deal of observer bias in the selection of the specific region(s) to image in animals or in the human skin samples. ... The authors answer addresses this issue.

5. While it is quite possible that the results about aging and RTM disappearance are correct, the methods used to argue that older hosts have fewer perivascular RTM can be subject to artifacts. For the mice, the authors rely on expression and detection of fluorescent reporters to assess the presence and number of RTM. But if the expression of the relevant marker gene is simply lower on average in the RTM of aged mice and the laser power settings used for the imaging do not compensate for this (which would cause more intense damage and hence, is not a good solution to the problem), then one can miss RTM present in older animals because a greater fraction of them may be simply too dim to be counted....

The authors have now used several orthogonal methods to address this concern and I do not believe that my concern about signal loss can explain their data.

6. The CSF-1 studies in Fig 4 are quite intriguing. However, there should have been a third group with no treatment at all, because injection itself will lead to neutrophil responses that promote secondary monocyte infiltration and macrophage differentiation (See Lammermann et al. and Uderhardt et al.). If an increase in monocytes/macrophages improves capillary status in older mice as claimed, then even the PBS treated animals should be 'better' than the untreated mice of the same age due to this increase in tissue macrophages following repeated inoculation. The authors nicely clarify why this is not a likely scenario, although it relies on the absence of detection of myeloid cell influx under conditions that as I state above, I find remarkable. However, that is what their data show, and given the findings, I do not think my objection here is valid.

7. I leave the issue of novelty to the editors.

8. In Fig. 1d, there are many clots in even 2 month old mice. Why is this seen? When imaging skin, or any tissue, pressure from the objective can be an issue and in this case could lead to collapse of microvessels and stasis of RBC, as well as cell damage that promotes neutrophil swarming.

The authors reprise their statements about lack of overt inflammation in their preparations but this is not the issue. Rather, it is the pressure from the coverslip that can cause cell damage / death as I note above in the work of Egen. ED Fig. 3AB show that loss of flow is minimal in this experiment in 6 month old mice – whether in older mice with stiffer tissue more pressure is transmitted and more effects are seen is unknown.

Referee #4

(Remarks to the Author)

In this revised manuscript, Mesa et al. provide many new experiments and explanations that to a great extent sufficiently address concerns raised by us and other referees regarding their innovative intravital imaging study of macrophage function in the skin capillary plexus and its decline with aging. The addition of long-term fate mapping of CAM-less capillaries; cross-validation of macrophage subpopulations using improved cellular markers; assessment of turnover dynamics using bone marrow chimeras; and substantive supplementary data ruling out concerns of neutrophil swarming and tissue damage from

their laser-induced clotting model all bring additional strength to the main conclusions.

Several issues were nicely addressed as described below. Yet, one main concern remains: The long-term outcome of CAM ablation is still unclear, either the one after laser- or DT-mediated killing (now in new Figure 5a-d; previous 3d-g) or after clodronate ablations (now in ED Figure 4f,g; previous main Fig 1g,h). In previous main concern #1, we asked whether CAM ablations would mimic early in age, and longer than the only the observed 2 weeks, the strong reductions of capillary density seen in aging after 10 and 18 months. And equally important, whether after CAM ablations there would be still no CAM replacement after a more meaningfully longer period than the 14 days (laser, DT) or 13 days (clodronate).

Answering these two points would strongly support the hypothesis that CAM loss promotes capillary aging due to slow blood flow early after CAM ablation and akin to reduced CAMs in aging. The presented CAM dysfunction assays by Rac deficiency resulting in induced capillary pruning, while highly interesting, do not fully address the experimental loss of CAMs issue (like CAM loss in age). If any new significant macrophage numbers would come in soon after 2 weeks to rescue CAM loss and impaired blood flow, it would indicate that additional factors exist that inhibit macrophage proliferation or migration in aging.

Other well-addressed issues:

Although the initial manuscript already provided substantial evidence connecting the absence of CAMs with obstructed capillary blood flow, the connection to aging and loss of capillary density was not so clear. With revised Figure 2a-c the authors now demonstrate that CAM-less capillaries are significantly pruned in the aged mouse. Examination of capillary obstruction in the presence/absence of CAMs across different aging time points in ED Figure 3c further validates their pivotal role in repair and maintenance of capillary function.

The addition of Cx3cr1 and LysM quantifications in the upper dermis of young and old mice help support the previous observations using the global macrophage reporter Csf1r and increase confidence in the observed dynamics from their Cx3cr1 single-cell labelling experiments. The overlap between CD206+ and Csf1r+ cells is convincing, though the legend descriptions could be improved for clarity. Overall, their definitions of resident macrophages within the capillary niche are much improved and convincing.

The use of bone marrow chimeras is a sophisticated way of showing low contribution of monocytes to the upper dermis, eliminates monocyte recruitment as a mechanism for CAM expansion following CSF1 treatment, and posits a local self-renewal strategy contributing to niche disorganization. The additional data tracking single Cx3cr1+ macrophages in the upper and lower dermis properly address our concerns regarding movement between compartments.

Minor: the figure legend in ED Fig 1a still lists Csf1r-EGFP, even though there is none and it was stated in the response that it was corrected. It was not corrected.

Version 3:

Reviewer comments:

Referee #1

(Remarks to the Author)

The authors have done an excellent job in responding to my and fellow reviewers' comments. I think they provide compelling evidence on the role of capillary associated macrophages in maintaining skin capillary function. One comment the authors have not responded to is the human subject information (age, sex, site of biopsy). Related to this, the IHC that is provided in Extended Figure 3 to make the case that human CAMs also decline with age, does not clearly show the capillaries in aged skin. Two-color staining (e.g. CD68 + CD31) would make a stronger case.

Referee #4

(Remarks to the Author)

In this revised version, overall, the authors have sufficiently addressed our concerns regarding a longer-term outcome of laser-induced CAM loss. Specifically, the authors now included an additional revisit time point at 8 weeks following laser-induced ablation of CAMs in the upper dermis (Fig 5d) and similarly quantify the frequency of CAM replacement as with the 2-week time point. The sustained significant reduction in CAM replacement when compared to the "CAM loss + clot" model continues to support the conclusion that local tissue damage is required to replenish CAMs in the upper dermal niche and promote their renewal. Although they do not quantify the frequency of CAM replacement with the "CAM loss + clot" model at 8 weeks, the authors demonstrate elsewhere in their manuscript (e.g. Fig 5h, Fig 3e, ED Fig 5c) that blood flow is reestablished as early as 7 days following laser-induced capillary clotting/damage. Therefore, it can reasonably be inferred that the frequency of CAM replenishment in the "CAM loss + clot" model represents a new homeostatic condition directly comparable to the 8-week time point without clotting.

The authors also now include an analysis of capillary blood flow at 8 weeks following laser-induced CAM loss (ED Fig 9a). This directly addresses the original concern as to whether laser-induced CAM loss in young mice mirrors the reduced capillary density brought about by the homeostatic loss of CAMs in the aged mouse.

Ideally, the authors would have also extended their revisit window to 8 weeks following chemical ablation of CAMs using high dose IP injection of DT and observed a similar reduction in CAM recruitment and capillary blood flow/density. Additionally, the described relationship between tissue damage and CAM recruitment could have been strengthened by performing laser-induced capillary clots at the 8-week time point and observing whether CAMs are still recruited a week or even a few days later. Even in the absence of this, the data provided in this revised manuscript has successfully addressed our initial concerns and supports the model that the loss of CAMs in the upper dermal niche promotes capillary aging, and that their recruitment is dependent on a tissue damage response.

To that point, we agree with other referees in that the central theme of the study appears to be the interaction between macrophages and vasculature, rather than their absolute presence/absence in the upper dermal niche. This focus could be better reflected in the title, perhaps as “Loss of macrophages in the upper dermal niche promotes skin capillary aging.”

Referees' comments:

Referee #1 (Remarks to the Author):

In this study titled, "Niche-specific macrophage loss promotes skin capillary aging," the authors recognize that the effects of aging on tissue resident macrophage function and the cellular mechanisms behind such effects are poorly understood. As such, Mesa et al. seek to further elucidate the organization, behavior, and function of tissue resident macrophages within the skin capillary plexus niche (capillary-associated macrophages, or CAMs, hereafter) within an aging context.

Using intravital two-photon microscopy to track CAMs in vivo within the skin capillary nexus, the researchers report a close functional relationship between CAMs and blood capillaries, initially finding that capillaries that lacked associated macrophages displayed obstructed blood flow and that CAM populations appear to diminish at a faster rate than capillaries themselves. Furthermore, using laser-induced blood clotting models, the researchers find that CAMs support capillaries via phagocytosis of red blood cell debris at capillary repair sites. The authors also report that this support may be limited with age as they find that CAMs fail to sustain or replenish nearby macrophage populations, resulting in a cumulative loss of macrophage populations with age. Finally, the authors report that sufficient capillary repair in aged mice can be achieved with either extrinsic cues such as broad tissue damage or addition of growth factors such as CSF1, both of which resulted in greater capillary repair and tissue reperfusion. While this manuscript is of potential interest, there are major concerns regarding the conclusions that are derived from their experimental data. In particular, it is unclear a) if the discontinuous 3rd harmonics flow in fact represents clots b) what the cell type that mediate clearance is, and c) mechanisms involved in CAM-mediated clot clearance is not explored in-depth. Authors might consider the below comments to strengthen their manuscript.

Concerns and Limitations

Major:

1. The association between CAMs and blood vasculature, their subsequent interactions, and the potential regulation of vascular physiology, have been previously documented (PMID: 34489419, 23451163, 36737792), suggesting that the concept of this study is not completely novel. The finding that CAMs help sustain the blood flow in capillary is of potential interest, but there are questions regarding the validity of this concept as described below.

We agree with the referee that it is well-known that macrophages are found in association with the tissue vasculature. However, our understanding of their tissue functions remains quite limited. This is primarily due to our inability to specifically label and manipulate these macrophage populations. The study (PMID: 34489419) provided by the referee is a perfect example. The authors use broad macrophage depletion strategies such as PLX3397 (which depletes all macrophage populations in the body) or global gene knockout mice to assign specific capillary-associated macrophage control on brain vasodilation. While such studies provide a general link between macrophages and vascular function (as well as most other tissue functions), it remains unclear which macrophage populations control which tissue functions. Furthermore, it is largely unknown how these macrophage populations are affected by tissue aging and if this contributes to the age-associated defects we find in most tissues, such as reduced vascular density and blood flow. In our study, we identified a population of capillary-associated macrophages that have an essential role in maintaining capillary blood flow and that is lost with age through impaired self-renewal and tissue distribution, thus exposing the aging

capillary network to ischemic events. In addition, the novelty of our work lies in our ability to track and manipulate individual macrophage behaviors *in vivo* and specifically probe the functional role of these macrophages within the capillary niche.

2. Authors to identify obstructed capillary flow by utilizing 3rd harmonics. However, due to the discontinuous nature of the 3rd harmonics signal even without any manipulation, it is difficult to ascertain that discontinuous signal in fact represents disrupted capillary flow or clotting. It is rather difficult to accept that natural clots occur so frequently, as displayed in Figure 1d-f. In general, the difference in 3rd harmonics signals between capillaries with or without clots seem very subtle. Authors are encouraged to validate the existence of clots by other objective and quantifiable measures, both in experimental and WT controls.

We appreciate the referee's concern and have provided new experiments and quantifications to compare 3rd harmonic signal of red blood cell (RBC) tracking to the standard approach of intravenous injection of rhodamine dextran. We have provided this comparison both during homeostasis (revised ED Figure 2) and following laser-induced capillary damage/clotting (revised ED Figure 5).

During homeostasis we find robust overlap (>90%) of rhodamine dextran positive capillary segments with the presence of red blood cells (via 3rd harmonic signal) (revised ED Figure 2a-c and Supplemental Video 3). To assess if the red blood cells present in the capillary segment were flowing through the vasculature, we utilized "the discontinuous nature of the 3rd harmonics signal" to our advantage to distinguish flowing capillaries from those without RBC flow (revised ED Figure 2d-f). Specifically, we performed three-dimensional image acquisition of the superficial capillary plexus with 0.5 μ m resolution in the x,y dimensions and 1 μ m resolution in the z dimension. This allowed us to capture flowing red blood cells at different x,y positions for each z-section (~1sec per section) along the capillary segment (which is ~5-10 μ m in diameter). In contrast, RBCs that did not move during acquisition were found at the same x,y position. Pseudo coloring each z-step through a capillary segment clearly distinguishes flowing RBCs as a multicolor (rainbow) patchwork from non-flowing RBCs as white (full overlap of all colors). Therefore, utilizing this methodology we could readily identify capillary segments with or without RBC flow without the need for repeated intravenous injections of dyes or beads, which have limitations (including photobleaching, phototoxicity, and progressive diffusion out of the vasculature) that would be incompatible for the long-term repeated imaging required for this study. Utilizing 3rd harmonic imaging, we find that ~90% of capillaries have RBC flow when there is a CAM present on the vessel in mice (1-18 months old) (revised ED Figure 3c and Supplemental Video 5).

Lastly, during laser-induced clot formation, we show that capillary flow (visualized by both 3rd harmonics and I.V. rhodamine dextran) is completely blocked immediately following laser ablation (revised ED Figure 5 and Supplemental Video 6). Interestingly, we find that this is not due to a collapse of the capillary lumen, as we find the lumen to be transiently expanded with an obstructed mass/clot of RBC debris (revised ED Figure 5c-d).

3. It is not clearly defined which cell type(s) mediate the clearance of clots. Neither CX3CR1 nor CSF1R are exclusive macrophage markers, given their frequent expressions in various cell types, including monocytes and dendritic cells (PMID: 32582197, 29503738, 16034075), emphasizing the importance for the authors to incorporate additional markers in combination for distinguishing these subpopulations. In Figure 1b, understanding of the distribution of capillary-associated macrophages could be significantly improved via an orthogonal view of z-stack

images in the epidermis, upper dermis, and lower dermis. The use of other well-defined macrophage markers (e.g. CD64) is strongly encouraged.

We have provided additional data to confirm that the capillary-associated cells are indeed macrophages.

First, these capillary-associated cells express both CSF1R and CX3CR1, which would strongly suggest them to be of myeloid origin (macrophage/monocyte, dendritic cells) (PMID: 12393599, 29503738) (revised ED Figure 1a-b). Additionally, there is a small fraction of these cells that also express CCR2, which may suggest a recent monocyte origin. However, as they are no longer in the vasculature and have adopted a ramified morphology, this would suggest that they have differentiated into macrophages.

Second, we find that capillary-associated macrophages are also robustly labeled with LysM-Cre, which does not effectively label classical dendritic cell populations, but rather macrophages/monocytes and granulocytes (revised ED Figure 1g) (PMID: 10621974, 24857755).

Third, we also find that the vast majority of CSF1R expressing upper dermal cells also express the canonical marker for phagocytic tissue macrophages, CD206 (Mrc1) (PMID: 34995099, 36450771, 28432199) (revised ED Figure 1h).

Lastly, we find that with all these labeling approaches there is a consistent loss in cell density with age (revised ED Figure 1). Therefore, collectively we feel this provides comprehensive evidence that these capillary-associated CSF1R-expressing cells are indeed macrophages that are progressively lost with age.

It also appears that DT-induced or homeostatic loss of CAMs does not necessarily lead to clots as shown in Figure 3e and 3f.

We would like to clarify that while we did not present this data in Figure 3, we have already quantified capillary blood flow during both DT-induced (revised ED Figure 4a-e) and homeostatic loss of CAMs (revised Figure 1d-e; revised ED Figure 3c) and find significant loss in capillary RBC flow. Specifically, we find a reduction in flowing capillary density following 5 days after partial CAM depletion via DT-treatment in CX3CR1-DTR mice (revised ED Figure 4a-e). Additionally, we find across all ages assessed (1, 2, 4, 10, and 18 months) a significant loss in RBC blood flow in capillaries lacking associated macrophages (revised Figure 1d-e; revised ED Figure 3c). Therefore, these data support a model where macrophage loss predicts and contributes to vessel obstruction during the aging process.

To assess the long-term fate of vessels that lack a CAM, we performed a 6-month time-course in *Cx3cr1-GFP;R26-mTmG* mice from 1 to 7 months of age. We find that while CAM coverage decreases with age as previously shown in Figure 1f, we also find that these “macrophage-less” capillaries are much more likely to be pruned and permanently eliminated for the remainder of our revisits (revised Figure 2a-c). Collectively, these findings further strengthen the hypothesis that capillary-associated macrophages are required for maintaining proper capillary function and, without replenishment of new CAMs, these capillaries will eventually be pruned away, contributing to vascular rarefaction, a hallmark of vascular aging (PMID: 37193857, 34326210).

*4. The work is rather descriptive in nature without much insight into molecular mechanisms.
a. Although authors have shown conditional ablation of Rac leads to impaired clearance of laser-induced clot, this not shown for homeostatic and aged conditions.*

We thank the referee for the suggested experiments, which would provide additional mechanistic insight into the homeostatic role of Rac1 in maintaining capillary function. To this end, we performed experiments where we compared homeostatic capillary blood flow in *Cx3cr1-CreER;Rac1fl/fl* vs *Cx3cr1-CreER;Rac1fl/+* mice after a three month period following tamoxifen administration. We find that over this 3-month period there is an acceleration in rate of capillary pruning (~20% reduction in density) in the skin of mice containing Rac1-deficient macrophages (revised Figure 3h-j). Therefore, it is likely that Rac-1 dependent behaviors, such as phagocytosis of vascular and/or perivascular debris, is required for both repair and preservation of the skin microvascular network. In alignment with this working model, we find laser-induced clotting led to an accumulation of unengulfed perivascular debris in Rac-1 deficient mice, as compared to control mice (revised Figure 3e-f). Collectively, this work provides deeper insight into the molecular mechanisms that underpin the interplay between capillaries and their associated macrophages in actively removing debris and preserving vascular density across the lifetime of the organism.

b. How do the CAMs sense clots? Are there vasculature-derived signals that promote CAM-mediated phagocytosis?

This is an interesting question. Previous work has highlighted the role of multiple signaling pathways involved in sensing tissue damage and/or cellular debris (PMID: 26287597, 30955887). One such pathway is mediated through the chemokine-ligand interaction of CX3CR1-CX3CL1. CX3CL1 is expressed by many cell types including epithelial and endothelial populations and can be released following tissue damage and cell death (PMID: 31998312). CX3CR1 is expressed by many different macrophage populations and is known to support various aspects of macrophage behavior, including sensing apoptotic cells and tissue damage and well as homeostatic cellular morphology and projections (PMID: 34163473). In mouse skin, it is known to be highly expressed by nerve-bundle associated macrophages located deep in the skin hypodermis (PMID: 31201094). However, in the dermis we also find that nearly all CAMs express CX3CR1, as do a small fraction of perivascular macrophages in the lower dermis (revised ED Figure 1c).

Therefore, we performed multiple experiments to interrogate the role of CX3CR1 signaling in recruiting nearby CAMs to capillary damage. First, we tested if CX3CR1 had a functional role in CAM-mediated repair of capillary clots. We performed laser-induced clots in the both *Cx3cr1-gfp/+* (control) or *Cx3cr1-gfp/gfp* (KO) mice and found significant impairment in CAM recruitment in KO mice, as compared to controls (revised Figure 2g-h). Functionally, we found that CX3CR1-deficiency also led to a significant delay and overall reduction in capillary reperfusion by Day 7 post clot induction (revised Figure 2i).

Second, to assess the long-term effect of CX3CR1 loss with homeostatic aging, we tracked capillary blood flow during physiological aging. While 1 month old KO mice showed no significant change in the number of flowing capillaries, we found that by 6 months of age there was a significant loss in flowing capillaries (revised Figure 2j). Therefore, these experiments support a role for CX3CR1-signaling in facilitating the interactions between capillaries and their associated macrophages during both tissue injury and for the long-term preservation of the vascular network during physiological aging.

c. Do the vasculatures provide CSF1 for CAMs to survive perivascularly? If so, what effect would disrupting the CSF1-CSF1R axis have on clotting?

It is known that most resident macrophage populations depend on CSF1R ligands, CSF1 or IL34 (PMID: 34742624), for survival, as disruption of any of these components leads to widespread macrophage loss throughout the body. Multiple populations are known to produce CSF1R ligands in the skin, including fibroblast, epithelial and endothelial populations (PMID: 35930670, 30066957, 35508166). However, to the referee's point, it is unclear what CSF1-signaling is providing for this CAM population as well as how it impacts capillary function. In our previous submission, we demonstrated that exogenous CSF-treatment in aged mice increases the number of CAMs and importantly restores RBC flow in a fraction of capillaries as well as improves capillary repair after laser-induced clotting (revised Figure 6).

This result did not clarify the mechanism by which CSF1 was increasing CAM numbers in aged skin. To assess if CSF1-treatment modulated local CAM survival/proliferation or simply recruited new monocyte-derived macrophages from the blood, we performed a CSF1 treatment in bone marrow chimeric mice in which we could track the relative expansion of local and recruited populations. We found that CSF1-induced CAM expansion did not alter the ratio of resident (GFP+) and recruited (GFP+/dsRed+) CAMs (revised ED Figure 10). This finding strongly suggests that CSF1 is driving local proliferation of the existing CAM population, as a relative increase in GFP+/dsRed+ CAMs over GFP+ CAMs would have suggested monocyte recruitment. Interestingly, we also did not detect a relative increase in GFP+ CAMs, which suggests that CSF1 treatment could promote proliferation in both long-term (> 9months) and more recent monocyte-derived resident CAM populations.

Other comments:

5. Human data shown in Extended Data Figure 3 is difficult to assess given the small field of view that has been displayed. Also clarify patient demographics and their underlying conditions as these may affect the numbers of macrophages.

We have provided larger fields of view (revised ED Figure 3g) as well as a more detailed description of patient demographics. Specifically in the Methods section of the main text, we now include, "Clinical information and laboratory data were obtained from the electronic medical record. Sex and gender information was not used. Patients in the young group were below 40 years of age. Patients in the old group were above 40 years of age. Patients with skin or vascular pathologies were excluded. Written informed consent was obtained for postmortem examination from next of kin for 10 patients."

6. Measurement of blood flow velocity and capillary diameter should be carried out both at baseline and after laser ablations to provide essential quantitative data (PMID: 34896022)

As described in our response to the referee's second point, we have used the 3rd harmonic method to identify capillary segments with or without RBC flow, rather than flow velocity, as the methods to measure precise blood flow velocity require many repeated scans of the same capillary plane in a short period of time. Furthermore, such repeated measurements in the short term (as would be needed for most of our experimentation) could also lead to phototoxic effects such as altered cell behavior, cell death, and tissue damage.

Again to the referee's point, during laser-induced clot formation, we show that capillary flow (visualized by both 3rd Harmonics and I.V. rhodamine dextran) is completely stopped immediately following laser ablation (revised ED Figure 5a-c and Supplemental Video 6). We

measured capillary diameter as the referee requested and find the lumen to be temporarily expanded with an obstruction mass/clot of RBC debris (revised ED Figure 5 c-d).

Referee #2 (Remarks to the Author):

Mesa et al. show that macrophages are essential in maintaining the health and function of the skin and how their decline with age can contribute to the disorganization of the skin capillary niche. The authors utilize elegant mouse models and imaging technology to characterize macrophage renewal (replenished through local proliferation or recruitment of monocytes) within aging tissues and their response to tissue injury. The study focuses on capillary-associated macrophages (CAMs) and their selective loss over time.

This work is exciting and important for the field, showing that skin macrophages are required to maintain blood flow since loss of CAMs negatively impacts vascular repair and tissue perfusion in aging mice. The concept of a declining macrophage population over time has already been established in different tissues; however, the mechanisms are still not understood. This study shows, for the first time, spatiotemporally resolved macrophage proliferative/repopulation behavior in vivo, which raises additional exciting questions about the macrophage niche and how it can be used to improve tissue function, especially upon aging.

Some experiments and controls should be performed to strengthen the study and the main message of the manuscript. It will be essential to prove that it is not the aging process and the accompanying loss of tissue integrity that is leading to the loss of macrophages and, subsequently to capillary dysfunction. Otherwise, the authors should consider rephrasing their title and the text within the manuscript.

We appreciate the referee's comments and agree that many other components of the capillary or upper dermal niche will eventually change with age as well. We indeed find that there is a significant reduction in fibroblasts in both the upper and lower dermis of aged mice (data not shown) as has been previously demonstrated (PMID: 30415836), which could contribute to changes in the ECM with age. Therefore, we agree with the reviewer that these changes may contribute to the macrophage loss we observe. However, to be clear, we do not claim in the manuscript or title that macrophage loss is independent of or precedes all other signs of skin aging. Rather, this study supports a model where the progressive decline in macrophage density itself contributes to impaired capillary maintenance and ultimately long-term loss in skin capillaries, which is a major hallmark in vascular aging (PMID: 37193857, 34326210).

1. The authors should perform a thorough analysis of the extracellular matrix (collagens, laminins etc), especially if they would like to keep the claim of the title since aging of the niche, i.e. the obstruction of the macrophage niche by ECM, may be the leading cause for lack of Csf1 availability and, therefore, lack of macrophage proliferation/recruitment/identity. See also PMID: 37676943 where it has been shown that Kupffer cells can lose their identity if they cannot inhabit their typical tissue niche. The same could happen upon aging and the aging of capillaries may not have anything to do with macrophage loss.

As stated above, we do not claim in the manuscript or the title that macrophage loss contributes to "aging of the niche". Rather we claim that local macrophage loss promotes hallmarks of aging in the capillary vasculature itself, including repair defects, loss in blood flow, and increased rates of vessel pruning. Therefore, while we agree with the referee that it is important to understand

how other aspects of tissue aging, such as ECM changes, could contribute to macrophage loss, it is not related to the central focus of this study and should be addressed in future work.

To address the second point of the referee, "*and the aging of capillaries may not have anything to do with macrophage loss*", we have performed several experiments to test the role of macrophage loss (and not age) more directly on capillary flow and pruning.

First, to assess if diminished capillary repair is related to macrophage loss or, instead, from other tissue-wide factors (ECM) in aged mice, we took advantage of the inherent variability in macrophage density in older mice. Specifically, we performed the same laser-induced capillary clot model in regions of aged mouse skin with or without local macrophages (within 75 μ m from clot) (revised Figure 3a). To our surprise we found that regions of aged skin that retained CAMs also retained their ability to repair from laser-induced capillary damage, while regions in the same mice that lacked associated macrophages showed significant impairment to capillary reperfusion (revised Figure 3b). While it is still possible that local changes to the ECM could drive these local CAM changes, these data again support the major claim of the manuscript that local macrophage loss contributes to age-associated capillary dysfunction.

Second, to assess if macrophage functional loss (Rac1-deficiency), rather than other aging factors (such as tissue integrity) can drive capillary pruning, we performed experiments where we will compare capillary density in *Cx3cr1-CreER;Rac1^{fl/fl}* vs *Cx3cr1-CreER;Rac1^{fl/+}* mice after a three month period following tamoxifen administration. We found that over this 3-month period, there was acceleration in rate of capillary pruning (~20% reduction in density) in the skin of mice containing Rac1-deficient macrophages (revised Figure 3h-j). Therefore, it is likely that loss of Rac-1 dependent behaviors, such as phagocytosis of vascular and/or perivascular debris, directly contributes to impaired recovery and preservation of the skin microvascular network. In alignment with this working model, we found that following laser-induced clotting there was accumulation of unengulfed perivascular debris in Rac-1 deficient mice (revised Figure 3e-f).

Collectively, this work provides deeper insight into the molecular mechanisms that underpin the interplay between capillaries and their associated macrophages. Importantly for the referee's concern, we provide multiple lines of evidence that support a direct role of CAMs for supporting capillary repair and preserving vascular function across the lifetime of the organism.

2. Figure 1b: also young animals have few capillaries without macrophages in the upper dermis. Is the blood flow obstructed as well in these regions? If not, the pure lack of macrophages nearby capillaries may not be the sole reason for impaired flow.

We appreciate this question from the referee and have performed the suggested analysis across multiple ages. We find, across all ages assessed (1, 2, 4, 10, and 18 months), a significant loss in RBC blood flow in capillaries lacking associated macrophages (revised Figure 1d-e; revised ED Figure 3c). Therefore, this observation supports the model that local macrophage loss, rather than just biological age contributes to capillary function.

3. Figure 1e: What age is shown here? It would be interesting to see if this effect is seen throughout all ages or whether the lack of macrophages in combination with tissue aging is causing vessel obstruction.

We thank the referee for this question and would like to clarify that the mice used in Figure 1e were treated with tamoxifen at 1 month of age and imaged for capillary flow 1 month later. For

the referee's point on macrophage loss throughout all ages, as described in the previous point (#2), we do find that macrophage loss predicts capillary obstruction in an age-independent manner (revised ED Figure 3c).

To further address the referee's questions and assess the long-term fate of vessels that lack a CAM, we performed a 6-month time-course in *Cx3cr1-GFP;R26-mTmG* mice from 1 to 7 months of age. We find that while CAM coverage decreases with age as previously shown in Figure 1f, we also find that these "macrophage-less" capillaries are much more likely to be pruned and permanently eliminated for the remainder of our revisits (revised Figure 2a-c).

Collectively, these findings further strengthen the hypothesis that capillary-associated macrophages are required for maintaining proper capillary function, and without replenishment of new CAMs, these capillaries will eventually be pruned away contributing to vascular rarefaction, a hallmark of vascular aging (PMID: 37193857, 34326210).

4. Extravasation of leukocytes into many tissues is mainly happening at basement membranes of postcapillary venules, if the vasculature is intact. Could the authors try to visualize whether the lack/presence of infiltrating monocytes to replenish the empty niche (e.g. in their laser ablation model) is driven by the presence/absence of laminins?

This is an interesting question, and we agree that changes in laminin levels with age could alter transendothelial migration and potentially explain the lack of monocyte recruitment in skin capillaries after laser damage (PMID: 33193400). However, given that we do not see monocyte recruitment in either young or old mice, a much simpler explanation would be that our damage size is too small to elicit monocyte recruitment (PMID: 23708969, 30955887). To directly test the latter possibility, we decided to modify our laser ablation model to target a larger field of view ($500\mu\text{m}^2$) across the upper dermal capillary plexus (revised Figure 5e). We find that a larger damage area across the capillary niche rapidly recruits large numbers of monocytes from the blood into the capillary niche (revised Figure 5f-g). Therefore, while differences in ECM may still have some role in efficiency of monocyte recruitment to the capillaries, we find that the size of tissue damage is seemingly the main limiting factor for any leukocyte recruitment to the capillary niche.

*5. Although two depletion models have been used, both have certain caveats, such as sudden simultaneous and numerous cell death and systemic inflammation. A local depletion, maybe via painting of an α -Csf1r antibody, may provide final proof that macrophage loss is causing capillary dysfunction. At least depletion of another cell type in the dermis (e.g., using *Pdgfra-DTR* or similar) should be used as a control to ensure that the response to cell death per se is not leading to the observed phenotype.*

We greatly appreciate this comment from the referee. It is not clear if the "two depletion models" referred to are clodronate-liposomes (revised Figure 1g-h) and DT-treatment of *Cx3cr1-DTR* mice (revised ED Figure 4), since the referee goes on to say, "A local depletion, maybe via painting of an α -Csf1r antibody, may provide final proof that macrophage loss is causing capillary dysfunction." If this is the case, we would like to clarify that we have also used local depletion (via laser-induced cell ablation) of resident macrophages within $75\mu\text{m}$ of a capillary clot and find impaired clot repair and reperfusion (revised Figure 3e-f).

The referee's concern that, "At least depletion of another cell type in the dermis (e.g., using *Pdgfra-DTR* or similar) should be used as a control to ensure that the response to cell death per se is not leading to the observed phenotype" is a great idea. To this end, we performed the

same laser-induced clot model in *CSF1R-EGFP;CAG-dsRed* (global expression of the fluorescent protein dsRed - PMID: 15593332) mice, in which we can now target non-CAM perivascular dermal cells (*GFP-/dsRed+*) for laser-ablation. Performing these cellular ablations on the surrounding stroma prior to laser-induced clotting, we found no impairments in capillary repair as quantified by reperfusion recovery (revised ED Figure 7). Therefore, this new result strengthens our claim that macrophage loss, rather than generic local cell death / dermal damage, leads to the observed capillary repair defect.

6. Data on macrophage depletion efficiency in the DT model is missing. Please also show macrophage depletion IF pictures for day 13 after clodronate. Is the increasing capacity for flow correlating with the increasing numbers of macrophages? If that is the case, it would be interesting to address the ontogeny of macrophages supporting blood flow and, in case they are fetal-derived, if the macrophages replacing the depleted population can take over this function.

We thank the referee for their comments and have now provided macrophage depletion efficiency data for both *CX3CR1-DTR* and Clodronate depletion models (revised ED Figure 4c, g). We have also provided images for all revisit timepoints, including Day 13, during the clodronate depletion (revised ED Figure 4f). We find no significant increase in CAM density at Day 13 (revised ED Figure 4g), but it is important to highlight that in this experimental design we provided clodronate every 3 days to maintain low macrophage numbers. Therefore, it is hard to draw strong repopulation/ontogeny conclusions from clodronate depletion alone. Furthermore, clodronate is not specific to CAMs and targets most myeloid cells and therefore we did not utilize this strategy in later studies.

However, to test the referee's final point on macrophage ontogeny, we did generate bone marrow chimeras with *Csf1r-GFP;CAG-dsRed* bone marrow transferred into *Csf1r-GFP* mice. Importantly, the hind paws (where we perform intravital imaging) of these mice were lead-shielded to prevent loss of resident macrophage populations (revised Figure 4a). Tracking all skin macrophage populations over 10 weeks post transplantation showed that while lower dermal macrophages were largely replaced by monocytes, >90% of upper dermal macrophages remained host derived (revised Figure 4b-c). Interestingly, we capillaries with either monocyte-derived (*GFP+/dsRed+*) or host-derived (*GFP+/dsRed-*) CAMs showed no obvious differences in RBC blood flow (revised ED Figure 3d). Therefore, these data would suggest that CAM ontogeny does not play a major role in homeostatic capillary function.

Lastly, we found that CSF1-induced CAM expansion did not alter the ratio of resident (*GFP+*) and recruited (*GFP+/dsRed+*) CAMs. This finding strongly suggests that CSF1 is driving local proliferation of both long-term (> 9months) and more recent monocyte-derived resident CAM populations (revised ED Figure 10c-e).

7. The behavior of macrophages after a laser injury/clot formation in the skin resembles the behavior of cloaking macrophages described by Uderhardt et al (PMID: 30955887). Could the authors check whether it is not rather the cloaking behavior that is inhibited by Rac1 deficiency and not phagocytosis?

We thank the reviewer for this comment and agree that the initial recruitment and projections within the first few hours are like what has been previously described by Uderhardt et al (PMID: 30955887). However, it is important to highlight that macrophage "cloaking" occurs within minutes, while most clot/RBC debris is generated/released into the perivascular space and engulfed by macrophages over the first few days after laser damage (revised Figure 2e). We find these later timepoints of clot repair to be significantly impaired in *Rac1* deficient mice.

Specifically, we find a large increase in unengulfed RBC debris surrounding the damage site at day 3 and a significant decrease in capillary reperfusion at day 7 (revised Figure 3e-g). While we cannot rule out some disruption in initial recruitment or “cloaking”, our data suggest that Rac1 deficiency leads to defects later in the capillary repair process, including RBC phagocytosis and clearance/reperfusion of the capillary lumen.

In contrast, we do find in other experiments that deficiency in the chemokine receptor CX3CR1 does partially block the recruitment and therefore cloaking of the initial laser damage (revised Figure 2g-h).

8. Why is the readout for the clodronate and DT treatment different? Is there a difference between % of flowing capillaries and obstructed capillary flow? If so, then both measurements should be shown for both depletion models.

We thank the referee for this comment and would like to clarify that the readout for clodronate and DT are representing the same measurement. The DT treatment presented in ED Figure 4 (revised ED Figure 4e) displayed the data in an inverted manner by showing bar graphs of the fraction of capillaries without RBC flow. We have corrected this in the revised Extended data Figure 4e and now display our DT treatment data as we do for the clodronate experiment in main Figure 1h.

9. Age-related anatomical and functional changes of the vasculature have already been described in humans. Please refer to the proper original references (see also overview article PMID 25917013 on this topic).

We appreciate the referee's comment and have referred to the proper original references (PMID: 8241073, 9884387, 15302783).

Referee #3 (Remarks to the Author):

In this submission, Mesa et al. propose that an attrition of tissue resident, perivascular macrophages occurs with age and that this loss of ‘protective’ macrophages results in accumulation of clots in the capillary bed of skin, with associated decline in local tissue integrity. This concept fits with the now increasingly widely accepted view that such resident macrophages, mainly derived from fetal sources rather than blood monocytes, play a key role in tissue homeostasis rather than having a primary focus on anti-pathogen defense.

The thesis proposed by the authors is an interesting one and some of the data are consistent with the main conclusions. But unfortunately, it appears that there are some major issues with the methods employed that confound interpretation of the data, mostly relating to a failure to consider or test for neutrophil infiltration and swarming in the imaged tissues. They also fail to cite some highly relevant published work showing a role of tissue resident macrophages (TRM) in maintaining tissue integrity and engaging in many of the extension and debris clearance behaviors characterized in this report. Because much of the prior work on neutrophils and RTM related to these concerns comes from my laboratory, I am identifying myself - Ronald N. Germain - as the reviewer.

Substantial clarification, and, likely, new experiments are needed to resolve the experimental concerns, and significant recrafting of the text is required to deal properly with these prior findings.

1. Except for limited low plex imaging data on human skin, all the results derive from mouse experiments employing 2-photon imaging as a main technique. The key finding emerging from these microscopy experiments is clotting in micro-vessels, based on imaging of RBC flow. The paper provides no explanation for why such clots or flow compromise would occur in the absence of peri-vascular TRM without extrinsic perturbation; they only suggest that once formed, such clots would not be cleared efficiently when these myeloid cells are absent. This lack of an explanation for clot formation exposes a key problem with the study as performed. In the Methods, they write “Albino B6 (B6(Cg)-Tyrc-2J/J, Jax 000058), Csf1rEGFP (B6.Cg-Tg(Csf1r-EGFP)1Hume/J, Jax 018549), Ccr2RFP (B6.129(Cg)-Ccr2tm2.1lfc/J, Jax 017586), R26mTmG (B6.129(Cg)-Gt(ROSA)26Sortm4(ACTB-tdTomato,-EGFP)Luo/J, Jax 007676), R26nTnG (B6N.129S6-Gt(ROSA)26Sortm1(CAG-tdTomato*,-EGFP*)Ees/J, Jax 023537), LysMCre (B6.129P2-Lyz2tm1(cre)lfo/J, Jax 004781), Rac1f/f (Rac1tm1Djk/J, Jax 005550) mice were purchased from Jackson Laboratories.” The way this is phrased, only B6 mice lacking reporters are albino. There is no indication that the several reporter and DTR mice were bred onto a homozygous albino background. It is well known in the field that 2P imaging cannot be performed on non-albino black mice because of heat damage to melanocytes. This causes intense neutrophilic infiltration of the imaging site. Such neutrophil infiltration results in swarming as described in Lammermann et al. (Nature 2013 Jun 20;498(7454):371-5), who also show (see Fig. 3 in that Nature paper) that such swarms disrupt the local collagen matrix in the skin. Thus, if the experiments of the authors involve animals that are not albino, the imaging can induce neutrophil swarming and subsequent tissue damage that can compress microvessels, leading to the clots they see. Uderhardt et al. (Cell 2019 Apr 18;177(3):541-555) showed that RTM play a key role in preventing such neutrophil swarming and local tissue disruption and do so in a spatially constrained, RTM density-dependent manner. Hence, where macrophages are limiting, these effects of laser induced tissue damage and neutrophil swarming will lead to precisely the results reported here, but they will not be the results of ‘physiologic’ aging events, but rather, the consequence of experimental damage that the authors fail to consider.

We appreciate the referee’s concern, as we are also aware from our previous studies (PMID: 24097351, 25849774, 26110716, 27229141, 30269903) of the tissue damaging effects of performing multiphoton imaging on skin. Therefore, we would like to clarify that we did indeed cross all of the mice used in the study to the homozygous Albino B6 (B6(Cg)-Tyrc-2J/J, Jax 000058) background.

Further, it is widely accepted in the field of 2P imaging that all surgical preparations need to be tested using neutrophil reporter mice such as LysM-GFP animals to ensure that the preparation doesn’t routinely lead to such infiltration. Either histologic examination of the tissue imaged by 2P after the latter is completed can be used in each experiment to be sure this infiltration doesn’t occur intermittently even with a validated surgical method or a blood tracer can be added to the experiments to look for an absence of myeloid cell rolling in the microvasculature. Without evidence that such neutrophil response tracking has been done here and that only albino mice were used for all imaging experiments, it is simply impossible to credit the claims made, as a large part of the biology is potentially occurring “in the black”.

We appreciate this concern from the referee as well and would like to clarify that there are no “surgical preparations” in our studies. Unlike the model used in Uderhardt et al. (PMID: 30955887), we do not perform any surgery on our mice. Rather, we directly image the uninjured plantar skin of anesthetized mice as previously described in Rompolas, et al., (PMID: 27229141), Mesa, et al., (PMID: 30269903), Cockburn et al. (PMID: 36357619).

Regarding the referee's comment, "surgical preparation need to be tested using neutrophil reporter mice such as *LysM-GFP* animals to ensure that the preparation doesn't routinely lead to such infiltration" we have utilized *LysM-Cre* mice, which label neutrophils, monocytes, and macrophages (PMID: 32038641, 10621974, 32747818, 18261937, 24857755), with two different fluorescent reporters, *Rosa26-mTmG* mice and *LysM-Cre;Rosa26-dsRed*, for several multiday imaging experiments and see no evidence of *LysM*-expressing cell recruitment or swarming for any of our imaging preparations (revised ED Figure 1 and ED Figure 6f-h).

2. Even if proper breeding and use of albino mice is the case, many experiments in this paper have a similar major flaw. Laser damage has been shown in multiple papers to lead to rapid neutrophil infiltration and swarming (beside Lammermann and Uderhardt, see *J Invest Dermatol.* 2011 Oct;131(10):2058-68 and several other Weninger papers). Thus, even in albino mice, when laser ablation is used to remove the perivascular RTM, these experiments will cause intense neutrophil responses that will be unchecked due to the loss of the macrophages. The same problem arises when using DT in DTR mice and clodronate liposomes – both of these treatments are known to promote acute inflammation in response to the induced cell death. This is especially an issue when liposomes are directly injected into the skin site where imaging is then performed, as the injection itself will cause such a neutrophilic response. RTM can modulate this effect, giving a difference between the injected control mice and those receiving the liposomes. In short, the data are all consistent with the report of Uderhardt on the role of RTM in limiting tissue damage from neutrophils invasion and swarming, but this is never assessed or discussed in this paper. Further, the main thesis of RTM protection of vessels against clotting damage is not connected in the submission with the notion of local tissue damage and neutrophil responses, although these might occur at sites of clots and thus, one could imagine an amplification of local damage in regions devoid of RTM due to the addition of neutrophil activity on top of the blood flow limitation.

We appreciate the referee's comments and agree that it is important to understand if neutrophil swarming occurs in our system. We have performed several experiments to test this possibility and now provide comprehensive data to support that neutrophil swarming is not occurring after laser-induced clotting and/or laser-induced macrophage ablation.

1. **No recruitment of Gr-1+ (Ly-6G/Ly-6C) neutrophils or monocytes to laser-induced clots with macrophage ablation:** To assess any neutrophil or monocyte recruitment during laser ablations, we performed simultaneous laser-induced clots and CAM ablation experiments as in revised Figure 3, but with low-dose Gr-1-AF647 antibody I.V. labeling (Clone RB6-8C5) in *Csf1r-EGFP* mice, which labels both neutrophils and monocytes (AF647+/EGFP+) (PMID: 17438263). We also performed intradermal 28g insulin needle stick injuries in the plantar skin as a positive control for neutrophil swarming. Please see revised Extended Data Figure 6a-e that clearly shows that laser-induced clots and laser-induced CAM ablations do not recruit swarms of Gr-1+ cells at 6hr or 24hr post laser damage.
2. **No recruitment of *LysM*+ neutrophils or monocytes to laser-induced clots with macrophage ablation:** We utilized the *LysM-Cre* mouse, which targets neutrophils, monocytes, and macrophages (PMID: 32038641, 10621974, 32747818, 18261937, 24857755) crossed with the *Rosa26-dsRed* Cre-induced reporter, and performed simultaneous laser-induced clots and CAM ablation experiments as in revised Figure 3. Please see Extended Data Figure 6f-h that clearly shows that laser-induced clots and laser-induced CAM ablations do not result in recruitment of swarms of *LysM*-expressing cells at 24hr post laser damage.

3. **No recruitment of Csf1r-EGFP+ monocytes or neutrophils to laser-induced clots and/or macrophage depletion:** In Lammermann et al. and Uderhardt et al., it is clearly demonstrated that macro-lesions and macrophage-depleted micro-lesions generate tissue damage that will recruit swarms of both neutrophils and monocytes, the latter accumulating for at least one day after laser damage. To assess any monocyte or neutrophil recruitment during laser ablations and/or macrophage depletion, we used *Csf1r-EGFP* mice, which effectively labels both monocytes and neutrophils (express *Csf1r* mRNA but not protein) *in vivo* (PMID: 17438263, 32747818). Using this monocyte/neutrophil readout (*Csf1r-EGFP*), we have extensively looked at multiple timepoints (including 1 and 24 hours after laser damage) in our experiments and find no evidence for monocyte or neutrophil swarming after laser-induced clotting (revised Figure 3g and 6f; revised ED Figure 5), laser-induced macrophage ablation (revised Figure 4a), simultaneous laser-induced clotting and laser-induced macrophage ablation (revised Figure 3e), or macrophage depletion via clodronate (revised Figure 1g; revised ED Figure 4f) or diphtheria toxin (revised ED Figure 4b) treatment.
4. **Laser clot and macrophage ablation size is significantly smaller than micro- or macro-lesions described in Uderhardt et al.:** In the studies by Lammermann et al. and Uderhardt et al., the authors generate significantly larger sites of laser-induced damage ranging from ~30 microns (microlesion) to ~100 microns (macrolesions) in diameter. Both our laser-induced clotting and macrophage ablation models generate a damage site <10 microns in diameter, which is ~10-times smaller in area than microlesions and ~100-times smaller in area than macrolesions. Given that a reduction in lesion area can greatly diminish neutrophil recruitment (as Uderhardt et al. clearly demonstrate), we propose that our “nano-lesions”, which are ~10-times smaller in area than microlesions, may contribute to the differences we see following laser damage. Consistent with this idea, when we expand the laser damage area to 500µm² we find robust CCR2-dependent monocyte (CCR2-RFP+) recruitment into the damaged capillary niche (revised Figure 5e-g).

3. *The methods say that the observers are not blinded. This is a potential problem, as there can be a great deal of observer bias in the selection of the specific region(s) to image in animals or in the human skin samples. Two different levels of concern exist here. First, with the 2P experiments, the choice of the imaging region itself can bias the results, and at the very least, the microscopist should not know if the animals are deficient in RTM or their age when choosing regions to be assessed. Given that in most cases, regions with and without RTM will be obvious in the channel with the RTM fluorescent reporter, the imaging fields (ROIs) should be selected with this channel off and the identity of the animals masked so that truly random fields are collected and assessed. For the human skin data, the high power fields to be imaged should be selected with blinding to the age of the samples and preferably using random selection of the ROI.*

We appreciate the referee's comments and would like to clarify our efforts to have our observers blinded. For all animal imaging, 1mm x 2 mm imaging fields (ROIs) were acquired with only the second harmonic signal (collagen) as a reference guide to the same anatomical position (1mm proximal of the most proximal walking pad (<https://www.informatics.jax.org/cookbook/figures/figure6.shtml>)) on the mouse paw plantar skin. We have provided this information in the methods. For our human skin samples, the age group of samples were blinded to the researcher. Random ROI were selected for image analysis. Quantification of CAM coverage and capillary blood flow was performed blinded to minimize

researcher bias.

4. While it is quite possible that the results about aging and RTM disappearance are correct, the methods used to argue that older hosts have fewer perivascular RTM can be subject to artifacts. For the mice, the authors rely on expression and detection of fluorescent reporters to assess the presence and number of RTM. But if the expression of the relevant marker gene is simply lower on average in the RTM of aged mice and the laser power settings used for the imaging do not compensate for this (which would cause more intense damage and hence, is not a good solution to the problem), then one can miss RTM present in older animals because a greater fraction of them may be simply too dim to be counted. Better would be to do immunohistochemical analysis of the tissue using antibody staining for multiple markers, with the view that not all of them would suffer from such potential age related decrease and in any case, one can increase the laser power or gain during static imaging of fixed tissue without introducing inflammatory artifacts.

We appreciate the referee's concern that "the expression of the relevant marker gene is simply lower on average in the RTM of aged mice" could explain the decrease in capillary-associated macrophages. However, we would like to highlight that in addition to a single macrophage marker gene expression (i.e., *Csf1r-EFP*) that could change with age, we also performed lineage tracing with CAG-promoter-driven fluorescent protein expression in the *Rosa26* locus. Specifically, we showed that *Cx3cr1-CreER;Rosa26-mTmG* mice in revised Figure 4i had similar loss in capillary-associated macrophages with age.

To further examine the referee's concern, we have now tested additional markers to assess if these capillary-associated macrophages are indeed decreasing with age.

First, in addition to *Csf1r-EGFP* reporter mice, we utilized *Cx3cr1-GFP* mice and found a similar reduction in capillary-associated macrophages with age (revised ED Figure 1f).

Second, utilizing *LysMCre;Rosa26-mTmG* mice, we find a reduction in CAMs with age (revised ED Figure 1g).

Third, we performed immunohistochemical analysis on whole mount skin to assess expression of the canonical tissue macrophage marker, CD206 (*Mrc1*) (PMID: 34995099, 36450771, 28432199). We found that the vast majority of CSF1R⁺ cells are CD206⁺ in both young and old mice. Importantly, we find that CD206⁺ cells in the upper dermis (capillary niche) decreased in density with age (revised ED Figure 1h).

Therefore, we feel that the data collectively provide comprehensive evidence that these capillary-associated CSF1R-expressing macrophages are indeed progressively lost with age.

5. The CSF-1 studies in Fig 4 are quite intriguing. However, there should have been a third group with no treatment at all, because injection itself will lead to neutrophil responses that promote secondary monocyte infiltration and macrophage differentiation (See Lammermann et al. and Uderhardt et al.). If an increase in monocytes/macrophages improves capillary status in older mice as claimed, then even the PBS treated animals should be 'better' than the untreated mice of the same age due to this increase in tissue macrophages following repeated inoculation. If this is not the case, the authors need to explain why CSF-1 works but such 'physiologic' recruitment at damage sites does not. If the CSF-1 works by enhancing survival of existing RTM and recruited macrophages are unable to functionally compensate for the 'missing' RTM, that is a very important result, but more work is needed to examine this issue. In any case, the

enhancement of vessel integrity by increasing the coverage of RTM, if that is what the CSF-1 is doing, is simply another example of the density-dependent RTM protective effect shown in Uderhardt.

We appreciate the referee's comments but would like to clarify a few points from the data provided in our manuscript. First, while we do agree that a third group with no injection could be beneficial, we already do not find a significant increase in neutrophil, monocytes or macrophages (*Csf1r-EGFP* expressing cells – PMID: 17438263) in the PBS treated regions (Main Figure 6c). Specifically, we imaged the same skin regions immediately before the first injection, as well as 2 & 6 days after the last injection and found no changes to in the number of *Csf1r-EGFP* expressing cells. We would also like to clarify that to minimize injection-based inflammation, intradermal injections were performed at least 4mm proximal from our imaging region. While we acknowledge that some level of tissue stress/pressure or inflammation is likely generated, we did not find an increase in tissue macrophages as the referee suggests and therefore find this point to not be a major concern.

We agree with the referee that our data did not provide a clear mechanism for how CSF1-treatment increased CAM numbers in aged skin. To assess if CSF1-treatment modulated local CAM survival/proliferation or simply recruited new monocyte-derived macrophages from the blood, we performed a CSF1 treatment in chimeric bone marrow mice where we could track the relative expansion of local macrophages and recruitment of monocytes (revised ED Figure 10a-b). Consistent with our previous experiments, we found that PBS injected paws showed no significant increase in host or BM-derived monocyte/macrophages in our imaging areas (revised ED Figure 10c-d). Importantly we found that CSF1-induced CAM expansion did not alter the ratio of resident (GFP+) and recruited (GFP+/dsRed+) CAMs (revised ED Figure 10e). This result strongly suggests that CSF1 is driving local proliferation of the existing CAM population, as a relative increase in GFP+/dsRed+ CAMs over GFP+ CAMs would have suggested monocyte recruitment. Interestingly, we did not detect a relative increase in GFP+ CAMs either, which suggests that CSF1 treatment could promote proliferation in both long-term (> 9 months) as well as more recent monocyte-derived resident CAM populations.

The referee also states that *"If this is not the case, the authors need to explain why CSF-1 works but such 'physiologic' recruitment at damage sites does not."* We would like to clarify that we did show a role for damage-induced macrophage proliferation/recruitment in improving capillary repair in revised Extended Data Figure 9. Specifically, we performed larger laser damage to either capillary or overlying basal epithelial regions (500 μm^2) and found local capillary-associated macrophage proliferation (revised Figure 5h-i and ED Figure 9a-c). Additionally, we found that regions with damage-induced expansion of capillary-associated macrophages (which persisted for at least 4 weeks) also showed improved clot repair as compared to directly neighboring regions (Extended Data Figure 9i-l).

Collectively, the work provides novel insights for how aging macrophage populations in old mice can be rejuvenated through both growth factor and damage-induced expansion of resident macrophage populations.

6. As to the scholarship in the paper, as noted above, many of the salient observations made in his paper reprise work of Uderhardt showing how RTM prevent tissue damage (especially the data and conclusions associated with Fig. 2). Indeed the images provided by the authors do not show very local activity of macrophages, but exactly the extension of long processes from neighboring RTM to the site of damage/occlusion as reported in Uderhardt. Not only were these prior observations made using a laser injury model, but in two other situations that included

dystrophic mice and the diaphragm in mice deprived of RTM (albeit using DT with some of the same caveats I raise here). This prior work even used perfusion-fixed animals to show that ‘cloaking’ of damage sites by RTM occurs in the steady state. Uderhardt et al. also explicitly show the capture of cell debris, akin to clot clearance, and this activity was analyzed in greater detail in Science (2020 Jan 17;367(6475):301-305). Thus, the new information here is that RTM protection extends to capillaries by more or less the same mechanisms previously reported. IF the authors can show that the effect is unrelated to protection against neutrophil-induced damage, this would be novel, but otherwise it would conceptually be a reprise of the Uderhardt study.

We agree with the referee that Uderhardt et al. provides a major advancement for our understanding of the initial responses of resident macrophages to tissue damage and apologize for not referencing it in our main text. This has been corrected in the updated text.

As mentioned in the previous points above, we have provided evidence that neutrophil (and monocyte) swarming is not occurring in our models of laser-induced clotting or macrophage ablation. Therefore, as the referee suggests, our work highlights a novel concept of “nano-lesions” that persist without inflammatory neutrophil or monocyte responses but can still trigger local (<100µm) cloaking and Rac1-dependent phagocytosis from resident macrophages. This concept has broad implications for tissue aging as naturally occurring nano-lesions could contribute to cell loss (including macrophages) and minor tissue impairments that will accumulate over time as organisms age.

Additionally, earlier studies did not clarify long-term functional consequences of macrophage cloaking beyond preventing neutrophil accumulation within the first few hours after damage. Given that our imaging is non-invasive and non-surgical, we were able to overcome previous limitations and showed that initial macrophage cloaking leads to macrophage migration (partially CX3CR1-dependent) and persistent residency at the site of damage (revised Figure 2d-f), which alters the distribution/patterning of resident macrophages across the capillary niche.

Furthermore, our work is also novel in that it provides clear evidence that a resident macrophage population is not dependably replenished by proliferating neighbors within the niche or from blood monocytes, which further drives aberrant macrophage patterning and responses to future insults within an aged tissue niche (revised Figure 5a-d and ED Figure 9d-h).

Collectively, we agree with the referee that our work builds upon their previous foundational work but have also substantially advanced our understanding of the long-term cellular behaviors, fate decisions, and functional role of resident macrophages over the lifetime of an organism.

7. In Fig. 1d, there are many clots in even 2 month old mice. Why is this seen? When imaging skin, or any tissue, pressure from the objective can be an issue and in this case could lead to collapse of microvessels and stasis of RBC, as well as cell damage that promotes neutrophil swarming. The author state that they use a coverslip for their imaging and this can often lead to compression damage, both acutely at the global flow level as well as through local necrosis that causes neutrophil recruitment and swarming. This may be exaggerated in aged mice, because the issue is less compliant due to loss of elastin, making the vessels more susceptible to such artifactual obstruction / inflammation responses.

We appreciate the reviewers concerns and would like to clarify that we do not see any signs of overt inflammation such as recruitment of CSF1R-expressing, CX3CR1-expressing, CCR2-expressing, or LysM-expressing cells (which would include neutrophils) during our imaging. Furthermore, we regularly perform multiple revisits (over hours and days) of skin regions in the same mice and consistently find no signs of immune recruitment or overt changes to capillary blood flow.

To address the referee's concern, we assessed if utilizing a coverslip was altering capillary blood flow. Specifically, we performed intravital imaging of the same skin regions before and during coverslip application. We found no significant change in the number of capillaries with RBC flow (revised ED Figure 3a-b). It is important to highlight that unlike most skin imaging which occurs in the ear (which can easily lose blood flow from external pressure like a coverslip), we do all our intravital imaging on the plantar skin of the hind paw which has a much thicker epidermis and denser collagen network, providing additional resistance to tissue compression or deformation (PMID: 31633031).

Referee #4 (Remarks to the Author):

In this manuscript, Mesa et al. report interesting novel observations on the dynamics of capillary-associated macrophages (CAMs) in the skin capillary plexus and their impact on capillary functions for repair damage and during aging. The authors utilize powerful intravital two-photon microscopy combined with fluorescent reporter mice for longitudinal cell tracing to show that CAMs are lost with age greater than capillary reduction itself. They go on to show that CAM-less capillaries, either in aging or CAM-ablated cytotoxically or by laser, have reduced blood flow. They demonstrate that CAM loss is largely due to limited local CAM division and replenishment of lost CAMs, but find acute tissue injury such as blood clots can promote CAM proliferation and monocyte recruitment such that even aging capillaries can be rescued. Mechanistically, they identify that CSF1, a secreted growth factor in acute tissue injury, can rescue CAM replenishment in aging mice, which appears to rejuvenate capillary repair capacity.

Despite the complexity of the experiments, the authors manage to present the high quality and comprehensive data in a clear and appealing way to corroborate several main claims of how the CAMs regulate capillary aging. It was a pleasure to read! Overall, this manuscript offers a powerful discovery that CAMs are lost with aging and that CAMs are required for damage repair, maintain blood flow, and prevent such capillary aging-associated effects. With the CSF1 CAM recruitment experiments the authors also provide a potential therapeutic angle for improving blood flow in aged capillaries. In the end, there are a few select major concerns and minor suggestions that when addressed and implemented would help strengthening the claims further and shore up the findings.

Major concerns:

1. The authors claim that CAM loss promotes capillary aging because of slow blood flow within 14 days after CAM ablation and the association to fewer observed CAMs in aging. However, on its own the link seems a bit weak. It would strengthen the claims if the authors in addition to reduced blood flow after CAM ablation could establish an ongoing loss of capillaries and continued lack of CAM recovery in longer-term follow-up observations, akin to premature aging. For example, what happens for CAM ablation areas (Fig3d) after a couple months or more? Would it mimic the quite strong reductions of capillary density in aging after 10 and 18 months (Ext Data 3a)? Is there still no CAM replacement after a couple months of CAM ablations?

This is an interesting point raised by the referee and we agree that it would strengthen our claims. Therefore, to directly test the long-term effects of CAM dysfunction (*Rac1*-deficiency) on capillary maintenance during physiologically aging, we performed experiments where we compared capillary density in *Cx3cr1-CreER;Rac1fl/fl* vs *Cx3cr1-CreER;Rac1fl/+* mice after a three month period following tamoxifen administration. We found that over this 3-month period there was an acceleration in rate of capillary pruning (~20% reduction in density) in the skin of mice containing *Rac1*-deficient macrophages (revised Figure 3h-j). Therefore, it is likely that loss of *Rac1* dependent behaviors, such as phagocytosis of vascular and/or perivascular debris, directly contributes to impaired recovery and preservation of the skin microvascular network. In alignment with this working model, we found significant accumulation of unengulfed perivascular debris following laser-induced clotting in *Rac1* deficient mice (revised Figure 3f).

Collectively, this work provides deeper insight into the molecular mechanisms that underpin the link between capillaries and their associated macrophages during physiological aging. Important to the referee's concern, we provide stronger evidence that CAMs support capillary repair and preserve vascular density across the lifetime of the organism.

2. Fig. 1e shows that obstructed capillary flow rate is higher in CAM-less capillaries in 1-month old mice. What is the obstructed capillary flow rate in both CAM-present and CAM-less capillaries in aged mice? It would fit and strengthen the claims that CAMs presence and their repair capacity influence the blood flow in aging, if there is still relative obstructed capillary flow in CAM-less capillaries in aged mice.

We appreciate the referee's point, and therefore performed the suggested analysis across multiple ages to assess capillary RBC flow "*in both CAM-present and CAM-less capillaries in aged mice*". Specifically, we found across all ages tested (1, 2, 4, 10, and 18 months) a significant loss in RBC blood flow in capillaries lacking associated macrophages (revised Figure 1d-e; revised ED Figure 3c). Therefore, this observation supports a model where local macrophage loss in both young and old mice contribute to impaired capillary function.

To further address the referee's questions and assess the long-term fate of vessels that lack a CAM, we performed a 6-month time-course in *Cx3cr1-GFP;R26-mTmG* mice from 1 to 7 months of age. We found that while CAM coverage decreases with age as previously shown in Figure 1f, these "macrophage-less" capillaries are much more likely to be pruned and permanently eliminated for the remainder of our revisits (revised Figure 2a-c).

Collectively, these findings further strengthen the hypothesis that capillary-associated macrophages are required for maintaining proper capillary function, and without replenishment of new CAMs, these capillaries will eventually be pruned away contributing to vascular rarefaction, a hallmark of vascular aging (PMID: 37193857, 34326210).

3. As Rac1 is an important component for phagocytosis it is unsurprising, yet interesting to see, that Rac1 deletion leads to reduced blood clot clearance and reduced blood flow. It is not clear however if Rac1 deletion will influence blood flow in aged CAM-present capillaries to a level of CAM-less counterparts. If it does it would support phagocytosis as a main mechanism. If not, it would open interesting possibilities into CAMs secreting factors or exerting cell-cell contact effects that could affect the blood flow. It would be powerful to figure out how CAMs influence capillary aging.

We agree that the suggested experiments would provide additional mechanistic insight into the homeostatic role of Rac1 in maintaining capillary function. As mentioned in Major Concern #1, we directly tested the long-term effects on capillary aging in mice with Rac1-deficient CAMs. We found that over a 3-month period there was an accelerated rate of capillary pruning (~20% reduction in density) in the skin of mice containing Rac1-deficient macrophages (revised Figure 3h-j). Therefore, it is likely that loss of Rac-1 dependent behaviors, such as phagocytosis of vascular and/or perivascular debris, directly contributes to impaired recovery and preservation of the skin microvascular network across the lifetime of the organism.

4. The presence of high numbers of macrophages in the epidermis seemed surprising. It would make sense to make sure that the Csf1r reporter does not label other cells, such as Langerhans cells in the epidermis or other dendritic cells in the dermis.

We appreciate the referee's comments and have provided additional data to confirm that these capillary-associated cells are indeed macrophages.

First, Langerhans cells express CSF1R and are labeled with the *Csf1r-EGFP* reporter (PMID: 12393599).

Second, these capillary-associated cells express both CSF1R and CX3CR1 (revised ED Figure 1b), which would strongly suggest them to be of myeloid origin (macrophage/monocyte, dendritic cells) (PMID: 12393599, 29503738).

Third, we find that capillary-associated cells are also robustly labeled with LysM-Cre, which does not effectively label classical dendritic populations, but rather macrophages/monocytes and granulocytes (revised ED Figure 1g) (PMID: 10621974, 24857755).

Fourth, we find that the vast majority of CSF1R-expressing upper dermal cells also express the canonical tissue resident macrophage marker CD206 (*Mrc1*) (PMID: 34995099, 36450771, 28432199) (revised ED Figure 1h).

Lastly, we find that with all these labeling approaches there is a consistent loss in cell density with age (revised ED Figure 1). Therefore, collectively we feel this provides comprehensive evidence that these capillary-associated CSF1R-expressing cells are indeed macrophages that are progressively lost with age.

Minor concerns:

1. It is unclear whether CCR2-dependent replenishment (Ext data Fig. 8c) or CAM proliferation dependent division (Ext data Fig. 10) is the main source of replenishment. Does CSF1 promote CAM division or monocyte recruitment in aging mice?

This is an interesting point raised by the referee and we agree that our data did not provide a clear mechanism for how CSF1-treatment increased CAM numbers in aged skin. To assess if CSF1-treatment modulated local CAM survival/proliferation or simply recruited new monocyte-derived macrophages from the blood, we performed a CSF1 treatment in bone marrow chimeric mice in which we could track the relative expansion of local macrophages and recruitment of monocytes (revised ED Figure 10a-b).

Consistent with our previous experiments, we found that PBS injected paws showed no significant increase in host or BM-derived monocyte/ macrophages in our imaging areas CAMs

(revised ED Figure 10c-d). Importantly we found that CSF1-induced CAM expansion did not alter the ratio of resident (GFP+) and recruited (GFP+/dsRed+) CAMs (revised ED Figure 10e). This finding strongly suggests that CSF1 is driving local proliferation of the existing CAM population, as a relative increase in GFP+/dsRed+ CAMs over GFP+ CAMs would have suggested monocyte recruitment. Interestingly, we did not detect a relative increase in GFP+ CAMs either, which suggests that CSF1 treatment could promote proliferation in both long-term (> 9 months) and more recent monocyte-derived resident CAM populations.

2. The authors showed most macrophages in upper dermis are CX3CR1+ in 1-month old mice (Ext data Fig 1a and 1f), but Fig 1 only showed Csf1r+ macrophages during extended period up to aging. Is CX3CR1 the same percentage?

We thank the referee for the question. Along with data from the *Csf1r-EGFP* reporter mice, we now also provide macrophage aging density data utilizing the *Cx3cr1-GFP* mice and find a similar reduction in CAM density with age (revised ED Figure 1f).

3. Ext data Fig 1a and Ext data Fig 1f show CX3CR1+ both in upper dermis and lower dermis, but the numbers are very different. There are more CX3CR1+ cells in Ext data Fig 1f. Do CX3CR1+ cells in the lower dermis have the same division rate? If they have the same division rate, CX3CR1 decreases in the upper dermis could be related to macrophage migration.

We performed a similar lineage tracing experiment for both upper and lower CX3CR1+ dermal macrophages and found CX3CR1+ lower dermal macrophages have a significantly lower division rate as well as significantly higher loss rate (revised Figure 4d-g). Furthermore, in these experiments we intentionally induced sparse macrophage labeling to only track macrophages that were more than 500µm away from any other labelled cells (revised Figure 4e). From these datasets we found no evidence of migration of CX3CR1+ macrophages between the upper and lower dermal compartments (data not shown).

4. How does the macrophage population renewal relate to blood circulation? In Fig 1b, the size of blood vessels is different between upper dermis and lower dermis.

The upper dermis contains only the superficial capillary plexus, while the lower dermis is comprised of larger collecting vessels including arterioles, arteries, venules, and veins. Regarding renewal, as mentioned above we find lower dermal macrophage division is significantly lower (revised Figure 4d-g), but monocyte replenishment is significantly higher (revised Figure 4a-c), as compared to upper dermal macrophages.

5. In Ext data Fig 1a the figure legend describes “images of cells expressing Ccr2-RFP, Csf1r-EGFP or Cx3cr1-GFP”, but there is no Csf1r-EGFP.

We thank the referee for spotting this typo and have corrected the updated figure legend.

Response to Referees

Referee #1 (Remarks to the Author):

In this revised manuscript entitled “Niche-specific macrophage loss promotes skin capillary aging”, the authors report that capillary-associated macrophages (CAMs) in the upper dermis of the skin play an essential role in supporting capillary function. CAMs facilitated homeostatic capillary blood flow (measured by 3rd harmonic generation of RBCs in blood vessels) and maintained capillary density, and loss of CAMs and reduction of their capillary coverage with age resulted in impaired blood flow and pruning of CAM-unassociated capillaries. Mechanistically, authors show CX3CR1 to be essential for sensing laser-induced clotting and reperfusion. RAC1 was required for laser-induced clot removal and reperfusion. CAMs were maintained more by self-renewal than replacement by monocytes, but CAM self-renewal under paced its loss, ultimately resulting in progressive reduction with age. Finally, authors showed that local CAM replenishment by administering CSF1-FC was sufficient to improve capillary blood flow and capillary repair and reperfusion following laser-induced clot formation in aged mice.

The rather heavy focus on macrophages, most of which are incremental in nature, distract from the novel aspect of this work, which is the phenomenon that CAMs contribute to homeostatic blood flow of capillaries, the mechanisms of which are still insufficiently addressed.

Major comments

1. While the authors additionally showed that flowing RBCs within vessels can be further identified by labeling z-stacks in different colors (revised ED Figure 2d, f), this method has not been applied to other relevant Figures and ED Figures. In line with this, it is somewhat still unclear whether the reduced capillary blood flow and/or obstruction of capillaries in physiological condition is indeed due to the clot formation. Authors should consider applying the z-stack method and quantification to some representative data throughout the manuscript to reliably demonstrate reduced capillary blood flow and physiological clot formation. Additionally, please provide quantifications without both RBC (3rd HG) and RhDex labeling to represent capillary occlusion and clot formation in revised ED Figure 2.

We thank the referee for this helpful comment. We agree this will provide a clearer representation of RBC flow to the reader. Therefore, as suggested by the referee, we have performed the same pseudo-coloring z-stack method for several major experiments in the main figures, including Figure 3a, Figure 3c, Figure 3e, Figure 6d, and Figure 6f.

2. Authors have made effort to address mechanisms that underly CAM-mediated regulation of blood flow by showing requirement of CX3CR1 and RAC-1 using Cx3cr1-gfp/gfp mice and Cx3cr1-crexRac1f/f mice, respectively. Authors also show that impaired blood flow and clot removal in aged mice can be improved by injecting CSF1-FC protein. All of these are interesting, but each are incomplete, expected, or lack connection.

- CAM-mediated restoration of clots require CX3CR1, but the observation is rather

incomplete without showing the source of the ligand, whether it is activated by clots. Is it possible that clots activate CX3CL1 in capillaries?

- As commented in the last review, impaired phagocytosis in the absence of RAC-1 is expected. Presumably, the functional alteration of CAMs in the absence of RAC-1 is not limited to phagocytosis.

- Rescuing blood flow in aged mice with CSF1-FC protein has translational impact. However, information on which cells provide CSF1 to macrophages is lacking. Do cellular sources of CSF1 decrease with age, or does the expression of CSF1 in tissue decrease with age? Would deletion of CSF1 from the main source lead to impaired blood flow?

We appreciate the referee's comments and would like to provide additional insight into the cellular source of CSF1 and its activity on dermal macrophages. While we believe that this point may be beyond the primary scope of the study, previous work as well as some of our own data support a model in which (1) skin fibroblasts provide CSF1 to dermal macrophages, (2) loss of CSF1 expression leads to impaired skin and immune cell functions, and (3) the fibroblasts are progressively lost in both human and mouse dermis with age.

First, it is well established from *in vitro* studies that macrophages and fibroblasts can maintain a stable two-cell circuit by providing the other's growth factors (i.e. fibroblast expression of CSF1, and macrophage expression of platelet-derived growth factors - PMID: 29398113, 35930670).

Second, it was recently shown in skin that fibroblast-specific loss of CSF1 expression leads to broad depletion of skin dermal macrophages and to altered physiological processes, including impaired wound healing, alterations in ECM formation, and resistance to bacterial infection (PMID: 37402364, Vollmers AC et al., bioRxiv 2024).

Third, it has been demonstrated that fibroblast density is reduced with age in both mouse and human skin (PMID: 30415836, 10692106).

Finally, to assess local CSF1 expression and fibroblast density in the upper dermal capillary niche (<25µm from epidermis), we adapted the spatial transcriptomic approach NICHE-seq (PMID: 29217582) to specifically photo-label and isolate these spatial-defined cell populations from young (2-month), middle aged (9-month), and old (18-month) mice for single-cell RNA-seq (Reviewer Figure 1a,b). From this approach, we identified four stromal populations present in the upper dermis (Reviewer Figure 1c). Consistent with previous literature, we found that CSF1 was predominately expressed by a PDGFRα/β+ fibroblast population (Reviewer Figure 1d). Interestingly, we did not detect a significant change in CSF1 expression during normal skin aging (Reviewer Figure 1d). Rather, we found that PDGFRα/β+ stromal cell density was significantly decreased in old mouse skin (Reviewer Figure 1e,f).

Collectively, we feel that our results and multiple previous studies provide strong evidence that fibroblasts represent a major and functionally relevant source of CSF1 in the skin. It will be important for future studies to directly assess the likely interplay between age-associated fibroblast and macrophage loss across different tissue microenvironments.

[REDACTION]

Reviewer Figure 1. CSF1-producing stromal cells are lost from upper dermal skin niche with age. **a**, Representative optical sections of upper dermal capillary niche before and after 810nm two-photon stimulation in *UBC-PAGFP* mice (PMID: 21074050). **b**, Scheme of NICHE-seq of skin upper dermis. Following photo-stimulation, skin is digested and GFP+ cells are isolated for single-cell RNA sequencing. **c**, Annotated UMAP from all age groups. **d**, Violin plots of *Csf1*, *Pdgfra*, and *Pdgfrb* expression across all clusters and age groups. **e**,

Representative upper and lower dermal optical sections from whole mount skin samples of young (4-month-old) and old (18-month-old) *Csf1r-EGFP* mice stained with AF647 anti-mouse PDGFR α/β + (clone Y92) antibody. **f**, Quantifications for PDGFR α/β in the upper and lower dermis (n = 4 mice in each group; two 500 μm^2 regions per mouse; CAM density (4 vs 18-month-old) was compared by two-way repeated measures ANOVA and Fisher multiple comparison tests; mean \pm SD). Scale bar, 50 μm .

Others:

1. *Figure 2g Capillary clot recovery following CAM ablation: Did authors ablate CAM in this figure rather simply induced clot formation to see the role of CX3CR1 signaling? Please clarify the title of the figure.*

We thank the referee for catching this typo. To clarify, we did not perform CAM ablations in this experiment. Rather we only performed laser-induced capillary clots. The figure panel text for Figure 2g has been corrected.

2. *Authors show that donor-derived macrophages repopulate in the lower dermis, while CAMs in the upper dermis remain of host origin (Figure 4b). It is also shown that the origin of CAMs does not contribute to blood flow restoration in Extended Data Figure 3d. How can this statement be made when donor- and host-derived macrophages are distributed in different layers? Are clots effectively induced in the bigger vasculature in the lower dermis?*

We are sorry for not making this clear enough in the text. While the upper dermal macrophage population has very limited monocyte contribution, we do find ~5% of CAMs are BM-derived by 10 weeks post BM chimera. Therefore, we checked capillary blood flow in these mice comparing the ~95% host CAM covered capillaries with the ~5% donor CAM covered capillaries (Extended Data Figure 3d). We are thus comparing capillary flow within the upper dermis.

3. *In IHCs shown in Extended Data Figure 3g, the spatial association between upper dermal capillaries and CAMs is unclear with the single-color IHC.*

We are sorry for not clearly explaining this in the figure legend and have clarified in the updated figure legend. Specifically, this analysis did not look at the association of macrophages and capillaries, but rather the density of these two cell types in the human upper dermis. The upper dermis was defined as within 100 μm from the epidermal basement membrane boundary, which is highlighted in Extended Data Figure 3g by a yellow dashed line.

4. *Patient information is still missing in the revised manuscript. Additionally, the number of human samples analyzed (5 younger group, 6 older group, 11 total) is different from that described in the manuscript (10 written consents were obtained).*

We thank the referee for catching this text error. We have corrected this in the text to reflect the 11 samples analyzed (n = 5 young group, n = 6 old group).

Referee #2 (Remarks to the Author):

The authors have successfully addressed the major points raised in the initial review, and their new data provide valuable insights, unveiling interesting findings. However, the integration of these findings suggests a shift in the study's core message, which should now be reflected in the title and abstract and main text. Specifically, the revised data indicate that the loss of macrophages impacts capillary function independently of aging, with aging primarily serving as a confounding factor leading to macrophage depletion. Please see point 1 below for further clarification. Otherwise, the data is convincing and only minor points remain (see below).

1. The current title and abstract emphasize terms such as “niche,” “contribute,” and “aging,” suggesting that niche aging is the central theme of the study. However, the revised dataset indicates that the interaction between macrophages and vasculature, rather than aging itself, drives the observed capillary dysfunction. Aging primarily acts as a factor leading to macrophage depletion rather than being an intrinsic driver of the observed effects. In the authors’ rebuttal, they state: “To our surprise we found that regions of aged skin that retained CAMs also retained their ability to repair from laser-induced capillary damage, while regions in the same mice that lacked associated macrophages showed significant impairment to capillary reperfusion (revised Figure 3b).” This finding is, in fact, not surprising if aging serves mainly as a cause for macrophage loss rather than an active driver of capillary dysfunction. Additionally, other experiments in the revision point toward a “model that local macrophage loss, rather than just biological age contributes to capillary function.” Since the authors themselves draw this conclusion with the new datasets, maybe then the word “niche” should disappear from the manuscript in general, and the macrophage loss (due to age, or physical/chemical ablation) leading to tissue dysfunction should be highlighted instead.

This is an interesting point from the referee, and we would like to provide some clarity on how we use the term niche. We utilized the term niche throughout the manuscript to distinguish between sub-tissue compartments of the skin. From this, we identified niche-specific rates of macrophage loss with age (Figure 1 – upper dermal macrophage loss) as well as niche-specific macrophage self-renewal strategies (Figure 5 - monocyte recruitment vs local proliferation). Therefore, we agree with the referee that general or undefined “niche aging” is not the central theme of the study. Rather, the central theme of the study is that tissue resident macrophage self-renewal is niche-specific, which can lead to localized macrophage loss and tissue dysfunction.

2. The manuscript predominantly uses an unpaired Student’s t-test for statistical comparisons. However, this test is not appropriate for all datasets, especially when data

are not normally distributed and/or more than two conditions are being compared. For example, in Figure 4C, an ANOVA test would be more appropriate than a t-test. I recommend carefully reviewing all datasets and applying the appropriate statistical tests throughout the manuscript to ensure robust and accurate conclusions.

We appreciate the referee's comment and have carefully reviewed and updated our statistics. Specifically, we have performed both one-way and two-way ANOVA tests with either Tukey or Fisher multiple comparison tests (dependent on sample number) for the appropriate experiments that compare 3 or more conditions. For Figure 1h, capillary blood flow quantifications between treatment groups at the final time point (Day 13) no longer reached statistical significance and therefore we adjusted the time course in the main figure. However, in all other cases the updated statistical tests did not result in changed significance between conditions and therefore did not affect our conclusions drawn from the work done in this study.

3. For clarity and ease of review, I suggest that all changes in the revised manuscript be highlighted. This will facilitate a more efficient evaluation of the updated text and ensure that all key revisions are easily identifiable.

We thank the referee for the comment. We have yellow highlighted changes in the main text and figure legends.

Referee #3 (Remarks to the Author):

The authors have added a substantial amount of new data as well as text revisions in response to the original set of reviews. These additions and clarifications address many of the major concerns raised in response to the original submission, often by several of the referees and the major conclusions of the authors now appear more firmly supported. I leave it to the other reviewers to determine if the new data are sufficient to convince them that the issues of myeloid cell identity, effects of macrophage aging vs. changes in the surrounding tissue niche, the role of phagocytosis and clearance of debris presumably controlled by Rac, and so on are now adequately resolved. I concentrate here on the points raised just in my review:

1. In response to my point about use of albino mice for 2P imaging of animals on the black (B6) background, they write

“We appreciate the referee’s concern, as we are also aware from our previous studies (PMID: 24097351, 25849774, 26110716, 27229141, 30269903) of the tissue damaging effects of performing multiphoton imaging on skin. Therefore, we would like to clarify that we did indeed cross all of the mice used in the study to the homozygous Albino B6 (B6(Cg)-Tyrc-2J/J, Jax 000058) background.”

While I am surprised that this critical point was omitted in the original Methods, this assurance is an important one in evaluating the data and I accept that they have not inadvertently caused severe local inflammation during their 2P imaging sessions due to melanocyte damage.

We thank the referee for the comment.

2. I also asked that they check for neutrophil infiltration during their imaging sessions apart from the possible effect of melanin-related heat damage. They responded

“We appreciate this concern from the referee as well and would like to clarify that there are no “surgical preparations” in our studies. Unlike the model used in Uderhardt et al. (PMID: 30955887), we do not perform any surgery on our mice. Rather, we directly image the uninjured plantar skin of anesthetized mice as previous described in Rompolas, et al., (PMID: 27229141), Mesa, et al., (PMID: 30269903), Cockburn et al. (PMID: 36357619). Regarding the referee’s comment, “surgical preparation need to be tested using neutrophil reporter mice such as LysM-GFP animals to ensure that the preparation doesn’t routinely lead to such infiltration” we have utilized LysM-Cre mice, which label neutrophils, monocytes, and macrophages (PMID: 32038641, 10621974, 32747818, 18261937, 24857755), with two different fluorescent reporters, Rosa26-mTmG mice and LysM-Cre;Rosa26-dsRed, for several multiday imaging experiments and see no evidence of LysM-expressing cell recruitment or swarming for any of our imaging preparations (revised ED Figure 1 and ED Figure 6f-h).”

I need to state that nearly all these experiments look at 24hrs after the perturbation; this is a very late time point as neutrophil responses to tissue damage occur with 1-2 hours and are usually supplanted by monocyte recruitment by the 24 hr time point, so I do not find most of these data highly relevant to my point. Further, while they are correct that they did not do invasive surgery, they use a cover slip preparation and simple pressure on tissues can cause death of a few cells, sufficient to lead to neutrophil infiltration, Indeed, when Jackson Egen was in my group developing a cover-slipped liver imaging prep, he noticed neutrophils migrating to selected location in the liver parenchyma, He used propidium iodide in vivo to look for dead cells and showed that these neutrophils were migrating towards a limited number of individual dead hepatocytes. This was caused by the pressure of the coverslip and relieving this pressure prevented such neutrophil responses. These data also speak to the issue of whether a 10um lesion is “too small” to evoke neutrophil responses, This is incorrect, Uderhardt did experiments with lasers tuned to kill just a single cell and saw neutrophil swarming in the absence of tissue resident macrophages – not all his lesions were the 30u microlesions mentioned by the authors. He also observed neutrophil accumulations around single dead diaphragm muscle fibers in macrophage depleted hosts. Further, as I just stated Egen saw neutrophil responses to single dead hepatocytes and Robey studying toxoplasma saw swarming in response to death of single macrophages (PMID: 18718768). Thus, this issue can only be resolved by the type of careful reporter-based imaging I suggested.

We have discussed with the referee directly to resolve these points and have provided a summary below.

3. They go on to address this issue in detail, stating

“1. No recruitment of Gr-1+ (Ly-6G/Ly-6C) neutrophils or monocytes to laser-induced clots with macrophage ablation: To assess any neutrophil or monocyte recruitment during laser ablations, we performed simultaneous laser-induced clots and CAM ablation experiments as in revised Figure 3, but with low-dose Gr-1-AF647 antibody I.V. labeling (Clone RB6-8C5) in Csf1r-EGFP mice, which labels both neutrophils and monocytes (AF647+/EGFP+) (PMID: 17438263). We also performed intradermal 28g insulin needle stick injuries in the plantar skin as a positive control for neutrophil swarming. Please see revised Extended Data Figure 6a-e that clearly shows that laser-induced clots and laser induced CAM ablations do not recruit swarms of Gr-1+ cells at 6hr or 24hr post laser damage.”

While 6 hrs is probably early enough to see swarms if they existed, I do not understand the hesitancy to do imaging at earlier times and using the LysMGFP animals rather than low amounts of anti-GR1 ab as a label.

2. No recruitment of LysM+ neutrophils or monocytes to laser-induced clots with macrophage ablation: We utilized the LysM-Cre mouse, which targets neutrophils, monocytes, and macrophages (PMID: 32038641, 10621974, 32747818, 18261937, 24857755) crossed with the Rosa26-dsRed Cre-induced reporter, and performed simultaneous laser-induced clots and CAM ablation experiments as in revised Figure 3. Please see Extended Data Figure 6f-h that clearly shows that laser-induced clots and laser-induced CAM ablations do not result in recruitment of swarms of LysM-expressing cells at 24hr post laser damage.”

Again, 24 hrs is not the right time point for this experiment, as the initial neutrophil response will have been replaced by a monocytic response by then, as observed in Uderhardt and also by Kubes (PMID: 25800956).

We have discussed with the referee directly to resolve these points and have provided a summary below.

3. *“No recruitment of Csf1r-EGFP+ monocytes or neutrophils to laser-induced clots and/or macrophage depletion: In Lammermann et al. and Uderhardt et al., it is clearly demonstrated that macro-lesions and macrophage-depleted micro-lesions generate tissue damage that will recruit swarms of both neutrophils and monocytes, the latter accumulating for at least one day after laser damage. To assess any monocyte or neutrophil recruitment during laser ablations and/or macrophage depletion, we used Csf1r-EGFP mice, which effectively labels both monocytes and neutrophils (express Csf1r mRNA but not protein) in vivo (PMID: 17438263, 32747818). Using this monocyte/neutrophil readout*

(Csf1r-EGFP), we have extensively looked at multiple timepoints (including 1 and 24 hours after laser damage) in our experiments and find no evidence for monocyte or neutrophil swarming after laser-induced clotting (revised Figure 3g and 6f; revised ED Figure 5), laser-induced macrophage ablation (revised Figure 4a), simultaneous laser-induced clotting and laser-induced macrophage ablation (revised Figure 3e), or macrophage depletion via clodronate (revised Figure 1g; revised ED Figure 4f) or diphtheria toxin (revised ED Figure 4b) treatment.”

These are the most relevant data and I must admit that the evidence presented using this model is very clear about a lack of myeloid cell (neutrophil or monocyte) accumulation early or late after any of the clotting or depletion manipulations used by the authors. I cannot explain why they find that laser ablation of macrophages fails to incite a response, given all the data I note above about such responses to individual cell death in various tissues as studied by others, but accept that they have addressed the issue carefully and do not find such responses in their model.

We have discussed with the referee directly to resolve these points and have provided a summary below.

4. Laser clot and macrophage ablation size is significantly smaller than micro- or macro-lesions described in Uderhardt et al.: In the studies by Lammermann et al. and Uderhardt et al., the authors generate significantly larger sites of laser-induced damage ranging from ~30 microns (microlesion) to ~100 microns (macrolesions) in diameter. Both our laser-induced clotting and macrophage ablation models generate a damage site <10 microns in diameter ...”.

I have addressed this issue above – even single cell ablation or death causes neutrophil recruitment in multiple models including those without overt large scale tissue inflammation such as the liver cover slip situation or laser-induced death of a single cell.

Summary of discussion about neutrophil swarming:

1. One of the mouse strains used in the study, CSF1R-EGFP, also labels neutrophils.

We have provided sufficient imaging data at earlier time points, such as 1 and 6 hours, using the CSF1R-EGFP reporter mouse strain that labels macrophages, monocytes and neutrophils (PMID: 17438263, 32747818) and find no evidence for neutrophil swarming during CAM ablation and/or capillary laser-induced clotting.

2. The tissue damage threshold for neutrophil swarming may be tissue-specific.

Regarding the comment that a single-cell death could trigger neutrophil swarming: We do not contest the point that death of a diaphragm muscle fiber or a pathogen infected hepatocyte can lead to neutrophil swarming. Rather, we agree that the more harmonious conclusion would be that the sensitivity or threshold for neutrophil swarming may be tissue- and/or context-specific. Therefore, skin may require more cell death, tissue

damage, or inflammation to trigger neutrophil recruitment. Perhaps this is evolutionarily beneficial for barrier tissues generally, as they are constantly dealing with external stress. In contrast, this suggests that tissues more sensitive to neutrophil swarming, such as muscle or liver (as suggested by the referee), may be even more susceptible than dermis to resident macrophage loss and local tissue dysfunction with age.

3. Updated discussion section to reflect these points.

We have provided an updated discussion section in the main text that better reflects the limitations of the current study and highlights the potential for tissue-specific sensitivity to neutrophil swarming, which should be further investigated in the future.

4. The methods say that the observers are not blinded. This is a potential problem, as there can be a great deal of observer bias in the selection of the specific region(s) to image in animals or in the human skin samples. ... The authors answer addresses this issue.

5. While it is quite possible that the results about aging and RTM disappearance are correct, the methods used to argue that older hosts have fewer perivascular RTM can be subject to artifacts. For the mice, the authors rely on expression and detection of fluorescent reporters to assess the presence and number of RTM. But if the expression of the relevant marker gene is simply lower on average in the RTM of aged mice and the laser power settings used for the imaging do not compensate for this (which would cause more intense damage and hence, is not a good solution to the problem), then one can miss RTM present in older animals because a greater fraction of them may be simply too dim to be counted....

The authors have now used several orthogonal methods to address this concern and I do not believe that my concern about signal loss can explain their data.

6. The CSF-1 studies in Fig 4 are quite intriguing. However, there should have been a third group with no treatment at all, because injection itself will lead to neutrophil responses that promote secondary monocyte infiltration and macrophage differentiation (See Lammermann et al. and Uderhardt et al.). If an increase in monocytes/macrophages improves capillary status in older mice as claimed, then even the PBS treated animals should be 'better' than the untreated mice of the same age due to this increase in tissue macrophages following repeated inoculation. The authors nicely clarify why this is not a likely scenario, although it relies on the absence of detection of myeloid cell influx under conditions that as I state above, I find remarkable. However, that is what their data show, and given the findings, I do not think my objection here is valid.

7. I leave the issue of novelty to the editors.

8. In Fig. 1d, there are many clots in even 2 month old mice. Why is this seen? When imaging skin, or any tissue, pressure from the objective can be an issue and in this case could lead to collapse of microvessels and stasis of RBC, as well as cell damage that promotes neutrophil swarming.

The authors reprise their statements about lack of overt inflammation in their preparations but this u= is not the issue. Rather, it is the pressure from the coverslip that can cause cell damage / death as I note above in the work of Egen. ED Fig. 3AB show that loss of flow is minimal in this experiment in 6 month old mice – whether in older mice with stiffer tissue more pressure is transmitted and more effects are seen is unknown.

We appreciate the referee's concern. While it is possible that tissue stiffness changes with age could lead to some increased pressure on the tissue, we still do not find any evidence for acute decreases in capillary blood flow (RBC flow via Third Harmonic Generation) or neutrophil accumulation (via *Csf1r-EGFP* mice, *LysMCre* mice, or for Gr-1 fluorescent antibodies) during our repeated imaging sessions. Therefore, we find this unlikely to explain our results or significantly alter our conclusions.

Referee #4 (Remarks to the Author):

In this revised manuscript, Mesa et al. provide many new experiments and explanations that to a great extent sufficiently address concerns raised by us and other referees regarding their innovative intravital imaging study of macrophage function in the skin capillary plexus and its decline with aging. The addition of long-term fate mapping of CAM-less capillaries; cross-validation of macrophage subpopulations using improved cellular markers; assessment of turnover dynamics using bone marrow chimeras; and substantive supplementary data ruling out concerns of neutrophil swarming and tissue damage from their laser-induced clotting model all bring additional strength to the main conclusions.

Several issues were nicely addressed as described below. Yet, one main concern remains: The long-term outcome of CAM ablation is still unclear, either the one after laser- or DT-mediated killing (now in new Figure 5a-d; previous 3d-g) or after clodronate ablations (now in ED Figure 4f,g; previous main Fig 1g,h). In previous main concern #1, we asked whether CAM ablations would mimic early in age, and longer than the only the observed 2 weeks, the strong reductions of capillary density seen in aging after 10 and 18 months. And equally important, whether after CAM ablations there would be still no CAM replacement after a more meaningfully longer period than the 14 days (laser, DT) or 13 days (clodronate).

*Answering these two points would strongly support the hypothesis that CAM loss promotes capillary aging due to slow blood flow early after CAM ablation and akin to reduced CAMs in aging. The presented CAM dysfunction assays by *Rac* deficiency resulting in induced capillary pruning, while highly interesting, do not fully address the experimental loss of CAMs issue (like CAM loss in age). If any new significant macrophage numbers would come in soon after 2 weeks to rescue CAM loss and impaired blood flow, it would indicate that additional factors exist that inhibit macrophage proliferation or migration in aging.*

We agree with the referee that addressing these points would further clarify and support the role of CAM loss in capillary aging. To this end, we repeated our laser-induced CAM ablation experiment and extended our revisits to 8-weeks post ablation.

We find that:

1) CAM replacement is not significantly improved at 8-weeks, when compared to 2-weeks post ablation (Figure 5d).

2) We find a relatively small (~10%), but significant reduction in the local density of capillaries with blood flow (via RBC flow) by 8-weeks post-ablation when compared to neighboring control regions (Extended Data Figure 9a).

Therefore, as the referee suggests, this CAM ablation data along with Rac deficiency data support a clear role for CAM loss/dysfunction in driving age-associated capillary decline.

Other well-addressed issues:

Although the initial manuscript already provided substantial evidence connecting the absence of CAMs with obstructed capillary blood flow, the connection to aging and loss of capillary density was not so clear. With revised Figure 2a-c the authors now demonstrate that CAM-less capillaries are significantly pruned in the aged mouse. Examination of capillary obstruction in the presence/absence of CAMs across different aging time points in ED Figure 3c further validates their pivotal role in repair and maintenance of capillary function.

The addition of Cx3cr1 and LysM quantifications in the upper dermis of young and old mice help support the previous observations using the global macrophage reporter Csf1r and increase confidence in the observed dynamics from their Cx3cr1 single-cell labelling experiments. The overlap between CD206+ and Csf1r+ cells is convincing, though the legend descriptions could be improved for clarity. Overall, their definitions of resident macrophages within the capillary niche are much improved and convincing.

The use of bone marrow chimeras is a sophisticated way of showing low contribution of monocytes to the upper dermis, eliminates monocyte recruitment as a mechanism for CAM expansion following CSF1 treatment, and posits a local self-renewal strategy contributing to niche disorganization. The additional data tracking single Cx3cr1+ macrophages in the upper and lower dermis properly address our concerns regarding movement between compartments.

Minor: the figure legend in ED Fig 1a still lists Csf1r-EGFP, even though there is none and it was stated in the response that it was corrected. It was not corrected.

We apologize to the referee for overlooking this comment in the first resubmission. It has now been resolved.

Referee comments:

Referee #1 (Remarks to the Author):

The authors have done an excellent job in responding to my and fellow reviewers' comments. I think they provide compelling evidence on the role of capillary associated macrophages in maintaining skin capillary function. One comment the authors have not responded to is the human subject information (age, sex, site of biopsy). Related to this, the IHC that is provided in Extended Figure 3 to make the case that human CAMs also decline with age, does not clearly show the capillaries in aged skin. Two-color staining (e.g. CD68 + CD31) would make a stronger case.

We thank the referee for their comments. We have updated our methods section in the main text to include the human subject information used for this study including age range and site of biopsy. We unfortunately are unable to perform additional staining as suggested due to the limited amount of samples we received. However, we have provided additional clarity for how capillaries were scored based on a combination of morphology and ERG staining from clinical pathology researchers.

“Human skin samples

Written informed consent was obtained for postmortem examination from next of kin for all patients. Clinical information and laboratory data were obtained from the electronic medical record. Sex and gender information was not used. Patients in young group were below 40 years of age (19 - 37 years old). Patients in old group were above 40 years of age (79 - 97 years old). Patients with skin or vascular pathologies were excluded. Skin samples were obtained from the anterolateral chest and fixed in 10% formalin for at least 24h prior to processing. Slides were stained with hematoxylin and eosin, CD68 (Clone 514H12) and ERG (Clone EPR3864). Macrophages and capillaries were identified using a combination of morphology, CD68 and ERG staining. The age group of samples were blinded to the researcher and counting was performed on at least eight high-power fields (40x) within 100um of the epidermis.”

Referee #4 (Remarks to the Author):

In this revised version, overall, the authors have sufficiently addressed our concerns regarding a longer-term outcome of laser-induced CAM loss. Specifically, the authors now included an additional revisit time point at 8 weeks following laser-induced ablation of CAMs in the upper dermis (Fig 5d) and similarly quantify the frequency of CAM replacement as with the 2-week time point. The sustained significant reduction in CAM replacement when compared to the “CAM loss + clot” model continues to support the conclusion that local tissue damage is required to replenish CAMs in the upper dermal niche and promote their renewal. Although they do not quantify the frequency of CAM replacement with the “CAM loss + clot” model at 8 weeks, the authors demonstrate elsewhere in their manuscript (e.g. Fig 5h, Fig 3e, ED Fig 5c) that blood flow is

reestablished as early as 7 days following laser-induced capillary clotting/damage. Therefore, it can reasonably be inferred that the frequency of CAM replenishment in the “CAM loss + clot” model represents a new homeostatic condition directly comparable to the 8-week time point without clotting.

The authors also now include an analysis of capillary blood flow at 8 weeks following laser-induced CAM loss (ED Fig 9a). This directly addresses the original concern as to whether laser-induced CAM loss in young mice mirrors the reduced capillary density brought about by the homeostatic loss of CAMs in the aged mouse.

Ideally, the authors would have also extended their revisit window to 8 weeks following chemical ablation of CAMs using high dose IP injection of DT and observed a similar reduction in CAM recruitment and capillary blood flow/density. Additionally, the described relationship between tissue damage and CAM recruitment could have been strengthened by performing laser-induced capillary clots at the 8-week time point and observing whether CAMs are still recruited a week or even a few days later. Even in the absence of this, the data provided in this revised manuscript has successfully addressed our initial concerns and supports the model that the loss of CAMs in the upper dermal niche promotes capillary aging, and that their recruitment is dependent on a tissue damage response.

To that point, we agree with other referees in that the central theme of the study appears to be the interaction between macrophages and vasculature, rather than their absolute presence/absence in the upper dermal niche. This focus could be better reflected in the title, perhaps as “Loss of macrophages in the upper dermal niche promotes skin capillary aging.”

We thank the referee for the insightful comments. We agree that the paper provides a strong focus on macrophage and vasculature interactions. However, an equally central focus of the study, in our opinion, is that skin macrophage self-renewal strategies are niche-specific (in particular dermal macrophages - upper vs lower dermis). Therefore, to better reflect these two central points of the study, we have edited the title as follows, “Niche-specific dermal macrophage loss promotes skin capillary aging”.